# GradMetaNet: An Equivariant Architecture for Learning on Gradients

**Yoav Gelberg**[*]
University of Oxford
yoav@robots.ox.ac.uk

**Yam Eitan**[*]
Technion
yam.eitan@campus.technion.ac.il

**Aviv Navon**
Independent Reseracher

**Aviv Shamsian**
Bar-Ilan University

**Theo (Moe) Putterman**
UC Berkeley

**Michael Bronstein**
University of Oxford, AITHYRA

**Haggai Maron**
Technion/NVIDIA

## Abstract

Gradients of neural networks encode valuable information for optimization, editing, and analysis of models. Therefore, practitioners often treat gradients as inputs to task-specific algorithms, e.g. for pruning or optimization. Recent works explore *learning* algorithms that operate directly on gradients but use architectures that are not specifically designed for gradient processing, limiting their applicability. In this paper, we present a principled approach for designing architectures that process gradients. Our approach is guided by three principles: (1) equivariant design that preserves neuron permutation symmetries, (2) processing sets of gradients across multiple data points to capture curvature information, and (3) efficient gradient representation through rank-1 decomposition. Based on these principles, we introduce GradMetaNet, a novel architecture for learning on gradients, constructed from simple equivariant blocks. We prove universality results for GradMetaNet, and show that previous approaches cannot approximate natural gradient-based functions that GradMetaNet can. We then demonstrate GradMetaNet's effectiveness on a diverse set of gradient-based tasks on MLPs and transformers, such as learned optimization, INR editing, and estimating loss landscape curvature.

## 1 Introduction

Gradients of neural networks are fundamental objects in deep learning, driving optimization and offering insights into model behavior. Beyond gradient descent and its variants [9, 40, 71], gradients are used in diverse applications that call for sophisticated processing. These applications broadly span three areas: model optimization, editing, and analysis. In accelerated **optimization**, several approaches use (multi-)gradient information to improve convergence speed. These approaches range from classical curvature-aware methods like natural gradient descent [3] powered by efficient approximate curvature-based preconditioners [25, 27, 29, 49, 55, 86], to learned optimizers [6, 8, 56]. In model **editing**, gradient information guides pruning algorithms for weight compression [32, 45, 83, 88], and enables targeted behavior modification in large language models [18, 59]. For model **analysis** and interpretability, gradient information is used to compute influence functions that trace the impact of individual training samples [28, 42], estimate model uncertainty [17, 36], and more.

While most approaches rely on predefined algorithms and heuristics, recent works explore *learnable* processing of gradients for downstream tasks [18, 41, 59, 92]. Learned methods offer two key

---

[*]Equal contribution

39th Conference on Neural Information Processing Systems (NeurIPS 2025).

advantages. First, they are essential when no predefined algorithm is known. In model editing, for instance, updating a model using gradients of the editing objective while maintaining performance on a validation set requires intricate gradient adaptations that are difficult to model analytically. Learned approaches can effectively discover these gradient adaptations through supervision [18, 59]. Second, learned approaches offer a powerful mechanism for approximating computationally expensive methods. Methods such as natural gradient descent require hand-crafted approximations for practical application. Learned approaches, if successful, can bridge this gap by discovering efficient approximations tailored for a specific distribution of models. Unfortunately, existing learned approaches use architectures not specifically designed for processing *gradient information*. For example, De Cao et al. [18], Mitchell et al. [59] don't account for the parameter symmetries in the gradient representation (as they process gradients of a single model), while recent weight-space methods [41, 92] use inefficient gradient representations and process only a single gradient. As a result, they are unable to capture curvature information that is critical to many tasks.

**Our approach.** In this paper, we introduce GradMetaNet, an architecture designed for learning on gradients of deep models such as *MLPs* and *transformers*. GradMetaNet's design is guided by the following principles: (1) **Respecting symmetries:** Neural parameter spaces exhibit inherent symmetries, leading to redundancies in gradient representations. GradMetaNet's design is derived to respect these symmetries, reducing the number of parameters and improving sample efficiency. As demonstrated in previous work [11, 12, 16, 22, 43, 47, 79, 91], equivariant design is crucial for learning on data with symmetries. (2) **Processing sets of gradients:** Many applications, such as curvature-aware optimization, pruning, and uncertainty estimation, require access to *collections* of gradients on different datapoints which encode the local geometry of the loss. GradMetaNet is thus designed to efficiently handle sets of gradients computed on different datapoints. (3) **Efficient representation:** As gradients are extremely high-dimensional, we encode them efficiently. Gradients of neural networks, evaluated on a single data point, admit a rank-1 decomposition which provides a compact representation that scales linearly (rather than quadratically) with the number of neurons.

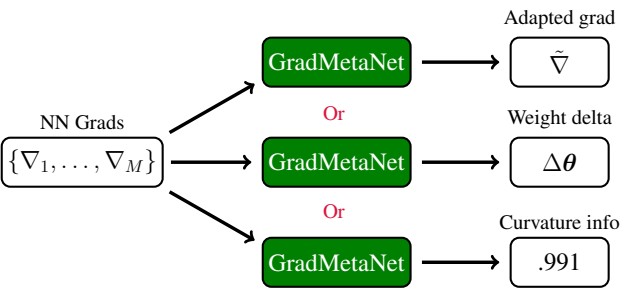

Figure 1: We propose GradMetaNet, a novel architecture that processes sets of gradients and can learn to compute gradient adaptations, parameter edits, or scalar values such as curvature information or influence functions.

Decomposed gradients have a simpler symmetry structure compared to the raw weight representation, allowing us to construct GradMetaNet using simple equivariant building blocks [31, 76, 91] and to incorporate attention mechanisms. This enables us to prove universality results still unknown for weight-space models. Additionally, we formally demonstrate the necessity of processing sets of gradients, proving that several fundamental gradient-based algorithms cannot be approximated based on a single averaged gradient.

We evaluate GradMetaNet on several gradient learning tasks, comparing to equivariant weight-space architectures and other natural baselines. First, we demonstrate GradMetaNet's ability to predict local curvature information using a small sample of gradients, achieving a 26.3% improvement over standard approximations, and outperforming other learned approaches. We then integrate GradMetaNet into learned optimizer architectures and apply it to train image classifiers and transformer language models, achieving up to a 4.63× reduction in steps compared to Adam, and a 1.78× improvement over other learned baselines. Finally, we use GradMetaNet for model editing, where we improve on current **state-of-the-art** results in editing MNIST and CIFAR10 INRs by up to 22.5%. Across all tasks, GradMetaNet consistently outperforms baselines, highlighting the value of efficient gradient representations and equivariant processing of sets of gradients.

## 2 Related Work

Several recent works have explored methods for learning over neural network weights [5, 20, 34, 35, 37, 41, 48, 61, 62, 67, 73–75, 82, 92, 93]. These methods often use equivariant architectures [7, 13, 15, 22, 24, 43, 47, 53, 68, 69, 72, 87, 91] that respect the internal symmetry of neural

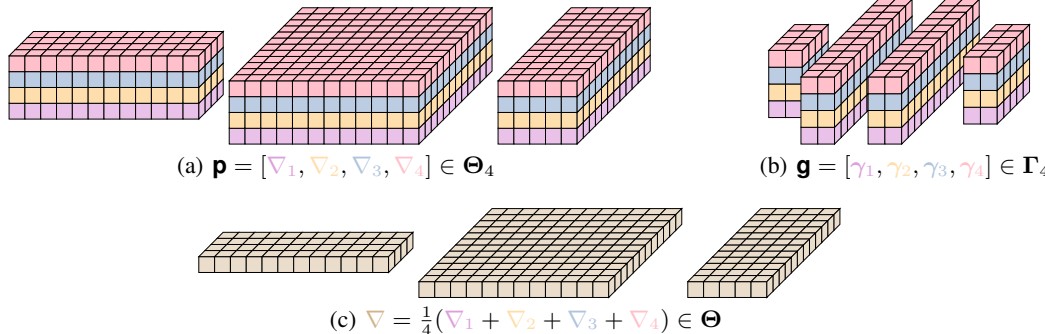

(a) $\mathbf{p} = [\nabla_1, \nabla_2, \nabla_3, \nabla_4] \in \boldsymbol{\Theta}_4$

(b) $\mathbf{g} = [\gamma_1, \gamma_2, \gamma_3, \gamma_4] \in \boldsymbol{\Gamma}_4$

(c) $\nabla = \frac{1}{4}(\nabla_1 + \nabla_2 + \nabla_3 + \nabla_4) \in \boldsymbol{\Theta}$

Figure 2: Gradient information on a batch of datapoints in different tensor representations. In 2(a), a stack of the weight-shaped gradients, one for each datapoint. In 2(b), a stack of the rank-1 gradient decompositions. In 2(c), the gradient of the average loss on the batch. All of these tensors are naturally computed when backpropagating the loss on the batch. GradMetaNet process tensors $\mathbf{g} \in \boldsymbol{\Gamma}_b$.

network weight spaces. A particularly promising application is processing gradients in weight space for tasks such as learned optimization [41, 92]. While these approaches respect the natural symmetries of gradients, they typically operate on a single gradient, missing valuable information encoded in gradient statistics. Furthermore, these methods process high-dimensional, full-size gradients limiting scalability. Other works, such as De Cao et al. [18], Mitchell et al. [59], analyze gradients of a *fixed* pre-trained model, and are not suitable for settings involving different models or evolving parameter configurations (e.g., learned optimization), as they are not equivariant to permutation symmetries.

Among classical, non-learned approaches, methods such as K-FAC and its variants [25, 27, 55] offer efficient ways to extract curvature information from gradients. These methods, widely used for curvature-aware optimization [25, 27, 29, 49, 55, 86], pruning [83, 88], uncertainty estimation [17, 36], and influence function estimation [28], need to make probabilistic assumptions on the distribution of gradients for computational feasibility. We advance this perspective by introducing a *learnable* approach to modeling these gradient distributions using GradMetaNet, offering greater flexibility and expressiveness.

## 3 Background

**Notation.** Throughout the paper, we denote models by $\boldsymbol{f_\theta} : \mathcal{X} \to \mathcal{Y}$, where $\boldsymbol{\theta} \in \boldsymbol{\Theta}$ are the parameters. When $\boldsymbol{f_\theta}$ is a multi-layer perceptron (MLP), we write the input dimension as $d_0$, the output dimension as $d_L$, the hidden dimensions as $d_1, \ldots, d_{L-1}$, and denote the activation function by $\sigma$. In this case, the parameters are given by $\boldsymbol{\theta} = (\boldsymbol{W}_1, \boldsymbol{b}_1, \ldots, \boldsymbol{W}_L, \boldsymbol{b}_L)$. Given a dataset $\mathcal{D} \subseteq \mathcal{X} \times \mathcal{Y}$, a loss function $\ell : \mathcal{Y} \times \mathcal{Y} \to \mathbb{R}$, and a batch $\mathcal{B} \subseteq \mathcal{D}$, we denote the loss on the batch by

$$\mathcal{L}_\mathcal{B}(\boldsymbol{\theta}) := \frac{1}{|\mathcal{B}|} \sum_{(\boldsymbol{x}, \boldsymbol{y}) \in \mathcal{B}} \ell(\boldsymbol{f_\theta}(\boldsymbol{x}), \boldsymbol{y}).$$

The parameter gradients of the loss on the batch are denoted by $\nabla_\mathcal{B} := \nabla_{\boldsymbol{\theta}} \mathcal{L}_\mathcal{B}(\boldsymbol{\theta})$. For a single data point $(\boldsymbol{x}, \boldsymbol{y})$, we write $\mathcal{L}_{(\boldsymbol{x}, \boldsymbol{y})}(\boldsymbol{\theta}) := \ell(\boldsymbol{f_\theta}(\boldsymbol{x}), \boldsymbol{y})$ and $\nabla_{(\boldsymbol{x}, \boldsymbol{y})} := \nabla_{\boldsymbol{\theta}} \mathcal{L}_{(\boldsymbol{x}, \boldsymbol{y})}(\boldsymbol{\theta})$.

**Rank-1 decomposition of gradients.** While general parameter gradients have the same shape as parameters, for many neural architectures, the gradient $\nabla_{(\boldsymbol{x}, \boldsymbol{y})}$ admits a rank-1 decomposition through the computation graph of $\boldsymbol{f_\theta}$. For an MLP $\boldsymbol{f_\theta}$, let $\boldsymbol{a}^{(l)}$ and $\boldsymbol{u}^{(l)}$ denote $\boldsymbol{x}$'s activation and pre-activation vectors at layer $l$. The backpropagated signal (pre-activation gradient) at layer $l$ is

$$\boldsymbol{g}^{(l)} := \frac{\partial \mathcal{L}_{(\boldsymbol{x}, \boldsymbol{y})}(\boldsymbol{\theta})}{\partial \boldsymbol{u}^{(l)}} = \frac{\partial \ell(\boldsymbol{f_\theta}(\boldsymbol{x}), \boldsymbol{y})}{\partial \boldsymbol{u}^{(l)}}. \tag{1}$$

Applying the chain rule yields the following expressions for the weight and bias gradients:

$$\nabla_{\boldsymbol{W}_l} \mathcal{L}_{(\boldsymbol{x}, \boldsymbol{y})}(\boldsymbol{\theta}) = \boldsymbol{g}^{(l)}(\boldsymbol{a}^{(l-1)})^\top, \quad \nabla_{\boldsymbol{b}_l} \mathcal{L}_{(\boldsymbol{x}, \boldsymbol{y})}(\boldsymbol{\theta}) = \boldsymbol{g}^{(l)}. \tag{2}$$

See full derivation in Appendix A.1. This decomposition allows us to represent $\nabla_{(\boldsymbol{x}, \boldsymbol{y})}$ using the tuple $(\boldsymbol{\gamma}^{(0)}, \ldots, \boldsymbol{\gamma}^{(L)})$, where $\boldsymbol{\gamma}^{(l)} := (\boldsymbol{a}^{(l)}, \boldsymbol{g}^{(l)})$. Note that $\boldsymbol{a}^{(l)}$ and $\boldsymbol{g}^{(l)}$ are naturally computed during

backpropagation, so they can be extracted *without additional cost*, e.g., using hooks in standard frameworks like PyTorch [66]. See code example in Appendix A.2.

**Decomposition for transformer gradients.** Similar gradient decompositions exist for many other neural architectures [21, 27]. In Appendix B.1 we derive such a decomposition for *all components of the transformer*. To illustrate the structure of the decomposition, we focus here on the feedforward (MLP) layers, which account for the majority of parameters. Given an input sequence $\boldsymbol{s} = (\boldsymbol{x}_1, \ldots, \boldsymbol{x}_T)$, let $\boldsymbol{a}_t^{(l)}$ denote the activation of token $\boldsymbol{x}_t$ at layer $l$ of the MLP component in a transformer block, and let $\boldsymbol{g}_t^{(l)}$ be the corresponding pre-activation gradient signal, computed with respect to the loss on the *entire sequence* $\mathcal{L}_{\boldsymbol{s}}(\boldsymbol{\theta})$. We similarly get:

$$\nabla_{\boldsymbol{W}_l}\mathcal{L}_{\boldsymbol{s}}(\boldsymbol{\theta}) = \sum_{t=1}^{T} \boldsymbol{g}_t^{(l)}(\boldsymbol{a}_t^{(l-1)})^{\top}, \quad \nabla_{\boldsymbol{b}_l}\mathcal{L}_{\boldsymbol{s}}(\boldsymbol{\theta}) = \sum_{t=1}^{T} \boldsymbol{g}_t^{(l)}. \tag{3}$$

We can therefore represent the gradient using $\boldsymbol{\gamma}_1^{(l)}, \ldots, \boldsymbol{\gamma}_T^{(l)}$ where $\boldsymbol{\gamma}_t^{(l)} := (\boldsymbol{a}_t^{(l)}, \boldsymbol{g}_t^{(l)})$. In other words, while we incur an additional sequence dimension, the rank-1 decomposition still holds per-token.

**Approximate curvature from gradient statistics.** Gradients statistics across datapoints encode information about the local geometry of the loss landscape. For example, the Fisher information matrix (FIM)

$$\boldsymbol{F}_{\boldsymbol{\theta}} = \mathbb{E}_{\substack{\boldsymbol{x}\sim\mathcal{D},\\ \boldsymbol{y}\sim p_{\boldsymbol{\theta}}(\boldsymbol{y}|\boldsymbol{x})}} \left( \nabla_{\boldsymbol{\theta}}\log(p_{\boldsymbol{\theta}}(\boldsymbol{y}|\boldsymbol{x}))\nabla_{\boldsymbol{\theta}}\log(p_{\boldsymbol{\theta}}(\boldsymbol{y}|\boldsymbol{x}))^{\top} \right)$$

is a second-order approximation of the change in the model's predictive distribution $p_{\boldsymbol{\theta}}(\boldsymbol{y}|\boldsymbol{x})$ with respect to a change in the parameters, and when $\boldsymbol{\theta}$ is a local minimum, the FIM is identical to the Hessian. Gradient decompositions similar to the one discussed above have been used to derive tractable approximation of the FIM [25, 27, 29, 49, 55, 86]. In this work we directly process sets of gradients to enable learning the local geometry

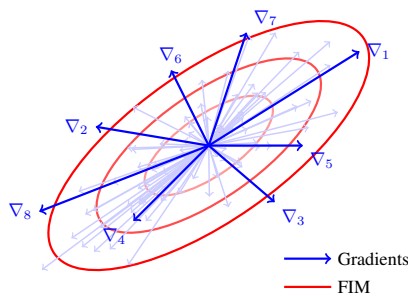

Figure 3: Fisher information as a second-order approximation to the loss.

of the loss landscape from their statistics. For background on the Fisher Information Matrix, see Appendix A.3, and for a detailed overview, refer to Martens [54], Pascanu [65].

## 4 Symmetries of Decomposed Gradients

**Weight-space symmetries.** Many neural architectures exhibit parameter space symmetries: parameter transformations that leave the network's function unchanged. In particular, MLPs exhibit well-documented permutation invariance [1, 33, 61, 77, 93]: permuting the neurons of a hidden layer, while keeping track of the connections to the neighboring layers, alters the weight matrices but preserves the function represented by the network. This parameter space symmetry can be expressed by an action of the permutation symmetry group $G := S_{d_1} \times \cdots \times S_{d_{L-1}}$. For $\boldsymbol{\theta} = (\boldsymbol{W}_1, \boldsymbol{b}_1, \ldots, \boldsymbol{W}_L, \boldsymbol{b}_L) \in \boldsymbol{\Theta}$ and $h = (\tau_1, \ldots, \tau_{L-1}) \in G$, the action $h \cdot \boldsymbol{\theta} = (\boldsymbol{W}_1', \boldsymbol{b}_1', \ldots, \boldsymbol{W}_L', \boldsymbol{b}_L')$ is given by

$$\begin{aligned}
\boldsymbol{W}_1' &= \boldsymbol{P}_{\tau_1}^{\top}\boldsymbol{W}_1, & \boldsymbol{b}_1' &= \boldsymbol{P}_{\tau_1}^{\top}\boldsymbol{b}_1, \\
\boldsymbol{W}_l' &= \boldsymbol{P}_{\tau_l}^{\top}\boldsymbol{W}_l\boldsymbol{P}_{\tau_{l-1}}, & \boldsymbol{b}_l' &= \boldsymbol{P}_{\tau_l}^{\top}\boldsymbol{b}_l, \quad l \in \{2, \ldots, L-1\} \\
\boldsymbol{W}_L' &= \boldsymbol{W}_L\boldsymbol{P}_{\tau_{L-1}}, & \boldsymbol{b}_L' &= \boldsymbol{b}_L,
\end{aligned}$$

where $\boldsymbol{P}_{\tau_l} \in \{0, 1\}^{d_l \times d_l}$ is the permutation matrix corresponding to $\tau_l \in S_{d_l}$ (see Figure 4 for an illustration). The action of $G$ preserves the function represented by the network: $\boldsymbol{f}_{g\cdot\boldsymbol{\theta}} \equiv \boldsymbol{f}_{\boldsymbol{\theta}}$. Permutation symmetries naturally extend to many other neural architectures, see Kofinas et al. [41], Lim et al. [48], Zhou et al. [92] for a detailed discussion.

**Decomposed gradient symmetries.** Weight-space symmetries naturally extend to decomposed gradients. Following the discussion in Section 3, we define the decomposed gradient space of an MLP $\boldsymbol{f}_{\boldsymbol{\theta}}$ as

$$\boldsymbol{\Gamma} := \boldsymbol{\Gamma}^{(0)} \oplus \cdots \oplus \boldsymbol{\Gamma}^{(L)}, \tag{4}$$

where $\boldsymbol{\Gamma}^{(l)} := \mathbb{R}^{d_l \times 2}$, referred to as the *neuron space* of the $l$-th layer, contains pairs $\boldsymbol{\gamma}^{(l)} = (\boldsymbol{a}^{(l)}, \boldsymbol{g}^{(l)})$ of activations and pre-activation gradients. $\oplus$ denotes a direct sum of vector spaces.

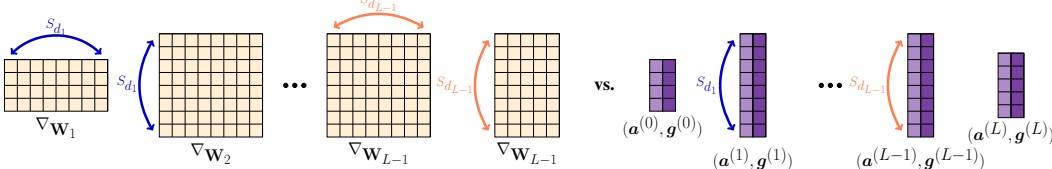

Figure 4: The action of $G = S_{d_1} \times \cdots \times S_{d_{L-1}}$ on parameter space performs simultaneous permutation of rows and columns of consecutive weight matrices. In contrast, $G$'s action on the decomposed gradient space permutes the neuron space of each hidden layer independently.

$G$'s action extends naturally to $\Gamma$. For a decomposed gradient $\boldsymbol{\gamma} = (\boldsymbol{\gamma}^{(0)}, \ldots, \boldsymbol{\gamma}^{(L)}) \in \Gamma$ and $h = (\tau_1, \ldots, \tau_{L-1}) \in G$, the action $h \cdot \boldsymbol{\gamma} = (h \cdot \boldsymbol{\gamma}^{(0)}, \ldots, h \cdot \boldsymbol{\gamma}^{(L)})$ is given by:

$$h \cdot \boldsymbol{\gamma}^{(l)} = (\boldsymbol{P}_{\tau_l}^\top \boldsymbol{a}^{(l)}, \boldsymbol{P}_{\tau_l}^\top \boldsymbol{g}^{(l)}), \text{ for } l = 1, \ldots, L-1, \quad h \cdot \boldsymbol{\gamma}^{(l)} = \boldsymbol{\gamma}^{(l)}, \text{ for } l = 1, L. \quad (5)$$

Let $\boldsymbol{\Phi}_{(\boldsymbol{x},\boldsymbol{y})} : \boldsymbol{\Theta} \to \boldsymbol{\Gamma}$ be the function that maps parameters $\boldsymbol{\theta}$ to the decomposition of the gradient $\nabla_{(\boldsymbol{x},\boldsymbol{y})} = \nabla_{\boldsymbol{\theta}} \mathcal{L}_{(\boldsymbol{x},\boldsymbol{y})}(\boldsymbol{\theta})$. $\boldsymbol{\Phi}_{(\boldsymbol{x},\boldsymbol{y})}$ is $G$-*equivariant*, that is:

$$\boldsymbol{\Phi}_{(\boldsymbol{x},\boldsymbol{y})}(h \cdot \boldsymbol{\theta}) = h \cdot \boldsymbol{\Phi}_{(\boldsymbol{x},\boldsymbol{y})}(\boldsymbol{\theta}). \quad (6)$$

This equivariance applies to any transformation that modifies or extracts information from the function represented by $\boldsymbol{f_\theta}$ using its gradients. As illustrated in Figure 4, $G$'s action on $\boldsymbol{\Gamma}$ is simpler than its action on $\boldsymbol{\Theta}$, since the permutations act independently on the different neuron spaces.

**Sets of decomposed gradients.** When computing the gradient of the loss over a batch $\mathcal{B} \subseteq \mathcal{D}$, we naturally obtain a set $\{\nabla_{(\boldsymbol{x},\boldsymbol{y})}\}_{(\boldsymbol{x},\boldsymbol{y}) \in \mathcal{B}}$ of individual gradients[2]. As discussed in Section 3, this set contains implicit information about the local geometry of the loss landscape, which is critical for many tasks. Therefore, when designing methods for learning on gradients, it's beneficial to process the entire collection rather than the gradient of the average loss. This intuition is formally motivated in Section 6.

As illustrated in Figure 2, gradients across a batch of size $b$ can be efficiently represented as a tensor $\mathbf{g} \in \boldsymbol{\Gamma}_b$, where the batched decomposed gradient space $\boldsymbol{\Gamma}_b$ is

$$\boldsymbol{\Gamma}_b := \boldsymbol{\Gamma}_b^{(0)} \oplus \cdots \oplus \boldsymbol{\Gamma}_b^{(L)}, \quad \boldsymbol{\Gamma}_b^{(l)} := \mathbb{R}^{b \times d_l \times 2} \quad (7)$$

See formal definitions of all parameter and gradient spaces in Appendix C. Since the order of gradients in the batch is arbitrary, the set symmetry group is extended to $G_b := S_b \times G$. The action of $(\tau, h) \in G_b$ permutes the batch indices using $\tau$ and independently applies $h$ across the neurons:

$$((\tau, h) \cdot \mathbf{g}^{(l)})_{j,:,:} = h \cdot \mathbf{g}^{(l)}_{\tau^{-1}(j),:,:}. \quad (8)$$

When modeling functions $\boldsymbol{\Phi} : \boldsymbol{\Gamma}_b \to \boldsymbol{\Theta}$, we want to respect decomposed gradient symmetries ($G$-equivariance) and be independent of gradient ordering ($S_b$-invariance). We thus aim to design models that satisfy $\boldsymbol{\Phi}((\tau, h) \cdot \mathbf{g}) = h \cdot \boldsymbol{\Phi}(\mathbf{g})$.

**Extension to transformers.** The analysis above extends naturally to decomposed transformer gradients. The sequence dimension is treated as a batch dimension (with optional added sequence PE), and the neuron spaces correspond to the input and output of every linear layer and every query/key/value/output projection. Additionally, the neuron spaces across the residual stream are tied together, having the same symmetry group. For a detailed discussion, see Appendix B.2.

**Feature spaces.** As with other equivariant architectures, it is useful to extend $\boldsymbol{\Gamma}$ and $\boldsymbol{\Gamma}_b$ to more general feature spaces $\boldsymbol{\Gamma}_b[f]$ and $\boldsymbol{\Gamma}[f]$ by assigning an $f$-dimensional feature vector to each entry. See Appendix C for a formal definition.

## 5  GradMetaNet

In this section, we introduce GradMetaNet, an architecture for learning on gradients designed to process **sets** of **decomposed** gradients in a $G_b$-**equivariant** way. As the symmetry structure of $\boldsymbol{\Gamma}_b$ is simpler than that of $\boldsymbol{\Theta}_b$, we can design GradMetaNet using simpler equivariant layers compared to its weight-space counterparts [62, 92, 93]. Specifically, $\mathbf{g} \in \boldsymbol{\Gamma}_b$ can be viewed as a set of decomposed gradients $\{\boldsymbol{\gamma}_1, \ldots, \boldsymbol{\gamma}_b\}$, each of which is a concatenation of sets of neuron-level features $\boldsymbol{\gamma}_i = (\boldsymbol{\gamma}_i^{(0)}, \ldots, \boldsymbol{\gamma}_i^{(L)})$. To further simplify the symmetry structure, we incorporate

---

[2]By "naturally" we mean that these gradients are always computed when backpropagating the average loss on the batch, and can be be extracted using hooks *without additional cost*.

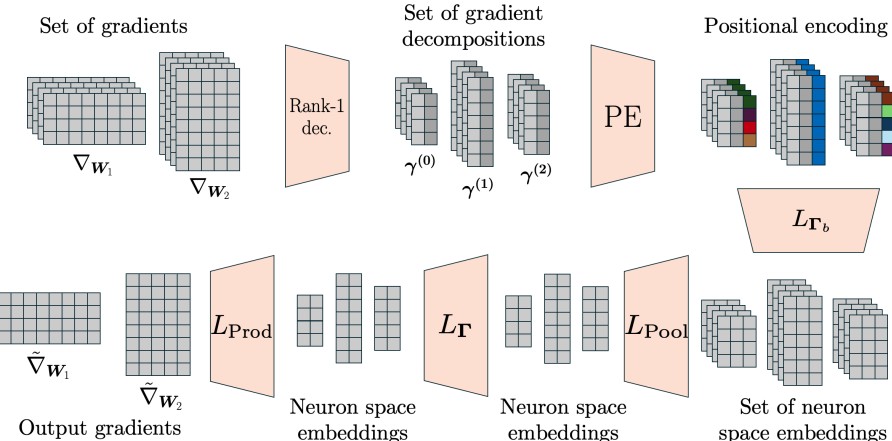

Figure 5: GradMetaNet pipeline: gradients are decomposed into rank-1 factors and positional encoding is applied. The input is then transformed by a stack of $L_{\Gamma_b}$ equivariant interactions-across-sets layers. $L_{\text{Pool}}$ pools these representations into $\Gamma[f]$, removing the batch dimension. Then a stack of $L_{\Gamma}$ layers updates this representation, and $L_{\text{Prod}}$ maps the result back to $\Theta$.

a positional encoding map that enables us to treat each $\gamma_i$ as a *single* bag of neuron-level features. This allows us to implement GradMetaNet using simple, well-established equivariant layers for sets [31, 68, 76, 91]. As illustrated in Figure 5, a GradMetaNet model $\Phi$ is composed of a positional encoding map followed by a stack of equivariant linear layers of several types, interleaved with pointwise non-linearities.

$$\Phi = L_{\text{Prod}} \circ \sigma \circ L_{\Gamma}^{(k_2)} \circ \cdots \circ \sigma \circ L_{\Gamma}^{(1)} \circ L_{\text{Pool}} \circ \sigma \circ L_{\Gamma_b}^{(k_1)} \circ \cdots \circ \sigma \circ L_{\Gamma_b}^{(1)} \circ \text{PE}. \quad (9)$$

The following is a description of each layer:

(I) Similarly to Lim et al. [48], Zhou et al. [93], we use a positional encoding map $\text{PE} : \Gamma_b \to \Gamma_b[f]$ that concatenates a *layer identifier* each neurons in the intermediate layers and a *neuron identifier* to each neuron in the first and last layers. We use sinusoidal PE [80].

(II) $L_{\Gamma_b} : \Gamma_b[f_{\text{in}}] \to \Gamma_b[f_{\text{out}}]$ are then parametrized as the interactions-across-sets layers introduced in Hartford et al. [31] (batch dimension × neuron dimension).

(III) The pooling layer $L_{\text{Pool}} : \Gamma_b[f_{\text{in}}] \to \Gamma[f_{\text{out}}]$ is designed to be $S_b$-invariant and $G$-equivariant, and is implemented as $L_{\text{Pool}}(\mathbf{g}^{(l)})_{j,:} = M_1 \sum_{i'=1}^{b} \mathbf{g}_{i',j,:}^{(l)} + M_2 \sum_{l'=0}^{L} \sum_{i'=1}^{b} \sum_{j'=1}^{d_{l'}} \mathbf{g}_{i',j',:}^{(l')}$, for learnable $M_1, M_2 \in \mathbb{R}^{f_{\text{out}} \times f_{\text{in}}}$.

(IV) $L_{\Gamma} : \Gamma[f_{\text{in}}] \to \Gamma[f_{\text{out}}]$ are parameterized as equivariant DeepSets layers [91].

(V) Finally, similarly to the generalized product layer in Navon et al. [62], $L_{\text{Prod}} : \Gamma[f_{\text{in}}] \to \Theta$ applies a pointwise MLP to the features associated with the neurons connected to each weight: $\mathbf{W}_{i,j}^{(l)} = \text{MLP}_1([\mathbf{g}_{i,:}^{(l)}, \mathbf{g}_{j,:}^{(l+1)}]), \mathbf{b}_i^{(l)} = \text{MLP}_2(\mathbf{g}_{i,:}^{(l)})$.

For detailed descriptions and implementation details for all layers, a $G$-invariant head for invariant tasks, and other design choices, see Appendix D.1.

**Extension to transformers.** As formally detailed and motivated in Appendix B, decomposed transformer gradients have an additional sequence dimension with a set symmetry structure, and the neuron spaces across the residual stream are tied together. Therefore, when applying GradMetaNet, we treat the sequence dimension as the batch dimension (with optional sequence PE), stack all the neuron features across the residual stream together to a single neuron space, and extend our positional encoding to include the attention head indices. For an extended discussion see Appendix B.2 and B.3.

**GradMetaNet++.** Similarly to Kasten et al. [39], Lee et al. [46], Romero et al. [70], we can preserve equivariance by replacing summation in steps (II) and (VI) with attention mechanisms. Therefore, we introduce an attention-based variant of GradMetaNet, termed GradMetaNet++, where $L_{\Gamma_b}$ and $L_{\Gamma}$ are implemented using attention across the neuron and batch dimensions. This variant demonstrates significant performance improvements on some tasks, consistent with findings in previous studies. For a detailed description of GradMetaNet++ refer to Appendix D.2.

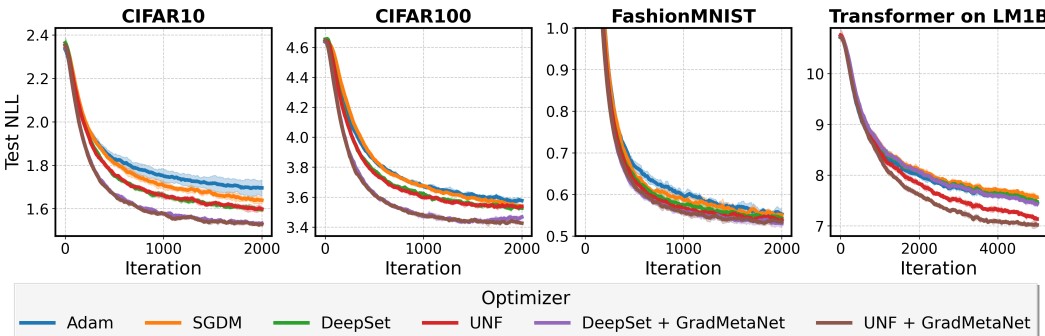

Figure 6: Test loss curves for MLP image classification tasks and a transformer language model trained on LM1B, using different optimizers and (learning rate tuned) Adam. Curves are smoothed and averaged over 5 random initializations, with shaded regions representing standard deviation.

## 6 Theoretical Analysis

**Importance of processing sets of gradients.** Instead of processing the gradient of the average loss as in other gradient learning methods [18, 41, 92], GradMetaNet processes sets of (decomposed) gradients computed on individual datapoints. This approach is motivated by the fact that a set of gradients encodes strictly more information than the corresponding average gradient, enabling e.g. curvature estimations that cannot be computed using the average alone. This intuition is formalized in Appendix E.1, leading to Proposition E.6 whose informal statement appears below.

**Proposition 6.1.** *Let $\{\nabla_{(\boldsymbol{x},\boldsymbol{y})}\}_{(\boldsymbol{x},\boldsymbol{y})\in\mathcal{B}}$ be gradients computed on on a set of datapoints $\mathcal{B} \subseteq \mathcal{D}$. There exist functions–such as natural gradient approximations or pruning saliency scores–that cannot be reconstructed from the average gradient $\nabla_{\mathcal{B}}$ alone.*

**Expressive power.** Restricting a model to be equivariant with respect to a specific group action can potentially reduce its expressive power [51, 52, 60]. However, we demonstrate that GradMetaNet does not suffer from such limitations. Specifically, we prove a universal approximation property for equivariant functions defined on a compact domain that doesn't intersect a certain low-dimensional subset $\mathcal{E} \subset \boldsymbol{\Gamma}_b$. Formally:

**Theorem 6.2.** *Let $\mathcal{K} \subset \boldsymbol{\Gamma}_b$ be a compact domain such that $\mathcal{K} = \cup_{g \in G_b} g \cdot \mathcal{K}$ and $\mathcal{K} \cap \mathcal{E} = \emptyset$. GradMetaNet models are universal approximators (in the $\|\cdot\|_\infty$-sense) of continuous $G_b$-equivariant functions from $\mathcal{K}$ to $\boldsymbol{\Theta}$.*

$\mathcal{E}$ is the set of neuron features that have identical activations and backpropagated signals for at least two neurons (see Appendix E.2 for a precise definition). Similarly to Finkelshtein et al. [23], Maron et al. [53], the inclusion of $\mathcal{E}$ in Theorem 6.2 is essential (see Appendix E.3). However, $\mathcal{E}$ is a union of subspaces of co-dimension 2, has Lebesgue measure 0, and the conditions for membership in $\mathcal{E}$ are highly unlikely in practice, making this assumption mild.

**Corollary 6.3.** *Let $\mathcal{B}$ and $\{\nabla_{(\boldsymbol{x},\boldsymbol{y})}\}_{(\boldsymbol{x},\boldsymbol{y})\in\mathcal{B}}$ be as in Proposition 6.1. Several natural functions–such as natural gradient approximations and pruning saliency scores–can be effectively approximated by GradMetaNet, which has access to $\{\nabla_{(\boldsymbol{x},\boldsymbol{y})}\}_{(\boldsymbol{x},\boldsymbol{y})\in\mathcal{B}}$, but cannot be approximated by methods that rely solely on $\nabla_{\mathcal{B}}$.*

For formal statements and proofs of Proposition 6.1, Theorem 6.2, and Corollary 6.3, see Appendix E. In summary, GradMetaNet incorporates meaningful inductive biases for processing sets of gradients while retaining high expressive power, enabling it to represent all continuous functions on sets of gradients under mild assumptions.

## 7 Experiments

In this section, we evaluate GradMetaNet on a variety of learning tasks on gradients. We empirically demonstrate the importance of each of our design principles by ablating components and comparing to other baselines. We then showcase GradMetaNet's effectiveness for three applications: curvature information estimation, learned optimization, and INR editing. We include full experimental details in Appendix F and additional experimental results in Appendix G.

## 7.1 Curvature Information Estimation

To demonstrate GradMetaNet's ability to learn to approximate loss landscape curvature, we train it to predict the diagonal of the Fisher Information Matrix (FIM) from small samples of gradients. The diagonal of the FIM encodes the curvature along individual parameter directions, capturing the network's sensitivity to a small change in each parameter.

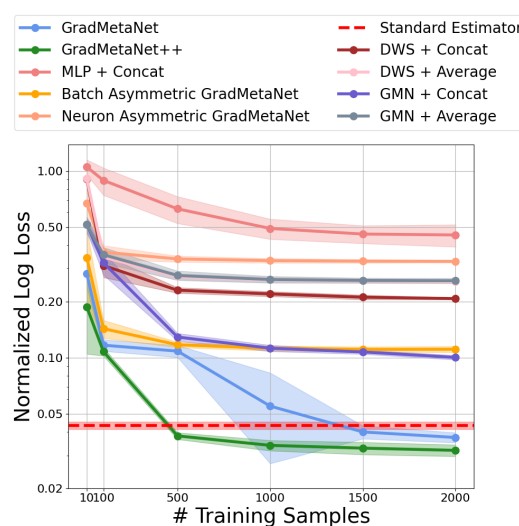

Figure 7: Comparison of gradient-learning models trained to predict the FIM diagonal from a sample of 128 gradients. Results are averaged over 5 seeds; shading represents standard deviation.

**Data.** We first create a set of randomly initialized MLPs with 1-dimensional input and output. We then generate the targets by computing the FIM diagonal for each model over a sample of 1024 inputs in $[-1, 1]$. The input to each baseline is a smaller gradient sample computed over 128 points sampled from $[-1, 1]$.

**Baselines.** We compare GradMetaNet and GradMetaNet++ against a range of baselines, including architectures that (1) rely solely on the average gradient: DWS+Average [61], GMN+Average [41], (2) use full gradients instead of the rank-1 decomposition: DWS+Concat, GMN+Concat, or (3) (partially) disregard symmetries: Batch Asymmetric GradMetaNet, Neuron Asymmetric GradMetaNet, and MLP+Concat. See Appendix F.1 for full descriptions of the baselines.

**Results and discussion.** To measure the sample efficiency of each baseline, we repeat the experiment with a varying number of training examples (each training sample is still a set of 128 gradients, but we vary the *number of such sets* the models see during training). As seen in Figure 7, GradMetaNet and GradMetaNet++ perform significantly better than baselines across varying training set sizes. We also compare to a non-learnable approximation that directly estimates the diagonal of the FIM using the 128 input gradients (rather than the full set of 1024 datapoints). Only GradMetaNet and GradMetaNet++ outperform this baseline, achieving an improvement of 13.7% and 26.3% respectively. These results demonstrate GradMetaNet's potential to learn more accurate approximations of gradient-based algorithms. In Appendix G we discuss a scaled-up version of this experiment for models with over 1M parameters.

## 7.2 Learned Optimizers

Optimizing deep neural networks is a fundamental challenge in deep learning. While classical optimizers [19, 40, 63, 81] have become standard tools, their design largely relies on intuition and empirical validation. A promising alternative is to *learn* the optimization algorithm itself through meta-training [6, 8, 56]. Learned optimizers can potentially discover more effective update rules by adapting to the statistical patterns in loss landscapes. Most optimizers (learnable and hand-crafted) only process the averaged batch gradient, and therefore don't have access to local curvature information. In this experiment, we integrate GradMetaNet into learned optimizer architectures, providing it with the raw information required for computing the curvature in the form of sets of gradients on individual datapoints across batches.

**Setup.** Following Harrison et al. [30], Zhou et al. [92], we parametrize our learned optimizer rules as

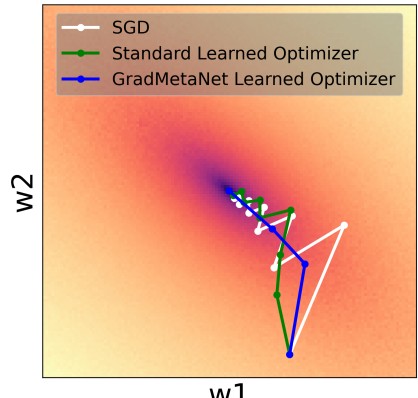

Figure 8: GradMetaNet-based learned optimizers can account for loss-landscape curvature, avoiding redundant steps.

$$\boldsymbol{\theta}_{t+1} \leftarrow \boldsymbol{\theta}_t + \alpha \left( \boldsymbol{v}_t^\mu + \beta \boldsymbol{F}_\phi(\boldsymbol{\theta}_t, \nabla_t, \{\boldsymbol{v}_t^{\mu_i}\}_{i=1}^k, t) \right), \tag{10}$$

where $\nabla_t$ is the current gradient, and $\{v_t^{\mu_i}\}_{i=1}^{k}$ are momentum terms with different decay rates. The standard architecture for $F_\phi$ is DeepSets (DS) [91], which applies a per-parameter MLP to the input features. More recently, researchers have explored using equivariant weight-space architectures like Universal Neural Functionals (UNF) [92] to implement $F_\phi$. For GradMetaNet-based learned optimizers we parametrize $F_\phi$ as

$$F_\phi\left(\boldsymbol{\theta}_t, \nabla_t, \{v_t^{\mu_i}\}_{i=1}^{k}, t, F_\psi^{\text{GradMetaNet}}\left(\mathbf{g}_t, \{\mathbf{v}_t^{\mu_i}\}_{i=1}^{k}\right)\right), \tag{11}$$

Table 1: Multiplicative improvement in the number of steps to reach Adam's best test loss. For each task, we run Adam, record its best test loss $L$ and the number of steps $N$ required to reach it. We then run standard and GradMetaNet-based learned optimizers, measure their steps to reach $L$, and report the improvement relative to $N$. Results are averaged over 5 runs. Full Results in Table 6.

| Dataset | Optimizer | Avg. step reduction factor vs. Adam (↑) |
|---|---|---|
| F-MNIST | SGDM | 1.13x |
| | DS | 1.27x |
| | UNF | 1.16x |
| | DS + GradMetaNet | 1.44x |
| | UNF + GradMetaNet | **1.51x** |
| CIFAR10 | SGDM | 1.41x |
| | DS | 2.32x |
| | UNF | 2.64x |
| | DS + GradMetaNet | **4.63x** |
| | UNF + GradMetaNet | 4.26x |
| CIFAR100 | SGDM | 1.06x |
| | DS | 1.79x |
| | UNF | 1.58x |
| | DS + GradMetaNet | **3.15x** |
| | UNF + GradMetaNet | 2.85x |
| LM1B | SGDM | 1.01x |
| | DS | 0.88x |
| | UNF | 1.48x |
| | DS + GradMetaNet | 1.09x |
| | UNF + GradMetaNet | **1.82x** |

where $\mathbf{g}_t \in \Gamma_b$ is the set of decomposed gradients across the current batch, and $\{\mathbf{v}_t^{\mu_i}\}_i$ are exponential moving averages of past $\mathbf{g}_t$s with different decay rates. The learnable meta-parameters $\alpha$, $\beta$, $\mu$, $\phi$, and $\psi$ are optimized during meta-training (using PES [85]) to minimize the training loss after $T$ steps. We evaluate five types of architectures: DeepSets, UNF, DeepSets + GradMetaNet, UNF + GradMetaNet, and learnable SGD with momentum (taking $\beta = 0$ in Equation 10).

**Tasks.** We use three types of optimization tasks: (1) optimizing a 2-parameter linear regression, constructed to have non-diagonal curvature, (2) optimizing MLPs for classifying CIFAR10, CIFAR100 [44], and FashionMNIST [90] images, and (3) optimizing transformer-based language models on LM1B [14]. For a detailed description of each task and the meta-training setup, see Appendix F.2.

**Results and discussion.** As Figure 8 demonstrates, for the 2-parameter regressions task, GradMetaNet-based learned optimizers can use the curvature of the loss landscape to avoid redundant steps. Consequently, as seen in Table 1 and Figure 6, GradMetaNet-based learned optimizers consistently outperform baselines across optimization tasks for both MLP and transformers, demonstrating the value of processing sets of gradients. In Appendix G we include additional experiments showing that GradMetaNet-based learned optimizers can generalize across tasks and architectures and show promise in scaling to larger-scale optimization.

## 7.3 INR Editing

Implicit Neural Representations (INRs) [64, 78] use neural networks to encode images as functions. In this experiment, we explore the task of INR editing, where the goal is to adapt the weights of an INR to modify the image it represents. This involves directly adjusting the INR's parameters by learning a metanetwork predicting a parameter delta $\Delta\boldsymbol{\theta}$, and updating the model with $\boldsymbol{\theta}' = \boldsymbol{\theta} + \Delta\boldsymbol{\theta}$.

**Data.** Following previous works [38, 93, 94], we use two standard benchmarks: figure dilation for MNIST INRs and contrast enhancement for CIFAR-10 INRs. For each INR, we compute the parameter gradients with respect to the MSE loss between the INR output and the target edited image, evaluated over randomly sampled points. The input data consists of both the parameters of the INR and the corresponding gradients.

**Baselines.** We evaluate GradMetaNet and GradMetaNet++ in combination with several weight-space architectures. GradMetaNet and GradMetaNet++ process the gradients to produce outputs in $\Theta$, which are then used as additional weight features for the base weight-space network. This hybrid approach is compared against two baselines: (1) the base weight-space network, and (2) the base weight-space network augmented with probing features. Following Kofinas et al. [41], the probing features are activations evaluated at randomly sampled grid points, incorporated as additional bias features.

Table 2: Results for the INR editing tasks on MNIST (dilation) and CIFAR10 (contrast). We report the MSE ($\downarrow$) in $10^{-2}$ for MNIST and $10^{-3}$ for CIFAR10, averaged over 3 seeds.

| | MNIST | | | CIFAR10 | | |
|---|---|---|---|---|---|---|
| | DWS [61] | GMN [41] | ScaleGMN [38] | DWS [61] | GMN [41] | ScaleGMN [38] |
| WS | $2.29 \pm 0.01$ | $1.96 \pm 0.02$ | $1.99 \pm 0.02$ | $5.57 \pm 0.02$ | $5.09 \pm 0.05$ | $5.23 \pm 0.13$ |
| WS + Probing | $2.36 \pm 0.06$ | $1.85 \pm 0.00$ | $1.92 \pm 0.04$ | $4.22 \pm 0.08$ | $3.81 \pm 0.02$ | $3.87 \pm 0.05$ |
| WS + GradMetaNet | $2.28 \pm 0.02$ | $\mathbf{1.70 \pm 0.01}$ | $1.70 \pm 0.00$ | $4.10 \pm 0.10$ | $3.65 \pm 0.01$ | $3.69 \pm 0.09$ |
| WS + GradMetaNet++ | $\mathbf{1.95 \pm 0.01}$ | $1.71 \pm 0.00$ | $\mathbf{1.60 \pm 0.00}$ | $\mathbf{3.86 \pm 0.02}$ | $\mathbf{2.99 \pm 0.03}$ | $\mathbf{3.00 \pm 0.00}$ |

**Results and Discussion.** As seen in Table 2, both GradMetaNet and GradMetaNet++ consistently improve the performance of weight-space models, achieve greater performance gains than probing, and improve the current state-of-the-art for weight-space model editing.

## 8 Conclusion and Limitations

**Conclusion.** We introduce GradMetaNet, an equivariant architecture for processing sets of decomposed gradients, supported by both theoretical and experimental results. Theoretically, we demonstrate that under mild assumptions GradMetaNet can approximate any function processing sets of gradients, and that average gradient methods are unable to approximate several natural gradient-based functions. Experimentally, we demonstrate GradMetaNet's ability to predict local curvature, enhance learned optimizers, and achieve state-of-the-art performance in model editing.

**Limitations and future work.** The current implementation of GradMetaNet is limited to MLPs and transformers; in future work, we hope to extend GradMetaNet to support other neural architectures. Additionally, GradMetaNet++'s use of attention mechanisms limits its usage in large-scale settings. Finally, scaling GradMetaNet, GradMetaNet++, or their variants to larger models, such as state-of-the-art LLMs and generative models, is an interesting direction for future work.

## Acknowledgments

YG is supported by the UKRI Engineering and Physical Sciences Research Council (EPSRC) CDT in Autonomous and Intelligent Machines and Systems (grant reference EP/S024050/1). MB is supported by EPSRC Turing AI World-Leading Research Fellowship No. EP/X040062/1 and EPSRC AI Hub on Mathematical Foundations of Intelligence: An "Erlangen Programme" for AI No. EP/Y028872/1. HM is a Robert J. Shillman Fellow and is supported by the Israel Science Foundation through a personal grant (ISF 264/23) and an equipment grant (ISF 532/23).

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

## A Extended Background

### A.1 Gradient Decomposition for MLPs

Using the notation introduced in Section 3, for $l = 1, \ldots L$, $n = 1, \ldots, d_{l-1}$, $m = 1, \ldots, d_l$ we apply the chain rule to get

$$
\left( \nabla_{\boldsymbol{W}_l} \mathcal{L}_{(\boldsymbol{x}, \boldsymbol{y})}(\boldsymbol{\theta}) \right)_{n,m} = \sum_{k=1}^{d_l} \left( \frac{\partial \mathcal{L}_{(\boldsymbol{x}, \boldsymbol{y})}(\boldsymbol{\theta})}{\partial (\boldsymbol{u}^{(l)})_k} \right) \cdot \left( \frac{\partial (\boldsymbol{u}^{(l)})_k}{\partial (\boldsymbol{W}_l)_{n,m}} \right) = \sum_{k=1}^{d_l} (\boldsymbol{g}^{(l)})_k (\delta_{n,k} (\boldsymbol{a}^{(l-1)})_n)
$$
$$
= (\boldsymbol{g}^{(l)})_m (\boldsymbol{a}^{(l-1)})_n,
$$

(12)

where $\delta_{n,k}$ is the Dirac delta defined by $\delta_{n,k} = \begin{cases} 1 & n = k \\ 0 & n \neq k. \end{cases}$. Similarly

$$\left( \nabla_{\boldsymbol{b}_l} \mathcal{L}_{(\boldsymbol{x},\boldsymbol{y})}(\boldsymbol{\theta}) \right)_m = \sum_{k=1}^{d_l} \left( \frac{\partial \mathcal{L}_{(\boldsymbol{x},\boldsymbol{y})}(\boldsymbol{\theta})}{\partial (\boldsymbol{u}^{(l)})_k} \right) \cdot \left( \frac{\partial (\boldsymbol{u}^{(l)})_k}{\partial (\boldsymbol{b}_l)_m} \right) = \sum_{k=1}^{d_l} (\boldsymbol{g}^{(l)})_k \delta_{m,k} = (\boldsymbol{g}^{(l)})_m. \quad (13)$$

Therefore,

$$\nabla_{\boldsymbol{W}_l} \mathcal{L}_{(\boldsymbol{x},\boldsymbol{y})}(\boldsymbol{\theta}) = \boldsymbol{g}^{(l)} (\boldsymbol{a}^{(l-1)})^\top, \quad \nabla_{\boldsymbol{b}_l} \mathcal{L}_{(\boldsymbol{x},\boldsymbol{y})}(\boldsymbol{\theta}) = \boldsymbol{g}^{(l)} \quad (14)$$

as discussed in Section 3.

## A.2 Extracting Activations and Pre-Activation Gradient Signals

As mentioned in Section 3, the activations ($\boldsymbol{a}^{(l)}$) and pre-activation gradient signals ($\boldsymbol{g}^{(l)}$) used for the gradient decomposition are naturally computed during backpropagation and don't need to be recomputed. The following is a PyTorch code example for extracting these components without additional cost using forward/backward hooks:

```python
import torch
import torch.nn as nn
import torch.nn.functional as F

class MLP(nn.Module):
    def __init__(self):
        super(MLP, self).__init__()
        self.fc1 = nn.Linear(8, 32)
        self.fc2 = nn.Linear(32, 16)
        self.fc3 = nn.Linear(16, 3)

    def forward(self, x):
        x = F.relu(self.fc1(x))
        x = F.relu(self.fc2(x))
        x = self.fc3(x)
        return x

activations = {}
tangents = {}

def forward_hook(module, inp, out):
    activations[module] = inp[0].detach()

def backward_hook(module, grad_inp, grad_out):
    tangents[module] = grad_out[0].detach()

model = MLP()

# Set hooks
model.fc1.register_forward_hook(forward_hook)
model.fc1.register_full_backward_hook(backward_hook)
model.fc2.register_forward_hook(forward_hook)
model.fc2.register_full_backward_hook(backward_hook)
model.fc3.register_forward_hook(forward_hook)
model.fc3.register_full_backward_hook(backward_hook)

# Backpropagate loss
x = torch.randn(4, 8) # (batch, input)
target = torch.randn(4, 3) # (batch, input)
output = model(x)
loss = F.mse_loss(output, target)
loss.backward()

print(activations)
print(tangents)
```

## A.3  The Fisher Information Matrix and Its Uses

Many gradient-based algorithms [17, 27, 55, 83, 88] use the Fisher Information Matrix (FIM) to approximate the curvature of the loss landscape. Below, we define the FIM and present two common gradient-based algorithms that utilize it. The FIM has numerous other applications, this section serves only as a basic introduction. For a more comprehensive overview, we refer readers to [54, 65].

**The Fisher information matrix.**  Consider a supervised learning problem of predicting outputs $\boldsymbol{y} \in \mathcal{Y}$ from inputs $\boldsymbol{x} \in \mathcal{X}$. We assume a probabilistic model of the form $p_{\boldsymbol{\theta}}(\boldsymbol{y}|\boldsymbol{x}) = p(\boldsymbol{y}|\boldsymbol{f}_{\boldsymbol{\theta}}(\boldsymbol{x}))$, where $p$ is called the *likelihood*. For classification tasks we may assume a softmax likelihood, $p(\boldsymbol{y} = k|\boldsymbol{f}_{\boldsymbol{\theta}}(\boldsymbol{x})) = \mathrm{softmax}(\boldsymbol{f}_{\boldsymbol{\theta}}(\boldsymbol{x}))_k$, and for regression, we usually take a Gaussian likelihood $p(\boldsymbol{y}|\boldsymbol{f}_{\boldsymbol{\theta}}(\boldsymbol{x})) = \mathcal{N}(\boldsymbol{y}; \boldsymbol{f}_{\boldsymbol{\theta}}(\boldsymbol{x}), \boldsymbol{I})$. $p_{\boldsymbol{\theta}}(\boldsymbol{y}|\boldsymbol{x})$ is called the *predictive distribution*. The FIM is defined by

$$\boldsymbol{F} = \mathbb{E}_{\boldsymbol{x}\sim\mathcal{D}, \boldsymbol{y}\sim p_{\boldsymbol{\theta}}(\boldsymbol{y}|\boldsymbol{x})} \left( \nabla_{\boldsymbol{\theta}} \log(p_{\boldsymbol{\theta}}(\boldsymbol{y}|\boldsymbol{x})) \nabla_{\boldsymbol{\theta}} \log(p_{\boldsymbol{\theta}}(\boldsymbol{y}|\boldsymbol{x}))^{\top} \right) \in \boldsymbol{\Theta} \otimes \boldsymbol{\Theta} \tag{15}$$

The FIM is a second order approximation of the change in the model's predictive distribution with respect to the parameters

$$\mathbb{E}_{\boldsymbol{x}\sim\mathcal{D}} \left( D_{\mathrm{KL}} \left( p_{\boldsymbol{\theta}}(\boldsymbol{y}|\boldsymbol{x}) \parallel p_{\boldsymbol{\theta}+\boldsymbol{\delta}}(\boldsymbol{y}|\boldsymbol{x}) \right) \right) = \frac{1}{2} \boldsymbol{\delta}^{\top} \boldsymbol{F} \boldsymbol{\delta} + \mathcal{O}\left( \|\boldsymbol{\delta}\|^3 \right) \tag{16}$$

and thus contains information about the geometry of the space distributions and the loss landscape. Additionally, when $\boldsymbol{\theta}$ is a local minimum, the FIM is identical to the Hessian of the loss. The FIM is computed using gradients of the model on *a single datapoint* and specifically using only gradients of the output $\nabla_{\boldsymbol{\theta}} \boldsymbol{f}_{\boldsymbol{\theta}}(\boldsymbol{x}_i)$. For regression

$$
\begin{aligned}
\boldsymbol{F} &= \mathbb{E}_{\boldsymbol{x}\sim\mathcal{D}, \boldsymbol{y}\sim p_{\boldsymbol{\theta}}(\boldsymbol{y}|\boldsymbol{x})} \left( \nabla_{\boldsymbol{\theta}} \log(p_{\boldsymbol{\theta}}(\boldsymbol{y}|\boldsymbol{x})) \nabla_{\boldsymbol{\theta}} \log(p_{\boldsymbol{\theta}}(\boldsymbol{y}|\boldsymbol{x}))^{\top} \right) \\
&= \mathbb{E}_{\boldsymbol{x}\sim\mathcal{D}, \boldsymbol{y}\sim\mathcal{N}(\boldsymbol{y};\boldsymbol{f}_{\boldsymbol{\theta}}(\boldsymbol{x}),\boldsymbol{I})} \left( \left( \nabla_{\boldsymbol{\theta}} \frac{1}{2} \|\boldsymbol{f}_{\boldsymbol{\theta}}(\boldsymbol{x}) - \boldsymbol{y}\|^2 \right) \left( \nabla_{\boldsymbol{\theta}} \frac{1}{2} \|\boldsymbol{f}_{\boldsymbol{\theta}}(\boldsymbol{x}) - \boldsymbol{y}\|^2 \right)^{\top} \right) \\
&= \mathbb{E}_{\boldsymbol{x}\sim\mathcal{D}} \left( \nabla_{\boldsymbol{\theta}} \boldsymbol{f}_{\boldsymbol{\theta}}(\boldsymbol{x}) \nabla_{\boldsymbol{\theta}} \boldsymbol{f}_{\boldsymbol{\theta}}(\boldsymbol{x})^{\top} \underbrace{\mathbb{E}_{\boldsymbol{y}\sim\mathcal{N}(\boldsymbol{y};\boldsymbol{f}_{\boldsymbol{\theta}}(\boldsymbol{x}),\boldsymbol{I})} \left( \|\boldsymbol{f}_{\boldsymbol{\theta}}(\boldsymbol{x}) - \boldsymbol{y}\|^2 \right)}_{\equiv \boldsymbol{I}} \right) \\
&= \mathbb{E}_{\boldsymbol{x}\sim\mathcal{D}} \left( \nabla_{\boldsymbol{\theta}} \boldsymbol{f}_{\boldsymbol{\theta}}(\boldsymbol{x}) \nabla_{\boldsymbol{\theta}} \boldsymbol{f}_{\boldsymbol{\theta}}(\boldsymbol{x})^{\top} \right),
\end{aligned}
\tag{17}
$$

and for classification

$$
\begin{aligned}
\boldsymbol{F} &= \mathbb{E}_{\boldsymbol{x}\sim\mathcal{D}, \boldsymbol{y}\sim p_{\boldsymbol{\theta}}(\boldsymbol{y}|\boldsymbol{x})} \left( \nabla_{\boldsymbol{\theta}} \log(p_{\boldsymbol{\theta}}(\boldsymbol{y}|\boldsymbol{x})) \nabla_{\boldsymbol{\theta}} \log(p_{\boldsymbol{\theta}}(\boldsymbol{y}|\boldsymbol{x}))^{\top} \right) \\
&= \mathbb{E}_{\boldsymbol{x}\sim\mathcal{D}} \left( \sum_{k=1}^{C} p_{\boldsymbol{\theta}}(\boldsymbol{y} = k|\boldsymbol{x}) \nabla_{\boldsymbol{\theta}} \log(p_{\boldsymbol{\theta}}(\boldsymbol{y} = k|\boldsymbol{x})) \nabla_{\boldsymbol{\theta}} \log(p_{\boldsymbol{\theta}}(\boldsymbol{y} = k|\boldsymbol{x}))^{\top} \right) \\
&= \mathbb{E}_{\boldsymbol{x}\sim\mathcal{D}} \left( \sum_{k=1}^{C} \frac{1}{p_{\boldsymbol{\theta}}(\boldsymbol{y} = k|\boldsymbol{x})} \nabla_{\boldsymbol{\theta}} p_{\boldsymbol{\theta}}(\boldsymbol{y} = k|\boldsymbol{x}) \nabla_{\boldsymbol{\theta}} p_{\boldsymbol{\theta}}(\boldsymbol{y} = k|\boldsymbol{x})^{\top} \right)
\end{aligned}
\tag{18}
$$

**The natural gradient.**  The natural gradient is defined by preconditioning the gradient using the FIM

$$\nabla_{\mathrm{nat}} := \boldsymbol{F}_{\boldsymbol{\theta}}^{-1} \nabla_{\boldsymbol{\theta}} \mathcal{L}(\boldsymbol{\theta}) \tag{19}$$

Motivated from the perspective of information geometry [4], the natural gradient defines the direction in parameter space that gives the largest change in the loss per unit of change in the *predictive distribution* of the model, measured by KL-divergence. This is to be contrasted with the standard gradient, which is the direction that gives the largest change in the loss per unit of change in *parameters*. Natural gradients are a fundamental tool in optimization and can be used to accelerate training [25, 27, 55].

Natural gradient optimization requires dynamic computation, storing, and inversion of the FIM, whose size grows quadratically with the number of parameters, making it intractable at scale. This necessitates the use of approximations, such as K-FAC [26, 27, 55].

**FIM usage in pruning.** In pruning, we want to assign each weight of a trained model a saliency score that indicates its importance to the model's performance. The goal is to remove a certain percentage of weights with the lowest saliency, resulting in a compressed model with fewer parameters that still performs relatively well. The original OBD [45] and OBS [32] algorithms use the Hessian to

compute saliency scores. However, since the model is trained, the parameters are likely close to a local minimum, and the FIM is often used as an approximation of the Hessian.

Given parameters of a trained model $\boldsymbol{\theta}$, the OBD pruning saliency scores are given by

$$\boldsymbol{\theta}_{\text{OBD}} := \boldsymbol{\theta}^2 \odot \text{diag}(\boldsymbol{F}_{\boldsymbol{\theta}}), \tag{20}$$

where $\boldsymbol{\theta}^2$ is a matrix of the square of each parameter, and $\odot$ stands for point-wise product. The optimal brain surgeon (OBS) pruning saliency scores are given by

$$\boldsymbol{\theta}_{\text{OBS}} := \boldsymbol{\theta}^2 \oslash \text{diag}(\boldsymbol{F}_{\boldsymbol{\theta}}^{-1}), \tag{21}$$

where $\oslash$ denotes point-wise division.

# B  Extension to Transformers

In this section, we extend the results from the main text–originally detailed for MLPs–to transformer architectures. Similar extensions apply to a wide range of neural models, including CNNs, RNNs, and state-space models, as the key requirement is that the architecture consists of linear layers interleaved with nonlinearities. We begin by presenting the gradient decomposition for transformers, and then analyze its symmetry structure and the applicability of GradMetaNet.

## B.1  Gradient Decomposition for Transformers

As mentioned in Section 3, the gradient decomposition used in the paper generalizes to other types of neural architectures. For example, Grosse and Martens [27] discuss an extension to CNNs and Eschenhagen et al. [21] generalized this decomposition to other modern neural architectures, including transformers. Most transformer parameters are split between the fully-connected components (usually called FFNs or MLPs) and the attention layers. We have covered the MLP case in Section 3 and cover the rest in this section. Throughout this section, we use the following notation:

$$T = \text{sequence length}, \ d = d_{\text{model}}, \ h = \#\text{heads}, \ d_k = \frac{d}{h}, \ d_{\text{ff}} = \text{FFN expansion}, \ L = \#\text{blocks}.$$

For the reader's convenience, we use brown for transformer block indices, blue for attention head indices, and green for token indices. This notation and the notation for the rest of the section largely mirror Vaswani et al. [84].

**Forward computation of transformer block $l$.**  Table 3 details the forward pass of the $l$-th transformer block on an input sequence $\boldsymbol{s} = (\boldsymbol{x}_1, \ldots, \boldsymbol{x}_T)$. $l$ is the index of the transformer block and $j$ is the head index.

Table 3: Transformer forward-pass.

| Object | Definition | Shape |
|---|---|---|
| Hidden input | $\boldsymbol{H}^{(l-1)}$ | $T \times d$ |
| Queries | $\boldsymbol{Q}^{(l,j)} = \boldsymbol{H}^{(l-1)}\boldsymbol{W}_Q^{(l,j)} + \boldsymbol{b}_Q^{(l,j)}$ | $T \times d_k$ |
| Keys | $\boldsymbol{K}^{(l,j)} = \boldsymbol{H}^{(l-1)}\boldsymbol{W}_K^{(l,j)} + \boldsymbol{b}_K^{(l,j)}$ | $T \times d_k$ |
| Values | $\boldsymbol{V}^{(l,j)} = \boldsymbol{H}^{(l-1)}\boldsymbol{W}_V^{(l,j)} + \boldsymbol{b}_V^{(l,j)}$ | $T \times d_k$ |
| Attention weights | $\boldsymbol{A}^{(l,j)} = \text{softmax}\big(\boldsymbol{Q}^{(l,j)}\boldsymbol{K}^{(l,j)\top}/\sqrt{d_k}\big)$ | $T \times T$ |
| Head output | $\boldsymbol{O}^{(l,j)} = \boldsymbol{A}^{(l,j)}\boldsymbol{V}^{(l,j)}$ | $T \times d_k$ |
| Merged heads | $\boldsymbol{O}^{(l)} = [\boldsymbol{O}^{(l,1)}, \ldots, \boldsymbol{O}^{(l,h)}]\boldsymbol{W}_O^{(l)} + \boldsymbol{b}_O^{(l)}$ | $T \times d$ |
| Post-MHA state | $\widehat{\boldsymbol{H}}^{(l)} = \text{LayerNorm}\big(\boldsymbol{H}^{(l-1)} + \boldsymbol{O}^{(l)}\big)$ | $T \times d$ |
| Feed-forward pre-act. | $\boldsymbol{P}^{(l)} = \widehat{\boldsymbol{H}}^{(l)}\boldsymbol{W}_1^{(l)} + \boldsymbol{b}_1^{(l)}$ | $T \times d_{\text{ff}}$ |
| Feed-forward out. | $\boldsymbol{U}^{(l)} = \sigma(\boldsymbol{P}^{(l)})\boldsymbol{W}_2^{(l)} + \boldsymbol{b}_2^{(l)}$ | $T \times d$ |
| Layer output | $\boldsymbol{H}^{(l)} = \text{LayerNorm}\big(\widehat{\boldsymbol{H}}^{(l)} + \boldsymbol{U}^{(l)}\big)$ | $T \times d$ |

The transformer block parameters are presented in Table 4. Notes:

- In most implementations, the key, query, and value projections have no bias terms, i.e.

$$\boldsymbol{b}_Q^{(l,j)} \equiv \boldsymbol{b}_K^{(l,j)} \equiv \boldsymbol{b}_V^{(l,j)} \equiv \boldsymbol{0}. \tag{22}$$

We include these bias terms for generality, but they can be removed in most cases.

Table 4: Transformer parameters.

| Parameter | Description | Shape |
|---|---|---|
| $\boldsymbol{W}_Q^{(l,j)}$ | query projection (head $j$) | $d \times d_k$ |
| $\boldsymbol{W}_K^{(l,j)}$ | key projection (head $j$) | $d \times d_k$ |
| $\boldsymbol{W}_V^{(l,j)}$ | value projection (head $j$) | $d \times d_k$ |
| $\boldsymbol{b}_Q^{(l,j)}$ | query bias | $d_k$ |
| $\boldsymbol{b}_K^{(l,j)}$ | key bias | $d_k$ |
| $\boldsymbol{b}_V^{(l,j)}$ | value bias | $d_k$ |
| $\boldsymbol{W}_O^{(l)}$ | output projection (concat. heads $\rightarrow d$) | $(h\,d_k) \times d$ |
| $\boldsymbol{b}_O^{(l)}$ | output bias | $d$ |
| $\boldsymbol{W}_1^{(l)}$ | FFN expansion ($d \rightarrow d_{\mathrm{ff}}$) | $d \times d_{\mathrm{ff}}$ |
| $\boldsymbol{b}_1^{(l)}$ | FFN bias (layer 1) | $d_{\mathrm{ff}}$ |
| $\boldsymbol{W}_2^{(l)}$ | FFN contraction ($d_{\mathrm{ff}} \rightarrow d$) | $d_{\mathrm{ff}} \times d$ |
| $\boldsymbol{b}_2^{(l)}$ | FFN bias (layer 2) | $d$ |

- Our derivation follows the *post-LayerNorm* (Post-LN) convention: residual add $\rightarrow$ LayerNorm (as in the original transformers paper Vaswani et al. [84]). The analysis for Pre-LN transformers requires a slight adjustment to the derivation.
- Outside of the transformer block we would also typically have: the token embedding matrix $\boldsymbol{E} \in \mathbb{R}^{|\mathcal{V}| \times d}$, positional embeddings $\mathrm{PE} \in \mathbb{R}^{T \times d}$ (or a sinusoidal schedule), and an unembedding matrix $\boldsymbol{E}^\top \in \mathbb{R}^{d \times |\mathcal{V}|}$.

**Back-propagated (pre-activation) gradient signals.** Notice that all of the transformer parameters act as linear transformations (that are then sometimes passed to attention computations, LayerNorms, etc.). This means that we can use the observation from Appendix A.1 regarding gradient computation for linear layers. Specifically, for every tensor computed in the transformer forward-pass (Table 3) that is the *output of a linear transformation*, we store the gradient w.r.t. that tensor:

$$\boldsymbol{g}_Q^{(l,j)} = \frac{\partial \mathcal{L}_{\boldsymbol{s}}}{\partial \boldsymbol{Q}^{(l,j)}}, \quad \boldsymbol{g}_K^{(l,j)} = \frac{\partial \mathcal{L}_{\boldsymbol{s}}}{\partial \boldsymbol{K}^{(l,j)}}, \quad \boldsymbol{g}_V^{(l,j)} = \frac{\partial \mathcal{L}_{\boldsymbol{s}}}{\partial \boldsymbol{V}^{(l,j)}},$$

$$\boldsymbol{g}_O^{(l)} = \frac{\partial \mathcal{L}_{\boldsymbol{s}}}{\partial \boldsymbol{O}^{(l)}}, \quad \boldsymbol{g}_1^{(l)} = \frac{\partial \mathcal{L}_{\boldsymbol{s}}}{\partial \boldsymbol{P}^{(l)}}, \quad \boldsymbol{g}_2^{(l)} = \frac{\partial \mathcal{L}_{\boldsymbol{s}}}{\partial \boldsymbol{U}^{(l)}}.$$

All of these tensors, referred to as *tangents*, share the same shapes as their forward counterparts.

**Outer-product parameter gradients.** Because every weight matrix appears in an affine map $\boldsymbol{Y} = \boldsymbol{X}\boldsymbol{W} + \boldsymbol{b}$, as analyzed in Appendix A.1, its gradient factorizes exactly into an *activation block* $\boldsymbol{X}$ and a *tangent block* $\boldsymbol{g}$:

$$\nabla_{\boldsymbol{W}}\mathcal{L} = \boldsymbol{X}^\top \boldsymbol{g}, \qquad \nabla_{\boldsymbol{b}}\mathcal{L} = \boldsymbol{g}^\top \mathbf{1}_T. \tag{23}$$

Carrying this out for *all* parameters in the transformer block yields

$$\nabla_{\boldsymbol{W}_Q^{(l,j)}}\mathcal{L}_{\boldsymbol{s}} = (\boldsymbol{H}^{(l-1)})^\top \boldsymbol{g}_Q^{(l,j)}, \qquad\qquad \nabla_{\boldsymbol{b}_Q^{(l,j)}}\mathcal{L}_{\boldsymbol{s}} = (\boldsymbol{g}_Q^{(l,j)})^\top \mathbf{1}_T,$$

$$\nabla_{\boldsymbol{W}_K^{(l,j)}}\mathcal{L}_{\boldsymbol{s}} = (\boldsymbol{H}^{(l-1)})^\top \boldsymbol{g}_K^{(l,j)}, \qquad\qquad \nabla_{\boldsymbol{b}_K^{(l,j)}}\mathcal{L}_{\boldsymbol{s}} = (\boldsymbol{g}_K^{(l,j)})^\top \mathbf{1}_T,$$

$$\nabla_{\boldsymbol{W}_V^{(l,j)}}\mathcal{L}_{\boldsymbol{s}} = (\boldsymbol{H}^{(l-1)})^\top \boldsymbol{g}_V^{(l,j)}, \qquad\qquad \nabla_{\boldsymbol{b}_V^{(l,j)}}\mathcal{L}_{\boldsymbol{s}} = (\boldsymbol{g}_V^{(l,j)})^\top \mathbf{1}_T,$$

$$\nabla_{\boldsymbol{W}_O^{(l)}}\mathcal{L}_{\boldsymbol{s}} = \big[\boldsymbol{O}^{(l,1)}, \ldots, \boldsymbol{O}^{(l,h)}\big]^\top \boldsymbol{g}_O^{(l)}, \qquad\qquad \nabla_{\boldsymbol{b}_O^{(l)}}\mathcal{L}_{\boldsymbol{s}} = (\boldsymbol{g}_O^{(l)})^\top \mathbf{1}_T,$$

$$\nabla_{\boldsymbol{W}_1^{(l)}}\mathcal{L}_{\boldsymbol{s}} = (\widehat{\boldsymbol{H}}^{(l)})^\top \boldsymbol{g}_1^{(l)}, \qquad\qquad \nabla_{\boldsymbol{b}_1^{(l)}}\mathcal{L}_{\boldsymbol{s}} = (\boldsymbol{g}_1^{(l)})^\top \mathbf{1}_T,$$

$$\nabla_{\boldsymbol{W}_2^{(l)}}\mathcal{L}_{\boldsymbol{s}} = \sigma(\boldsymbol{P}^{(l)})^\top \boldsymbol{g}_2^{(l)}, \qquad\qquad \nabla_{\boldsymbol{b}_2^{(l)}}\mathcal{L}_{\boldsymbol{s}} = (\boldsymbol{g}_2^{(l)})^\top \mathbf{1}_T.$$

**Token-wise view ("sum of rank-1" form).**   Unfolding, for example,

$$\nabla_{\boldsymbol{W}_Q^{(l,j)}} = (\boldsymbol{H}^{(l-1)})^\top \boldsymbol{g}_Q^{(l,j)} = \sum_{t=1}^{T} (\boldsymbol{g}_Q^{(l,j)})_t (\boldsymbol{H}_t^{(l-1)})^\top, \tag{24}$$

reveals that *each weight gradient is a sum of $T$ rank-1 outer products*. Storing the pair $\left(\boldsymbol{H}_t^{(l-1)}, \boldsymbol{g}_{Q,t}^{(l,j)}\right)$ for every token $t$ is therefore sufficient to reconstruct $\nabla_{\boldsymbol{W}_Q^{(l,j)}}\mathcal{L}_s$ exactly, and identical statements hold for $\boldsymbol{W}_K^{(l,j)}, \boldsymbol{W}_V^{(l,j)}, \boldsymbol{W}_O^{(l)}, \boldsymbol{W}_1^{(l)}$, and $\boldsymbol{W}_2^{(l)}$.

**Compact gradient representation.**   For transformer block $l = 1, \ldots, L$, and head $j = 1, \ldots, h$, define:

$$\mathbf{g}_{\text{res}}^{(l)} := \big( \underbrace{\boldsymbol{H}^{(l-1)}}_{\text{activation}} \ \underbrace{\boldsymbol{g}_O^{(l)}}_{\substack{\text{attn. out.} \\ \text{tangents}}}, \ \underbrace{\widehat{\boldsymbol{H}^{(l)}}}_{\text{post-LN}}, \ \underbrace{\boldsymbol{g}_2^{(l)}}_{\substack{\text{FFN}_2 \\ \text{tangents}}} \big) \in \boldsymbol{\Gamma}_T^{\text{res}} := \mathbb{R}^{T \times d \times 4},$$

$$\mathbf{g}_{\text{attn}}^{(l)} := \big( \ \underbrace{\boldsymbol{g}_Q^{(l,1:h)}}_{\text{query tangents}}, \ \underbrace{\boldsymbol{g}_K^{(l,1:h)}}_{\text{key tangents}}, \ \underbrace{\boldsymbol{g}_V^{(l,1:h)}}_{\text{value tangents}}, \ \underbrace{\boldsymbol{O}^{(l,1:h)}}_{\text{MHA outputs}} \big) \in \boldsymbol{\Gamma}_T^{\text{attn}} := \mathbb{R}^{T \times (hd_k) \times 4}, \tag{25}$$

$$\mathbf{g}_{\text{hidden}}^{(l)} := \big( \underbrace{\sigma(\boldsymbol{P}^{(l)})}_{\text{hidden rep.}}, \ \underbrace{\boldsymbol{g}_1^{(l)}}_{\substack{\text{FFN}_1 \\ \text{tangents}}} \big) \in \boldsymbol{\Gamma}_T^{\text{hidden}} := \mathbb{R}^{T \times d_{\text{ff}} \times 2},$$

Collecting the whole set, we get the tensors

$$\mathbf{g}^{(l)} = \big(\mathbf{g}_{\text{res}}^{(l)}, \mathbf{g}_{\text{attn}}^{(l)}, \mathbf{g}_{\text{hidden}}^{(l)}\big) \in \boldsymbol{\Gamma}_T^{(l)} := \boldsymbol{\Gamma}_T^{\text{res}} \oplus \boldsymbol{\Gamma}_T^{\text{attn}} \oplus \boldsymbol{\Gamma}_T^{\text{hidden}}, \tag{26}$$

and

$$\mathbf{g} := (\mathbf{g}^{(1)}, \ldots, \mathbf{g}^{(L)}) \in \boldsymbol{\Gamma}_T := \boldsymbol{\Gamma}_T^{(1)} \oplus \cdots \oplus \boldsymbol{\Gamma}_T^{(L)}. \tag{27}$$

## B.2   Transformer (Decomposed) Gradient Symmetries

The permutation symmetry groups of the parameter spaces of general architectures, and transformers in particular, were analyzed in Kofinas et al. [41], Lim et al. [48], Zhou et al. [92]. Kofinas et al. [41], Lim et al. [48] identify permutation symmetries with automorphisms of the computation graph of $\boldsymbol{f}_\theta$, and Zhou et al. [92] analyzes the permutation symmetries of multi-dimensional tensors. To give a flavor of the adaptations needed in the transformer case, we first look at the effects of the residual connections. Intuitively, we need to tie together the neuron spaces of dimension $d$ (i.e., $\boldsymbol{H}^{(l-1)}, \boldsymbol{O}^{(l)}$, $\widehat{\boldsymbol{H}^{(l)}}$, and $\boldsymbol{U}^{(l)}$) under the same symmetry group ($S_d$) because of the residual connections. With this "symmetry tying" the residual connections and LayerNorms preserve permutation symmetries, since if $\boldsymbol{\Phi}, \boldsymbol{\Psi} : \mathbb{R}^d \to \mathbb{R}^d$ are $S_d$-equivariant functions, we have

$$(\boldsymbol{\Phi}+\boldsymbol{\Psi})(\sigma\cdot\boldsymbol{x}) = \boldsymbol{\Phi}(\sigma\cdot\boldsymbol{x})+\boldsymbol{\Psi}(\sigma\cdot\boldsymbol{x}) = \sigma\cdot\boldsymbol{\Phi}(\boldsymbol{x})+\sigma\cdot\boldsymbol{\Psi}(\boldsymbol{x}) = \sigma\cdot(\boldsymbol{\Phi}(\boldsymbol{x})+\boldsymbol{\Psi}(\boldsymbol{x})) = \sigma\cdot(\boldsymbol{\Phi}+\boldsymbol{\Psi})(\boldsymbol{x}),$$

and

$$\text{LayerNorm}(\sigma \cdot \boldsymbol{x}) = \sigma \cdot \text{LayerNorm}(\boldsymbol{x}).$$

The analysis of the permutation symmetry group of transformer weight spaces provided in Kofinas et al. [41], Lim et al. [48], Zhou et al. [92] follows similar observations. The resulting symmetry group is

$$G = \underbrace{S_T}_{\substack{\text{tokens}}} \times \underbrace{S_d}_{\substack{\text{residual} \\ \text{stream}}} \times \overbrace{(S_{d_k})^h}^{\times L \text{ transformer blocks}} \times \underbrace{S_{d_{\text{ff}}}}_{\substack{\text{FFN} \\ \text{hidden dim}}}. \tag{28}$$

Where, in our decomposed gradients case, $S_T$ acts on the sequence dimension of all spaces, $S_d$ acts on the second axis (the $d$-dimension) of all $\boldsymbol{\Gamma}_T^{\text{res}}$s, each $(S_{d_k})^h$ acts independently on the $\boldsymbol{V}^{(l,j)}$ and $\boldsymbol{O}^{(l,j)}$ components of each head in $\boldsymbol{\Gamma}_T^{\text{attn}}$, and each $S_{d_{\text{ff}}}$ acts on the hidden dimension of the FFN represented in $\boldsymbol{\Gamma}_T^{\text{hidden}}$.

We note that transformer parameter spaces exhibit other neural symmetries that are not modeled by the *permutation* symmetry group $G$. These symmetries include ReLU scaling symmetries [38] and general attention symmetries (the transformation $(\boldsymbol{Q}^{(l,j)}, \boldsymbol{K}^{(l,j)}) \mapsto (\boldsymbol{Q}^{(l,j)}\boldsymbol{R}, \boldsymbol{K}^{(l,j)}\boldsymbol{R}^{-\top})$ for

$\boldsymbol{R} \in \mathrm{GL}_{d_k}(\mathbb{R})$ results in the same attention matrix). Accounting for these symmetries is left for future work.

### B.3 Adapting GradMetaNet to Transformers

To implement a GradMetaNet version that can process transformer gradients, we need to make the following adaptations. First, we treat the sequence dimension as a batch dimension, optionally with additional positional encoding for the token index. Note that this positional encoding is not strictly required since, as can be seen in Equation 24, the full gradient is a sum over the rank-1 components and is therefore an $S_T$-invariant function of them. We then treat $\boldsymbol{\Gamma}_T^{\mathrm{res}}$, $\boldsymbol{\Gamma}_T^{\mathrm{attn}}$, and $\boldsymbol{\Gamma}_T^{\mathrm{hidden}}$ as we treat neuron spaces with an additional positional encoding for each attention head to convert the $S_{(hd_k)}$-equivariance of $L_{\boldsymbol{\Gamma}}$ to $(S_{d_k})^h$-equivariance. As mentioned in Appendix B.2, we treat the $L$ copies of $\boldsymbol{\Gamma}_T^{\mathrm{res}}$ as a single neuron space, since the symmetry structure of the residual stream is tied together.

## C   Gradient and Weight Spaces

This section formally defines the vector spaces used to represent (sets of) gradients and weights. For batch size $b$ and feature dimension $f$, the feature vector spaces $\boldsymbol{\Gamma}[f]$ and $\boldsymbol{\Gamma}_b[f]$, $\boldsymbol{\Theta}[f]$, and $\boldsymbol{\Theta}_b[f]$ are defined by

$$
\begin{aligned}
\boldsymbol{\Gamma}[f] &:= \boldsymbol{\Gamma}^{(0)}[f] \oplus \cdots \oplus \boldsymbol{\Gamma}^{(L)}[f] \\
\boldsymbol{\Gamma}_b[f] &:= \boldsymbol{\Gamma}_b^{(0)}[f] \oplus \cdots \oplus \boldsymbol{\Gamma}_b^{(L)}[f] \\
\boldsymbol{\Theta}[f] &:= \mathcal{W}^{(1)}[f] \oplus \mathcal{U}^{(1)}[f] \oplus \cdots \oplus \mathcal{W}^{(L)}[f] \oplus \mathcal{U}^{(L)}[f] \\
\boldsymbol{\Theta}_b[f] &:= \mathcal{W}_b^{(1)}[f] \oplus \mathcal{U}_b^{(1)}[f] \oplus \cdots \oplus \mathcal{W}_b^{(L)}[f] \oplus \mathcal{U}_b^{(L)}[f]
\end{aligned}
\tag{29}
$$

where,

$$
\boldsymbol{\Gamma}^{(l)}[f] := \mathbb{R}^{d_l \times f}, \boldsymbol{\Gamma}_b^{(l)}[f] := \mathbb{R}^{b \times d_l \times f}
$$
$$
\mathcal{W}^{(l)}[f] := \mathbb{R}^{d_l \times d_{l-1} \times f}, \mathcal{W}_b^{(l)}[f] := \mathbb{R}^{b \times d_l \times d_{l-1} \times f}
\tag{30}
$$
$$
\mathcal{U}^{(l)}[f] := \mathbb{R}^{d_l \times f}, \mathcal{U}_b^{(l)}[f] := \mathbb{R}^{b \times d_l \times f}
$$

Note that $\boldsymbol{\Gamma}[2] = \boldsymbol{\Gamma}$, $\boldsymbol{\Gamma}_b[2] = \boldsymbol{\Gamma}_b$, $\boldsymbol{\Theta}[1] = \boldsymbol{\Theta}$ and $\boldsymbol{\Theta}_b[1] = \boldsymbol{\Theta}_b$.

## D   Architecture Details

The following is a detailed description of each of the layers used in GradMetaNet.

### D.1   GradMetaNet

**The positional encoding map.**   Similarly to Lim et al. [48], Zhou et al. [93], we use a positional encoding map $\mathrm{PE} : \boldsymbol{\Gamma}_b \to \boldsymbol{\Gamma}_b[f]$ that concatenates a layer identifier to neurons in intermediate layers and a neuron identifier to each neuron in the first and last layers, i.e.

$$
\begin{aligned}
\mathrm{PE}(\mathbf{g}^{(0)})_{i,j,:} &= \left[\mathbf{g}_{i,j,:}^{(0)}, \boldsymbol{e}_{\mathrm{in}}(j)\right], \\
\mathrm{PE}(\mathbf{g}^{(l)})_{i,j,:} &= \left[\mathbf{g}_{i,j,:}^{(l)}, \boldsymbol{e}_{\mathrm{layer}}(l)\right], \quad \text{for } l = 1, \ldots, L-1, \\
\mathrm{PE}(\mathbf{g}^{(L)})_{i,j,:} &= \left[\mathbf{g}_{i,j,:}^{(L)}, \boldsymbol{e}_{\mathrm{out}}(j)\right],
\end{aligned}
$$

where $[\cdot, \cdot]$ denotes concatenation along the feature axis. $\boldsymbol{e}_{\mathrm{in}}$ and $\boldsymbol{e}_{\mathrm{out}}$ assign unique identifiers to each neuron in the input and output layers, respectively, and $\boldsymbol{e}_{\mathrm{layer}}$ assigns unique identifiers to each hidden layer. We implement all encoding maps using sinusoidal positional encoding [80].

**Gradient-set-to-gradient-set layers.**   $L_{\boldsymbol{\Gamma}_b} : \boldsymbol{\Gamma}_b[f_{\mathrm{in}}] \to \boldsymbol{\Gamma}_b[f_{\mathrm{out}}]$ are parametrized similarly to the interactions-across-sets layers introduced in Hartford et al. [31], and are implemented as

$$
L_{\boldsymbol{\Gamma}_b}\left(\mathbf{g}^{(l)}\right)_{i,j,:} = M_1 \mathbf{g}_{i,j,:}^{(l)} + M_2 \sum_{i'=1}^{b} \mathbf{g}_{i',j,:}^{(l)} + M_3 \sum_{l'=0}^{L} \sum_{j'=1}^{d_l} \mathbf{g}_{i,j',:}^{(l')} + M_4 \sum_{l'=0}^{L} \sum_{i'=1}^{b} \sum_{j'=1}^{d_{l'}} \mathbf{g}_{i',j',:}^{(l')}, \tag{31}
$$

for learnable $M_1, M_2, M_3, M_4 \in \mathbb{R}^{f_{\mathrm{out}} \times f_{\mathrm{in}}}$.

**Gradient-set-to-gradient pooling layer.** $L_{\text{Pool}} : \mathbf{\Gamma}_b[f_{\text{in}}] \rightarrow \mathbf{\Gamma}[f_{\text{out}}]$ is implemented as

$$L_{\text{Pool}}(\mathbf{g}^{(l)})_{j,:} = M_1 \sum_{i'=1}^{b} \mathbf{g}^{(l)}_{i',j,:} + M_2 \sum_{l'=0}^{L} \sum_{i'=1}^{b} \sum_{j'=1}^{d_{l'}} \mathbf{g}^{(l')}_{i',j',:}, \tag{32}$$

for learnable $M_1, M_2 \in \mathbb{R}^{f_{out} \times f_{\text{in}}}$.

**Gradient-to-gradient layers.** $L_{\mathbf{\Gamma}} : \mathbf{\Gamma}[f_{\text{in}}] \rightarrow \mathbf{\Gamma}[f_{\text{out}}]$ are parameterized as equivariant DeepSets networks [91], and take the form

$$L_{\mathbf{\Gamma}}(\mathbf{g}^{(l)})_{i,:} = M_1 \mathbf{g}^{(l)}_{i,:} + M_2 \sum_{l'=1}^{L} \sum_{i'=1}^{d_l} \mathbf{g}^{(l')}_{i',:}, \tag{33}$$

for learnable $M_1, M_2, \in \mathbb{R}^{f_{\text{out}} \times f_{\text{in}}}$.

**Gradient-to-weight component.** Similarly to the generalized product layer in Navon et al. [62], $L_{\text{Prod}} : \mathbf{\Gamma}[f_{\text{in}}] \rightarrow \Theta$ applies a pointwise MLP to the features associated with the neurons connected to each weight, or in the case of biases, to the feature vectors corresponding to the respective neuron.

$$\boldsymbol{W}^{(l)}_{i,j} = \text{MLP}_1([\mathbf{g}^{(l)}_{i,:}, \mathbf{g}^{(l+1)}_{j,:}]), \boldsymbol{b}^{(l)}_i = \text{MLP}_2(\mathbf{g}^{(l)}_{i,:}). \tag{34}$$

## D.2   GradMetaNet++

Similarly to GradMetaNet, A GradMetaNet++ model $\mathbf{\Phi}$ comprises updates of different types: a positional encoding layer PE, gradient-set-to-gradient-set updates $U_{\mathbf{\Gamma}_b}$, and a gradient-set-to-weight component $U_{\text{Prod}}$. A GradMetaNet++ model is parameterized as

$$\mathbf{\Phi} = U_{\text{Prod}} \circ U_{\mathbf{\Gamma}_b}{}^{(k)} \circ \cdots \circ U_{\mathbf{\Gamma}_b}{}^{(1)} \circ \text{PE}. \tag{35}$$

We now describe each of these layers.

**The positional encoding map.** The positional encoding map used for GradMetaNet++ is identical to the one used in GradMetaNet and described in Section 5.

**Gradient-sets-to-gradient-set updates.** $U_{\mathbf{\Gamma}_b} : \mathbf{\Gamma}_b[f_{\text{in}}] \rightarrow \mathbf{\Gamma}_b[f_{\text{out}}]$ Are attention variants of the $L_{\mathbf{\Gamma}_b}$ layers described in Section 5. For a given $\mathbf{g} \in \mathbf{\Gamma}_b[f_{\text{in}}]$ in order to compute $U_{\mathbf{\Gamma}_b}(\mathbf{g})$ we first compute set-wise attention, given by:

$$\text{Attention}_b^h(\mathbf{g})^{(l)}_{i,j,:} = \sum_{j=1}^{b} \text{softmax}\left(\frac{\langle M_Q^h \mathbf{g}^{(l)}_{k,j,:}, M_K^h \mathbf{g}^{(l)}_{i,j,:} \rangle}{\sqrt{f_{\text{in}}}}\right) M_V^h \mathbf{g}^{(l)}_{k,j,:}. \tag{36}$$

$$\text{Attention}_b(\mathbf{g})^{(l)}_{i,j,:} = M_O[\text{Attention}_b^1(\mathbf{g})^{(l)}_{i,j,:}, \ldots, \text{Attention}_b^H(\mathbf{g})^{(l)}_{i,j,:}] \tag{37}$$

Here $\langle \cdot, \cdot \rangle$ denotes inner product and $M_K^h, M_Q^h, M_V^h \in \mathbb{R}^{f_{\text{in}} \times f_{\text{in}}}$ are learnable matrices used in each attention head and $M_O \in \mathbb{R}^{f_{\text{in}} H \times f_{\text{out}}}$ is a final aggregation linear layer. We then compute gradient-wise attention, given by:

$$\text{Attention}_g^h(\mathbf{g})^{(l)}_{i,j,:} == \sum_{l'=0}^{L} \sum_{k=1}^{d_{l'}} \text{softmax}\left(\frac{\langle M_Q^h \mathbf{g}^{(l')}_{i,k,:}, M_K^h \mathbf{g}^{(l)}_{i,j,:} \rangle}{\sqrt{f_{\text{in}}}}\right) M_V^h \mathbf{g}^{(l')}_{i,k,:}. \tag{38}$$

$$\text{Attention}_g(\mathbf{g})^{(l)}_{i,j,:} = M_O[\text{Attention}_g^1(\mathbf{g})^{(l)}_{i,j,:}, \ldots, \text{Attention}_g^H(\mathbf{g})^{(l)}_{i,j,:}] \tag{39}$$

Here we slightly abuse notation and denote by $M_K^h. M_Q^h, M_V^h \in \mathbb{R}^{f_{\text{in}} \times f_{\text{in}}}$, $M_O \in \mathbb{R}^{f_{\text{in}} H \times f_{\text{out}}}$ learnable matrices different from those in equations 36 and 37. finally, the value of $U_{\mathbf{\Gamma}_b}(\mathbf{g})$ is given by

$$U_{\mathbf{\Gamma}_b}(\mathbf{g})^{(l)}_{i,j,:} = \text{MLP}(\mathbf{g}^{(l)}_{i,j,:} + \text{Attention}_b(\mathbf{g})^{(l)}_{i,j,:} + \text{Attention}_g(\mathbf{g})^{(l)}_{i,j,:}). \tag{40}$$

**Gradient-batch-to-weight update.** As GradMetaNet++ prioritizes empirical improvements over computational efficiency, we directly use a gradient-batch-to-weight update, which we found to yield better performance.

The gradient-to-weight mapping, $U_{\text{Prod}} : \boldsymbol{\Gamma}_b[f_{\text{in}}] \to \boldsymbol{\Theta}$, applies a pointwise MLP to the feature vectors associated with the neurons connected to each weight in every element of the batch. In the case of biases, the MLP is applied to the feature vectors corresponding to the respective neuron. Finally, the results are summed across the batch. Formally, this can be expressed by $U_{\text{Prod}}(\mathbf{g}) = (\boldsymbol{W}_1, \boldsymbol{b}_1 \ldots, \boldsymbol{W}_L, \boldsymbol{b}_L)$ where:

$$(\boldsymbol{W}_l)_{i,j} = \sum_{k=1}^{b} \text{MLP}_1([\mathbf{g}^{(l)}_{k,i,:}, \quad \mathbf{g}^{(l+1)}_{k,j,:}]), (\boldsymbol{b}_l)_i = \sum_{k=1}^{b} \text{MLP}_2(\mathbf{g}^{(l)}_{k,i,:}). \tag{41}$$

### D.3   Invariant GradMetaNet.

In some cases (e.g. evaluating influence functions) we want GradMetaNet to output a single invariant vector $\in \mathbb{R}^{f_{\text{out}}}$ rather than a parameter vector $\in \boldsymbol{\Theta}$. In this case we replace the $L_{\text{Prod}}$ component described in Section 5 with a $L_{\text{Vec}}$ layer described below, For an element $\mathbf{g} \in \boldsymbol{\Gamma}[f_{\text{in}}]$,

$$L_{\text{Vec}}(\mathbf{g}) = M \sum_{l'=1}^{L} \sum_{i'=1}^{d_l} \mathbf{g}^{(l')}_{i',:}. \tag{42}$$

Where $M \in \mathbb{R}^{f_{\text{out}} \times f_{\text{in}}}$ is a learnable matrix. This results in invariant vector outputs.

### D.4   Computational Complexity.

We analyze the space and runtime complexity of GradMetaNet and GradMetaNet++, comparing them to alternative approaches. Throughout this discussion, we denote by $P$ the number of parameters in the underlying MLP $\boldsymbol{f}_{\boldsymbol{\theta}}$, whose gradients are being processed, and denote the number of neurons by $N$. Since the gradient-batch-to-gradient-batch update is the most computationally intensive component in both architectures, we focus our analysis on this operation.

**GradMetaNet.** The gradient-batch-to-gradient-batch update in GradMetaNet consists of a stack of layers $L_{\boldsymbol{\Gamma}_b} : \boldsymbol{\Gamma}_b[f_{\text{in}}] \to \boldsymbol{\Gamma}_b[f_{\text{out}}]$, as defined in Section 5. Each of these layers has both space and runtime complexity of $O(N \cdot b \cdot f_{\text{in}} \cdot f_{\text{out}})$. Thus, assuming a fixed hidden dimension and number of layers, GradMetaNet has a complexity of $O(N \cdot b)$.

**Gradient concatenation and averaging in weight space.** Both gradient concatenation and averaging methods process sets of gradients by utilizing weight-space architectures, such as those introduced by Lim et al. [48], Navon et al. [61]. These architectures employ layers $L_w : \boldsymbol{\Theta}[f_{\text{in}}] \to \boldsymbol{\Theta}[f_{\text{out}}]$ with time and space complexity $O(P \cdot f_{\text{in}} \cdot f_{\text{out}})$.

In the *concatenation* approach, gradients are concatenated, producing an input element in $\boldsymbol{\Theta}[b]$, resulting in an overall complexity of $O(P \cdot b)$ for a fixed hidden dimension and number of layers. This approach scales poorly compared to GradMetaNet when $b \cdot P > b \cdot N$. In contrast, the *averaging* approach reduces gradients to a single representation in $\boldsymbol{\Theta}[1]$, yielding a complexity of $O(P)$. However, this method is also suboptimal in the overparameterized regime ($P > b \cdot N$). In addition, even in cases where the batch size is sufficiently large such that $P < b \cdot N$, gradient-averaging methods may still be suboptimal due to their expressivity limitations (see Section 6).

**GradMetaNet++.** The gradient-batch-to-gradient-batch update in GradMetaNet++ is implemented using a stack of layers $U_{\boldsymbol{\Gamma}_b} : \boldsymbol{\Gamma}_b[f_{\text{in}}] \to \boldsymbol{\Gamma}_b[f_{\text{out}}]$, as detailed in Appendix D.2. Each layer has a time complexity of $O((N^2 \cdot b + b^2 \cdot N) \cdot f_{\text{in}} \cdot f_{\text{out}})$ and can be designed to achieve a space complexity of $O(N \cdot b \cdot f_{\text{in}} \cdot f_{\text{out}})$. While these layers have the highest time complexity among the approaches considered so far, their space complexity remains efficient. Moreover, constructing an attention-based variant for weight-space architectures would scale quadratically with $P$, making it far less practical in terms of scalability.

# E Theory

In this section, we provide proofs and further discussion for the results presented in Section 6. Throughout this section, we use $\nabla_i$ to denote gradients of the networks computed on a single datapoint.

## E.1 Importance of Processing Collections of Gradients

In Section 6, we discuss the expressivity limitations of processing the gradient of the average loss on the batch compared to the collection of gradients at each of the datapoints. In this section, we formalize these limitations. We start with some notation and definitions. As we saw in Appendix A.3, the FIM can be computed using gradients on individual datapoints. Given a set of such gradients $\mathcal{G} = \{\nabla_1, \ldots, \nabla_b\}$, the FIM computed using the gradients in $\mathcal{G}$ is denoted by $\boldsymbol{F}_{\mathcal{G}}$. As we saw in the main text, $\mathcal{G}$ can be thought of as an element of $\boldsymbol{\Theta}_b$.

**Definition E.1.** Let $\boldsymbol{\Phi} : \boldsymbol{\Theta} \times \boldsymbol{\Theta}_b \to \boldsymbol{\Theta}$ be a function whose inputs are parameters $\boldsymbol{\theta}$ and a set of gradients $\mathcal{G} = \{\nabla_1, \ldots, \nabla_b\}$. We say that $\boldsymbol{\Phi}$ non-trivially depends on the FIM if for some function $\boldsymbol{\Psi} : (\boldsymbol{\Theta} \otimes \boldsymbol{\Theta}) \times \boldsymbol{\Theta} \times \boldsymbol{\Theta} \to \boldsymbol{\Theta}$,

$$\boldsymbol{\Phi}(\boldsymbol{\theta}, \mathcal{G}) = \boldsymbol{\Psi}\left(\boldsymbol{F}_{\mathcal{G}}, \boldsymbol{\theta}, \frac{1}{b} \sum_{i=1}^{b} \nabla_i\right) \tag{43}$$

and there exists a pair of inputs $\boldsymbol{\theta}, \mathcal{G} = \{\nabla_1, \ldots, \nabla_b\}$ and $\boldsymbol{\theta}', \mathcal{G}' = \{\nabla'_1, \ldots, \nabla'_b\}$ where $\mathcal{G}$ and $\mathcal{G}'$ are admissible gradient sets [3] and such that $\boldsymbol{\theta} = \boldsymbol{\theta}'$, $\frac{1}{b} \sum_{i=1}^{b} \nabla_i = \frac{1}{b} \sum_{i=1}^{b} \nabla'_i$ but

$$\boldsymbol{\Phi}(\boldsymbol{\theta}, \mathcal{G}) = \boldsymbol{\Psi}\left(\boldsymbol{F}_{\mathcal{G}}, \boldsymbol{\theta}, \frac{1}{b} \sum_{i=1}^{b} \nabla_i\right) \neq \boldsymbol{\Psi}\left(\boldsymbol{F}_{\mathcal{G}'}, \boldsymbol{\theta}', \frac{1}{b} \sum_{i=1}^{b} \nabla'_i\right) = \boldsymbol{\Phi}(\boldsymbol{\theta}', \mathcal{G}'). \tag{44}$$

Many commonly used functions over sets of gradients non-trivially depend on the FIM. Before providing such examples, we first state the following trivial proposition.

**Proposition E.2.** *Let $\boldsymbol{\Phi} : \boldsymbol{\Theta} \times \boldsymbol{\Theta}_b \to \boldsymbol{\Theta}$ be a function that non-trivially depends on the FIM. There exist an $\epsilon > 0$ such that for any continuous function $\boldsymbol{\Lambda} : \boldsymbol{\Theta} \times \boldsymbol{\Theta} \to \boldsymbol{\Theta}$ it holds that:*

$$\max_{\boldsymbol{\theta}, \mathcal{G}} \left\| \boldsymbol{\Phi}(\boldsymbol{\theta}, \mathcal{G}) - \boldsymbol{\Lambda}\left(\boldsymbol{\theta}, \frac{1}{b} \sum \nabla_i\right) \right\| > \epsilon. \tag{45}$$

In other words, functions that non-trivially depend on the FIM cannot be approximated (in the $\ell_\infty$-sense) by continuous functions that rely only on the average gradient.

*Proof.* The proof follows trivially from Definition E.1. Let $\boldsymbol{\theta}, \mathcal{G} = \{\nabla_1, \ldots, \nabla_b\}$ and $\boldsymbol{\theta}', \mathcal{G}' = \{\nabla'_1, \ldots, \nabla'_b\}$ be a pair of inputs such that $\boldsymbol{\theta} = \boldsymbol{\theta}'$ and $\frac{1}{b} \sum_{i=1}^{b} \nabla_i = \frac{1}{b} \sum_{i=1}^{b} \nabla'_i$, but

$$\boldsymbol{\Phi}(\boldsymbol{\theta}, \mathcal{G}) \neq \boldsymbol{\Phi}(\boldsymbol{\theta}', \mathcal{G}') \tag{46}$$

For any $\boldsymbol{\Lambda} : \boldsymbol{\Theta} \times \boldsymbol{\Theta} \to \boldsymbol{\Theta}$ we have

$$\boldsymbol{\Lambda}\left(\boldsymbol{\theta}, \frac{1}{b} \sum \nabla_i\right) = \boldsymbol{\Lambda}\left(\boldsymbol{\theta}', \frac{1}{b} \sum \nabla'_i\right). \tag{47}$$

Therefore, if we choose $0 < \epsilon < 2\|\boldsymbol{\Phi}(\boldsymbol{\theta}, \mathcal{G}) - \boldsymbol{\Phi}(\boldsymbol{\theta}', \mathcal{G}')\|$ we have

$$\left\|\boldsymbol{\Phi}(\boldsymbol{\theta}, \mathcal{G}) - \boldsymbol{\Lambda}\left(\boldsymbol{\theta}, \frac{1}{b} \sum \nabla_i\right)\right\| + \left\|\boldsymbol{\Phi}(\boldsymbol{\theta}', \mathcal{G}') - \boldsymbol{\Lambda}\left(\boldsymbol{\theta}', \frac{1}{b} \sum \nabla'_i\right)\right\| \geq \|\boldsymbol{\Phi}(\boldsymbol{\theta}, \mathcal{G}) - \boldsymbol{\Phi}(\boldsymbol{\theta}', \mathcal{G}')\| > 2\epsilon. \tag{48}$$

This implies that either $\|\boldsymbol{\Phi}(\boldsymbol{\theta}, \mathcal{G}) - \boldsymbol{\Lambda}(\boldsymbol{\theta}, \frac{1}{b} \sum \nabla_i)\| > \epsilon$ or $\|\boldsymbol{\Phi}(\boldsymbol{\theta}', \mathcal{G}') - \boldsymbol{\Lambda}(\boldsymbol{\theta}', \frac{1}{b} \sum \nabla'_i)\|$ and so

$$\max_{\boldsymbol{\theta}, \mathcal{G}} \|\boldsymbol{\Phi}(\boldsymbol{\theta}, \mathcal{G}) - \boldsymbol{\Lambda}\left(\boldsymbol{\theta}, \frac{1}{b} \sum \nabla_i\right)\| > \epsilon \tag{49}$$

completing the proof. □

---

[3] Here, by "admissible", we mean that the elements of $\mathcal{G}$ and $\mathcal{G}'$ are actual MLP gradients, rather than arbitrary elements of $\boldsymbol{\Theta}$.

We want to show that the computation of the natural gradient and the OBD/OBS pruning saliency scores non-trivially depends on the FIM. To do so, we first formally define these computations as functions over $\Theta_b$.

**Definition E.3** (Natural gradient map). The natural gradient map $\mathbf{\Phi}_{\mathrm{nat}} : \mathbf{\Theta} \times \mathbf{\Theta}_b \to \mathbf{\Theta}$ is defined by

$$\mathbf{\Phi}_{\mathrm{nat}}(\nabla, \mathcal{G}) = (\boldsymbol{F}_\mathcal{G} + \epsilon \boldsymbol{I})^{-1} \nabla, \tag{50}$$

where $\boldsymbol{I}$ is the identity matrix and $\epsilon > 0$ is a damping factor. These are added since, while positive-definite, the FIM is not guaranteed to be invertible.

**Definition E.4** (OBD/OBS pruning saliency maps). The OBD saliency map $\mathbf{\Phi}_{\mathrm{OBD}} : \mathbf{\Theta} \times \mathbf{\Theta}_b \to \mathbf{\Theta}$ is defined by

$$\mathbf{\Phi}_{\mathrm{OBD}}(\boldsymbol{\theta}, \mathcal{G}) := \boldsymbol{\theta}^2 \odot \mathrm{diag}(\boldsymbol{F}_\mathcal{G}). \tag{51}$$

The OBS saliency map $\mathbf{\Phi}_{\mathrm{OBS}} : \mathbf{\Theta} \times \mathbf{\Theta}_b \to \mathbf{\Theta}$ is defined by

$$\mathbf{\Phi}_{\mathrm{OBS}}(\boldsymbol{\theta}, \mathcal{G}) := \boldsymbol{\theta}^2 \oslash \mathrm{diag}((\boldsymbol{F}_\mathcal{G} + \epsilon \boldsymbol{I})^{-1}). \tag{52}$$

We now show that both the natural gradient map and the OBD/OBS pruning saliency maps non-trivially depend on the FIM.

**Proposition E.5.** *The maps* $\mathbf{\Phi}_{\mathrm{nat}}$, $\mathbf{\Phi}_{\mathrm{OBD}}$, *and* $\mathbf{\Phi}_{\mathrm{OBS}}$ *non-trivially depend on the FIM.*

*Proof.* To start, we assume that $\boldsymbol{f_\theta}$ is a single-layer MLP, i.e., a linear map from the input space $\mathbb{R}^n$ to the output space $\mathbb{R}$. The proof can be extended to deeper MLPs by composing the linear map with an MLP that implements the identity function. Given a batch of datapoints $\mathcal{D} = \{(\boldsymbol{x}_1, \boldsymbol{y}_1), \ldots, (\boldsymbol{x}_N, \boldsymbol{y}_N)\} \subset \mathbb{R}^n \times \mathbb{R}$, the gradients of the output are

$$\nabla_i = \nabla_{\boldsymbol{\theta}} \boldsymbol{f_\theta}(\boldsymbol{x}_i) = \boldsymbol{x}_i. \tag{53}$$

Thus, as discussed in Appendix A.3, the FIM on $\mathcal{G} = \{\nabla_1, \ldots, \nabla_n\}$ can be computed as

$$\boldsymbol{F}_\mathcal{G} = \frac{1}{b} \sum_{i=1}^b \boldsymbol{x}_i \boldsymbol{x}_i^\top. \tag{54}$$

We begin by showing that $\mathbf{\Phi}_{\mathrm{nat}}$ non-trivially depends on the FIM. This is equivalent to showing that there exist two choices $\mathcal{B} = \{(\boldsymbol{x}_1, \boldsymbol{y}_1), \ldots, (\boldsymbol{x}_b, \boldsymbol{y}_b)\}$, $\mathcal{B}' = \{(\boldsymbol{x}'_1, \boldsymbol{y}'_1), \ldots, (\boldsymbol{x}'_b, \boldsymbol{y}'_b)\}$ with corresponding gradients $\mathcal{G} = \{\nabla_1, \ldots, \nabla_b\}$, $\mathcal{G}' = \{\nabla'_1, \ldots, \nabla'_b\}$ such that $\frac{1}{b} \sum_{i=1}^b \nabla_i = \frac{1}{b} \nabla'_i$ but, for some gradient $\nabla$,

$$(\boldsymbol{F}_\mathcal{G} + \epsilon \boldsymbol{I})^{-1} \nabla \neq (\boldsymbol{F}_\mathcal{G} + \epsilon \boldsymbol{I})^{-1} \nabla. \tag{55}$$

We now construct such $\mathcal{B}$ and $\mathcal{B}'$, but emphasize that this is only one of many possible ways to construct such an example. First, take $\mathcal{D}_n = \{(\boldsymbol{x}_1, \boldsymbol{y}_1), \ldots, (\boldsymbol{x}_n, \boldsymbol{y}_n)\}$ such that $\{\boldsymbol{x}_i\}_{i=1}^n$ is an orthonormal basis, meaning $\boldsymbol{x}_i^\top \boldsymbol{x}_j = \delta_{i,j}$. This means that FIM is the identity

$$\boldsymbol{F}_\mathcal{G} = \frac{1}{n} \sum_{i=1}^n \boldsymbol{x}_i \boldsymbol{x}_i^\top = \boldsymbol{I}. \tag{56}$$

Thus,

$$(\boldsymbol{F}_\mathcal{G} + \epsilon \boldsymbol{I})^{-1} = \mathrm{diag}((1 + \epsilon)^{-1}, \ldots, (1 + \epsilon)^{-1}). \tag{57}$$

Now, define $\mathcal{D}'_n = \{(\boldsymbol{x}'_1, \boldsymbol{y}'_1), \ldots, (\boldsymbol{x}'_n, \boldsymbol{y}'_n)\}_{i=1}^b$ such that $\boldsymbol{x}'_1 = 2\boldsymbol{x}_1, \ldots, \boldsymbol{x}'_n = 2\boldsymbol{x}_n$. The FIM in this case is

$$\boldsymbol{F}_{\mathcal{G}'} = 4\boldsymbol{I} \tag{58}$$

and

$$(\boldsymbol{F}_{\mathcal{G}'} + \epsilon \boldsymbol{I})^{-1} = \mathrm{diag}((4 + \epsilon)^{-1}, \ldots, (4 + \epsilon)^{-1}). \tag{59}$$

Therefore, for any non-zero gradient $\nabla$ we have

$$\mathbf{\Phi}_{\mathrm{nat}}(\nabla, \mathcal{G}) = (\boldsymbol{F}_\mathcal{G} + \epsilon)^{-1} \nabla = (1 + \epsilon)^{-1} \nabla \neq (4 + \epsilon)^{-1} \nabla = (\boldsymbol{F}_{\mathcal{G}'} + \epsilon)^{-1} \nabla = \mathbf{\Phi}_{\mathrm{nat}}(\nabla, \mathcal{G}') \tag{60}$$

This proves $\mathbf{\Phi}_{\mathrm{nat}}$ non-trivially depends on the FIM. To see that $\mathbf{\Phi}_{\mathrm{OBD}}$ and $\mathbf{\Phi}_{\mathrm{OBS}}$ non-trivially depends on the FIM, take $\mathcal{D}_n$ and $\mathcal{D}_n$ as before, and choose any non-zero $\boldsymbol{\theta}$.

$\square$

The next proposition now follows from Propositions E.2 and E.5.

**Proposition E.6.** $\Phi_{\mathrm{nat}}$, $\Phi_{\mathrm{OBD}}$, *and* $\Phi_{\mathrm{OBS}}$ *cannot be approximated (in the $\ell_\infty$ sense) by continuous functions that rely only on the average gradient.*

## E.2 Universal Approximation Results

In the discussion below, we are concerned with functions from a compact input domain $\mathcal{K} \subset \mathbf{\Gamma}_b[f]$ such that $\mathcal{K} \cap \mathcal{E} = \emptyset$, where

$$\mathcal{E} := \bigcup_{l=1}^{L-1} \bigcup_{i_1=1}^{l} \bigcup_{i_2=i_1+1}^{l} \left\{ \mathbf{g} \in \mathbf{\Gamma}_b[f] \middle| \sum_{j=1}^{b} \mathbf{g}_{i_1,j,:}^{(l)} = \sum_{j=1}^{b} \mathbf{g}_{i_2,j,:}^{(l)} \right\} \tag{61}$$

is a finite union of linear spaces of co-dimension $f$. Similar assumptions over $gK$ were used for the universality proofs in Finkelshtein et al. [23], Maron et al. [53].

For the readers convinience, we recall that for an MLP with input dimension $d_0$, output dimension $d_L$, hidden dimensions $d_1, \ldots, d_{L-1}$, we define $G = S_{d_1} \times \cdots \times S_{d_{L-1}}$, and $G_b = S_b \times G$. $G$ acts naturally on the spaces $\Theta[f]$ and $\mathbf{\Gamma}[f]$, while $G_b$ has natural actions on the space $\Theta_b[f]$ and $\mathbf{\Gamma}_b[f]$. See Appendix C of definitions. These actions preserve the inherent symmetries of the spaces they are defined over, and so we aim to respect them through equivariance/invariance.

**Main universality proofs.**

**Theorem E.7.** *Let $\mathcal{K} \subset \mathbf{\Gamma}_b[f]$ be a compact domain such that $\mathcal{K} = \cup_{g \in G_b} g \cdot \mathcal{K}$ and $\mathcal{K} \cap \mathcal{E} = \emptyset$. GradMetaNet models are universal approximators (in $\|\cdot\|_\infty$ sense) of continuous $G_b$-equivariant functions from $\mathcal{K}$ to weight space $\Theta$.*

*Proof.* Let $\mathbf{\Phi} : \mathcal{K} \to \Theta$ be a continuous $G_b$ equivaraint function. From proposition E.10 there exists a continuous $G_b$ equivariant function $\mathbf{\Psi} : \mathcal{K} \to \mathbf{\Gamma}[f']$ and an $G$ equivariant function $\mathbf{\Lambda} : \mathbf{\Psi}(\mathcal{K}) \to \Theta$ such that $\mathbf{\Phi} = \mathbf{\Lambda} \circ \mathbf{\Psi}$. From proposition E.18 there exist a stack of layers $L_{\mathrm{Prod}} \circ L_{\mathbf{\Gamma}} \circ \cdots \circ L_{\mathbf{\Gamma}} \circ \mathrm{PE}^4$ which can approximate $\mathbf{\Lambda}$ over $\mathbf{\Psi}(\mathcal{K})$ to any precision. Additionally, the function $\mathrm{PE} \circ \mathbf{\Psi}$ is also continuous and equivariant and so, from proposition from proposition E.15 there exist a stack of layers $L_{\mathrm{Pool}} \circ L_{\mathbf{\Gamma}_b} \circ \cdots \circ L_{\mathbf{\Gamma}_b} \circ \mathrm{PE}$ which can approximate $\mathrm{PE} \circ \mathbf{\Psi}$ over $\mathcal{K}$ to any precision. Composing the two components together allows us to construct a GradMetaNet model $L_{\mathrm{Prod}} \circ L_{\mathbf{\Gamma}} \circ \cdots \circ L_{\mathbf{\Gamma}} \circ L_{\mathrm{Pool}} \circ L_{\mathbf{\Gamma}_b} \circ \cdots \circ L_{\mathbf{\Gamma}_b} \circ \mathrm{PE}$ which can approximate $\mathbf{\Phi} = \mathbf{\Lambda} \circ \mathbf{\Psi}$ to any precision. $\square$

As a result of Theorem E.7, we obtain the following formal statement of Corollary 6.3.

**Corollary E.8.** *Let $\mathcal{K} \subset \mathbf{\Gamma}_b[f]$ be a compact domain such that $\mathcal{K} = \cup_{\tau \in G_b} \tau \cdot \mathcal{K}$ and $\mathcal{K} \cap \mathcal{E} = \emptyset$. there exist GradMetaNet models can approximate the natural gradients (see Definition E.3) of elements of $\mathcal{K}$ to arbitrary precision. Additionally, by incorporating the parameters $\boldsymbol{\theta}$ of the MLP whose gradients are provided as input to GradMetaNet into the gradient-to-weight update, GradMetaNet models can approximate pruning saliency scores (see Definition E.4) with arbitrary precision.*

*Proof.* As was discussed in Section A.3, the natural gradients can be expressed as a function from decomposed gradient space $\mathbf{\Gamma}_b[3]$ to parameter space $\Theta$. This function is both continuous and equivariant, and thus Theorem E.7 shows GradMetaNet can approximate natural gradients. Additionally, the functions $\mathbf{\Phi}_1(\mathbf{g}) = \mathrm{diag}(\boldsymbol{F})$ and $\mathbf{\Phi}_2(\mathbf{g}) = 1/\mathrm{diag}((\boldsymbol{F} + \epsilon \boldsymbol{I})^{-1})$ are continuous equivariant functions from $\mathbf{\Gamma}_b$ to $\Theta$ and thus can be approximated using GradMetaNet models. Recall that the OBD and OBD pruning saliency scores are computed by $\mathbf{\Phi}_1(\mathbf{g}) \odot \boldsymbol{\theta}^2$ and $\mathbf{\Phi}_2(\mathbf{g}) \odot \boldsymbol{\theta}^2$ respectively.

The parameters $\boldsymbol{\theta} = (\boldsymbol{W}_1, \boldsymbol{b}_1, \ldots, \boldsymbol{W}_L, \boldsymbol{b}_L) \in \Theta$, can be naturally added to the gradient-to-weight component $L_{\mathrm{Prod}}$ (See Section 5) the following way: $L_{\mathrm{Prod}} : \mathbf{\Gamma}[f_{\mathrm{in}}] \oplus \Theta \to \Theta$ applies a pointwise MLP to the feature vectors associated with the neurons connected to each weight **along with the weight of the original MLP**, or in the case of biases, to the feature vectors corresponding to the respective neuron. I.e., $L_{\mathrm{Prod}}(\mathbf{g}, \boldsymbol{\theta}) = (\boldsymbol{V}_1, \boldsymbol{c}_1, \ldots, \boldsymbol{V}_L, \boldsymbol{c}_L)$ where

$$\boldsymbol{V}_{i,j}^{(l)} = \mathrm{MLP}_1([\mathbf{g}_{i,:}^{(l)}, \mathbf{g}_{j,:}^{(l+1)}, \boldsymbol{W}_{i,j}^{(l)}]), \quad \boldsymbol{c}_i^{(l)} = \mathrm{MLP}_2([\mathbf{g}_{i,:}^{(l)}, \boldsymbol{b}_i^{(l)}]). \tag{62}$$

As we established GradMetaNet is able to approximate the functions $\mathbf{\Phi}_1, \mathbf{\Phi}_2$ the update in Equation 62 can easily approximate $\boldsymbol{\theta}_{\mathrm{OBD}} = \boldsymbol{\theta}^2 \odot \mathbf{\Phi}_1$, and $\boldsymbol{\theta}_{\mathrm{OBS}} = \boldsymbol{\theta}^2 \odot \mathbf{\Phi}_2$. This completes the proof. $\square$

---

[4] Here PE is defined for $\mathbf{\Gamma}[f'] = \mathbf{\Gamma}_1[f']$, we thus abuse notation writing PE without indicating which space it operates on.

Finally, we include a proof of universality for the invariant case

**Proposition E.9.** *Let $\mathcal{K} \subset \mathbf{\Gamma}_b[f]$ be a compact domain such that $\mathcal{K} = \cup_{\tau \in G_b} \tau \cdot \mathcal{K}$ and $\mathcal{K} \cap \mathcal{E} = \emptyset$. Invariant GradMetaNet models are universal approximators (in $\| \cdot \|_\infty$ sense) of continuous $G_b$-invariant functions from $\mathcal{K}$ to $\mathbb{R}^d$.*

*Proof.* Let $\mathbf{\Phi} : \mathcal{K} \to \mathbb{R}^d$ be a continuous $G_b$ invariant function. From proposition E.15, the gradient-bag-to-gradient component of GradMetaNet is a universal approximator of continuous equivariant functions from $\mathcal{K}$ to $\mathbf{\Gamma}[d]$. We can extend $\mathbf{\Phi}$ to be an equivariant function $\tilde{\mathbf{\Phi}} : \mathcal{K} \to \mathbf{\Gamma}[d]$ defined by

$$\tilde{\mathbf{\Phi}}(\mathbf{g})_{i,:}^{(l)} = \mathbf{\Phi}(\mathbf{g}). \tag{63}$$

Since $\mathbf{\Phi}$ is continuous and invariant, $\tilde{\mathbf{\Phi}}$ is continuous and equivariant and can thus be approximated by the gradient-bag-to-gradient component of our method. Finally applying the gradient-to-vector pooling layer $L_{\text{Vec}}$ we get that our model can apprximate the function

$$\frac{1}{d_0 + \cdots + d_L} \sum_{l=0}^{L} \sum_{i=1}^{d_l} \tilde{\mathbf{\Phi}}(\mathbf{g})_{i,:}^{(l)} = \mathbf{\Phi}(\mathbf{g}). \tag{64}$$

This completes the proof. $\qquad\square$

We now prove all the lemmas and propositions used in the above discussion.

**Proof of proposition E.10.**

**Proposition E.10.** *Let $\mathcal{K} \subset \mathbf{\Gamma}_b[f]$ be a compact domain such that $\mathcal{K} = \cup_{\tau \in G_b} \tau \cdot \mathcal{K}$ and $\mathcal{K} \cap \mathcal{E} = \emptyset$ and let $\mathbf{\Phi} : K \to \Theta$ be a continuous $G_b$-equivariant function (here the $S_b$ component of $G_b$ acts on $\Theta$ trivially). There exists a pair of continuous functions $\mathbf{\Psi} : K \to \mathbf{\Gamma}[f']$, $\mathbf{\Lambda} : \mathbf{\Psi}(K) \to \Theta$ such that:*

- *$\mathbf{\Psi}$ is continuous and $G_b$-equivariant .*
- *$\mathbf{\Lambda}$ is continuous and $G$-equivariant.*
- *$\mathbf{\Phi} = \mathbf{\Psi} \circ \mathbf{\Lambda}$.*

*Proof.* First, Lemma E.12 states that there exists a continuous and $G_b$-equivariant function $\mathbf{\Psi} : \mathcal{E}^c \to \mathbf{\Gamma}[f']$ (where $\mathcal{E}^c = \{\mathbf{g} \in \mathbf{\Gamma}_b[f] \mid \mathbf{g} \notin \mathcal{E}\}$), such that for every $\mathbf{g}_1, \mathbf{g}_2 \in \mathcal{E}^c$

$$\mathbf{\Psi}(\mathbf{g}_1) = \mathbf{\Psi}(\mathbf{g}_2) \iff \exists \tau \in S_b \text{ s.t. } \mathbf{g}_1 = \tau \cdot \mathbf{g}_2. \tag{65}$$

.

Now let $\pi : \mathbf{\Gamma}_b[f] \to \mathbf{\Gamma}_b[f]/S_b$ be the projection map to the quotient space induced by the orbits of $S_b$. Note that the group $G$ acts naturally on the quotient space $\mathbf{\Gamma}_b[f]/S_b$ by:

$$\tau \cdot \{\sigma \cdot \mathbf{g} \mid \sigma \in S_b\} = \{\tau \cdot \sigma \cdot \mathbf{g} \mid \sigma \in S_b\}. \tag{66}$$

Additionally, as the set $K$ is compact, the set $\tilde{\mathcal{K}} = \pi(\mathcal{K})$ is also compact. Since $\mathbf{\Phi}, \mathbf{\Psi}$ are invariant to the action of $S_b$ and equivariant to $G$, Lemma E.13 implies that there exist continuous $G$-equivariant functions $\tilde{\mathbf{\Psi}} : \tilde{K} \to \mathbf{\Gamma}[f']$, $\tilde{\mathbf{\Phi}} : \tilde{K} \to \Theta$ such that $\mathbf{\Psi} = \tilde{\mathbf{\Psi}} \circ \pi$ and $\mathbf{\Phi} = \tilde{\mathbf{\Phi}} \circ \pi$. As $\mathbf{\Psi}$ is $S_b$-injective, the function $\tilde{\mathbf{\Psi}}$ is injective and thus the function $\tilde{\mathbf{\Psi}}^{-1} : \mathbf{\Psi}(\mathcal{K}) \to \tilde{\mathcal{K}}$ is well defined. Additionally, as $\tilde{\mathbf{\Psi}}$ is $G$ equivariant $\tilde{\mathbf{\Psi}}^{-1}$ is also $G$ equivariant. We now define $\mathbf{\Lambda} = \tilde{\mathbf{\Phi}} \circ \tilde{\mathbf{\Psi}}^{-1} : \mathbf{\Psi}(\mathcal{K}) \to \Theta$. Since $\tilde{\mathbf{\Phi}}$ and $\tilde{\mathbf{\Psi}}^{-1}$ are $G$-equivariant, $\mathbf{\Lambda}$ is $G$-equivariant. Additionally,

$$\mathbf{\Lambda} \circ \mathbf{\Psi} = \tilde{\mathbf{\Phi}} \circ \tilde{\mathbf{\Psi}}^{-1} \circ \mathbf{\Psi} = \tilde{\mathbf{\Phi}} \circ \tilde{\mathbf{\Psi}}^{-1} \circ \tilde{\mathbf{\Psi}} \circ \pi = \tilde{\mathbf{\Phi}} \circ \pi = \mathbf{\Phi}. \tag{67}$$

Finally, by Lemma E.14 $\mathbf{\Lambda}$ is continuous, completing the proof. $\qquad\square$

We now state and prove all the lemmas used in the proof of proposition E.10, starting with Lemma 3 from [53] restated below:

**Lemma E.11.** *Let $H < S_n$ act on $\mathbb{R}^{n \times f}$ by applying the same element $\tau \in H$ to each channel, then there exists a polynomial function $U : \mathbb{R}^{n \times f} \to \mathbb{R}^{f'}$ for some $f' \in \mathbb{N}$ for which $U(\mathbf{x}) = U(\mathbf{y})$ if and only if $\mathbf{x} = \tau \cdot \mathbf{y}$ for some $\tau \in H$.*

**Lemma E.12.** *There exists a continuous $G_b$-equivariant function $\Psi : \mathcal{E}^c \to \Gamma[f']$ such that for every $\mathbf{g}_1, \mathbf{g}_2 \in \mathcal{E}^c$*

$$\Psi(\mathbf{g}_1) = \Psi(\mathbf{g}_2) \iff \exists \tau \in S_b \text{ s.t. } \mathbf{g}_1 = \tau \cdot \mathbf{g}_2. \tag{68}$$

*Proof.* Let $U$ be the polynomial invariant function established in Lemma E.11 where $H = G_b$, and define $S : \mathcal{E}^c \to \Gamma[f]$ by

$$S(\mathbf{g})^{(l)}_{i,:} = \sum_{j=1}^{b} \mathbf{g}_{j,i,:}. \tag{69}$$

We now define $\Psi : \mathcal{E}^c \to \Gamma[f']$ by:

$$\Psi(\mathbf{g})^{(l)}_{i,:} = [S^{(l)}(\mathbf{g})_{i,:}, U(\mathbf{g})] \tag{70}$$

where $[\dots]$ represents concatenation along the feature dimension. Note that we slightly abuse the notation of the feature dimension, denoting it as $f'$ multiple times. We first notice that since $S$ is equivariant and continuous and $U$ is invariant and continuous, $\Psi$ is also equivariant and continuous. Now, for input vectors $\mathbf{g}_1, \mathbf{g}_2 \in \mathcal{E}^c$ if there exists a group element $\tau \in S_b$ such that $\mathbf{g}_1 = \tau \cdot \mathbf{g}_2$ then from equivariance we have $\Psi(\mathbf{g}_1) = \Psi(\tau \cdot \mathbf{g}_2) = \tau \cdot \Psi(\mathbf{g}_2) = \Psi(\mathbf{g}_2)$ where the last inequality holds as $S_b$ acts trivially on the output space $\Gamma[f]$. On the other hand, assume $\Psi(\mathbf{g}_1) = \Psi(\mathbf{g}_2)$. We consider 2 cases:

**Case 1**: for every $\tau_1, \tau_2 \in G_b = S_b \times G$ it holds that $\tau_1 \cdot \tau_2 \cdot \mathbf{g}_2 \neq \mathbf{g}_1$. In this case from the definition of $u$ it holds that $U(\mathbf{g}_1) \neq U(\mathbf{g}_2)$ and so $\Psi(\mathbf{g}_1) \neq \Psi(\mathbf{g}_2)$.

**Case 2**: There exist a pair $\tau_1, \tau_2 \in G_b = S_b \times G$ such that $\tau_1 \cdot \tau_2 \cdot \mathbf{g}_2 = \mathbf{g}_1$. Assume by contradiction that $\tau_2$ is not the identity. Recall that for every $\mathbf{g} \notin \mathcal{E}, l \in [L], i \neq j \in [d_l]$, it holds that

$$\sum_{k=1}^{b} \mathbf{g}^{(l)}_{k,i,:} \neq \sum_{k=1}^{b} \mathbf{g}^{(l)}_{k,j,:}. \tag{71}$$

Thus,

$$S(\mathbf{g}_1) \neq \tau_2 \cdot S(\mathbf{g}_1) = S(\tau_2 \cdot \mathbf{g}_1) = S(\tau_1 \cdot \tau_2 \cdot \mathbf{g}_1) = S(\mathbf{g}_2). \tag{72}$$

This implies that $\Psi(\mathbf{g}_1) \neq \Psi(\mathbf{g}_2)$. We have thus shown that $\Psi(\mathbf{g}_1) = \Psi(\mathbf{g}_2) \iff \exists \tau \in S_b$ s.t. $\mathbf{g}_1 = \tau \cdot \mathbf{g}_2$ completing the proof. $\square$

**Lemma E.13.** *Let $\mathcal{K} \subset \Gamma_b[f]$ be a compact set such that $\mathcal{K} = \cup_{g \in G_b} g \cdot \mathcal{K}$. Furthermore, let $\Phi : \mathcal{K} \to \Gamma[f']$ be a continuous $G_b$-equivariant function. Finally, let $\pi : \Gamma_b[f] \to \Gamma_b[f]/S_b$ denote the projection map to the quotient space induced by the orbits of $S_b$ and define $\tilde{\mathcal{K}} = \pi(\mathcal{K})$. The following holds:*

- *$G$ acts on $\Gamma_b[f]/S_b$ by $\tau \cdot \{\sigma \cdot \mathbf{g} \mid \sigma \in S_b\} = \{\tau \cdot \sigma \cdot \mathbf{g} \mid \sigma \in S_b\}$.*
- *$\pi$ is $G$ equivariant.*
- *There exists a continuous $G$-equivariant function $\tilde{\Phi} : \tilde{K} \to \Gamma[f']$ such that $\Phi = \tilde{\Phi} \circ \pi$.*

*Proof.* Recall that each element in $\Gamma_b[f]/S_b$ is of the form $\pi(\mathbf{g}) = \{\sigma \cdot \mathbf{g} \mid \sigma \in S_b\}$. The first statement is trivial, as

$$e \cdot \{\sigma \cdot \mathbf{g} \mid \sigma \in S_b\} = \{\sigma \cdot \mathbf{g} \mid \sigma \in S_b\} \tag{73}$$

and for every $\tau_1, \tau_2 \in G$ we have

$$(\tau_1 \cdot \tau_2) \cdot \{\sigma \cdot \mathbf{g} \mid \sigma \in S_b\} = \{\tau_1 \cdot \tau_2 \cdot \sigma \cdot \mathbf{g} \mid \sigma \in S_b\} = (\tau_1 \cdot (\tau_2 \cdot \{\sigma \cdot \mathbf{g} \mid \sigma \in S_b\}). \tag{74}$$

To prove the second statement, notice that for every $\tau \in G, \sigma \in S_b$, we have $\tau \cdot \sigma = \sigma \cdot \tau$. Thus

$$\pi(\tau \cdot \mathbf{g}) = \{\sigma \cdot \tau \cdot v \mid \sigma \in S_b\} = \{\tau \cdot \sigma \cdot \mathbf{g} \mid \sigma \in S_b\} = \tau \cdot \pi(\mathbf{g}). \tag{75}$$

Finally, to prove the last statement we recall that since $\Phi$ is continuous and $S_b$ invariant, from the definition of the projection map $\pi$ there exists a continuous function $\tilde{\Phi} : \tilde{K} \to \Gamma[f']$ such that $\Phi = \tilde{\Phi} \circ \pi$. To show that $\tilde{\Phi}$ is $G$ equivariant, we notice that for every $\tilde{\mathbf{g}} \in \tilde{K}$ there exists $\mathbf{g} \in K$ such that $\pi(\mathbf{g}) = \tilde{\mathbf{g}}$, thus for very $\tau \in G$

$$\tilde{\Phi}(\tau \cdot \tilde{\mathbf{g}}) = \tilde{\Phi}(\tau \cdot \pi(\mathbf{g})) = \tilde{\Phi}(\pi(\tau \cdot \mathbf{g})) = \Phi(\tau \cdot \mathbf{g}) = \tau \cdot \Phi(\mathbf{g}) = \tau \cdot \tilde{\Phi}(\pi(\mathbf{g})) = \tau \cdot \tilde{\Phi}(\tilde{\mathbf{g}}) \tag{76}$$

and so $\tilde{\Phi}$ is equivariant, completing the proof. $\qquad\square$

**Lemma E.14.** *Let $\mathcal{K} \subset \mathbb{R}^f$ be a compact domain and $\Phi : \mathcal{K} \to \mathbb{R}^{f'}$ be a continuous function such that $\Phi = \Lambda \circ \Psi$. If $\Psi$ is continuous, then $\Lambda$ is continuous on $\Psi(\mathcal{K})$.*

The proof of this lemma is identical to that of Lemma 5 from [53], still we provide a proof below.

*Proof.* Assume that this is incorrect, then there is a sequence $\boldsymbol{y}_i = \Psi(\boldsymbol{x}_i)$ such that $\boldsymbol{y}_i \to \boldsymbol{y}_0$ but $\Lambda(\boldsymbol{y}_i) \not\to \Lambda(\boldsymbol{y}_0)$. Without loss of generality, assume that $\boldsymbol{x}_i \to \boldsymbol{x}_0 \in \mathcal{K}$ (otherwise choose a converging subsequence). We have

$$\Phi(\boldsymbol{x}_i) = \Lambda(\Psi(\boldsymbol{x}_i)) = \Lambda(\boldsymbol{y}_i) \not\to \Lambda(\boldsymbol{y}_0) = \Lambda(\Psi(\boldsymbol{x}_0)) = \Phi(\boldsymbol{x}_0) \tag{77}$$

which is a contradiction to the continuity of $\Phi$. $\qquad\square$

**Proof of proposition E.15.**

**Proposition E.15.** *$\mathcal{K} \subset \Gamma_b[f]$ be a compact domain such that $\mathcal{K} = \cup_{g \in G_b} g \cdot \mathcal{K}$ and $\mathcal{K} \cap \mathcal{E} = \emptyset$, and let $\Phi : \mathcal{K} \to \Gamma[f']$ be a continuous $G_b$-equivariant function. For every $\epsilon > 0$ There exists a stack of layers $\boldsymbol{F} = L_{\mathrm{Pool}} \circ L_{\Gamma_b} \circ \cdots \circ L_{\Gamma_b} \circ \mathrm{PE}$ (As defined in Section 5) $\boldsymbol{F}^{GradMetaNet}$ such that for every $\boldsymbol{g} \in K$:*

$$\|\Phi(\boldsymbol{g}) - \boldsymbol{F}(\boldsymbol{g})\| < \epsilon. \tag{78}$$

*Proof.* From Lemma E.16, there exists a continuous $S_b \times S_{d_0 + \cdots + d_L}$-equivariant function $\Psi : \Gamma_b[f + k] \to \Gamma[f']$ such that $\Phi = \Psi \circ \mathrm{PE}$. As $\Psi$ is continuous and $S_b \times S_{d_0 + \cdots + d_L}$-equivariant, it was shown e.g. in [23, 53] that for each $\epsilon > 0$ there exists a Deep Symmetric Sets network for sets of sets of the form $\tilde{\boldsymbol{F}} = L_{\mathrm{Pool}} \circ L_{\Gamma_b} \circ \cdots \circ L_{\Gamma_b}$ such that for every $\boldsymbol{g} \in \mathrm{PE}(K)$:

$$\|\Psi(\boldsymbol{g}) - \tilde{\boldsymbol{F}}(\boldsymbol{g})\| < \epsilon. \tag{79}$$

This implies for every $\boldsymbol{g} \in \mathcal{K}$:

$$\|\Phi(\boldsymbol{g}) - \boldsymbol{F}(\boldsymbol{g})\| = \|\Psi(\mathrm{PE}(\boldsymbol{g})) - \tilde{\boldsymbol{F}}(\mathrm{PE}(\boldsymbol{g}))\| < \epsilon \tag{80}$$

Completing the proof. $\qquad\square$

**Lemma E.16.** *Let $\Phi : \Gamma_b[f] \to \Gamma[f']$ be a continuous $G_b$ equivariant function and let $\mathrm{PE} : \Gamma_b[f] \to \Gamma_b[f + k]$ be a positional encoding layer as defined in Section 5. there exists an $S_n \times S_{d_0 + \cdots + d_L}$ equivariant function $\Psi : \Gamma_b[f + k] \to \Gamma[f']$ such that $\Phi = \Psi \circ \mathrm{PE}$.*

*Proof.* As PE is injective we can define $\Lambda : \mathrm{PE}(\Gamma_b[f]) \to \Gamma[f']$ by $\Lambda(\boldsymbol{g}) = \Phi(\mathrm{PE}^{-1}(\boldsymbol{g}))$. We now extend $\Lambda$ to the domain $\cup_{\sigma \in S_{d_0 + \cdots + d_L}} \sigma \cdot \mathrm{PE}(\Gamma_b[f])$ by defining for every $\boldsymbol{g} \in \mathrm{PE}(\Gamma_b[f]), \sigma \in S_{d_0 + \cdots + d_L}, \Lambda(\sigma \cdot \boldsymbol{g}) = \sigma \cdot \Lambda(\boldsymbol{g})$. We show this extension is well defined (i.e., that there is no case where $\sigma_1 \cdot \boldsymbol{g} = \sigma_2 \cdot \boldsymbol{g}$ but $\sigma_1 \cdot \Lambda(\boldsymbol{g}) \neq \sigma_2 \cdot \Lambda(\boldsymbol{g})$). Let $\boldsymbol{g}' \in \Gamma_b[f], \boldsymbol{g} = \mathrm{PE}(\boldsymbol{g}')$ and $\sigma_1 \neq \sigma_2 \in S_{d_0 + \cdots + d_L}$ such that $\sigma_1 \cdot \boldsymbol{g} = \sigma_2 \cdot \boldsymbol{g}$. It follows that $\sigma_2^{-1} \cdot \sigma_1 \cdot \boldsymbol{g} = \boldsymbol{g}$, and thus from the definition of the positional encoding map PE, $\sigma_2^{-1} \cdot \sigma_1 \in G$ and $\sigma_2^{-1} \cdot \sigma_1 \cdot \boldsymbol{g}' = \boldsymbol{g}'$. Thus, since $\Phi$ is $G$-equivariant it holds that

$$\Lambda(\sigma_1 \cdot \boldsymbol{g}) = \sigma_1 \cdot \Lambda(\boldsymbol{g}) = \sigma_2 \cdot \sigma_2^{-1} \cdot \sigma_1 \cdot \Phi(\boldsymbol{g}') = \sigma_2 \cdot \Phi(\sigma_2^{-1} \cdot \sigma_1 \cdot \boldsymbol{g}') = \sigma_2 \cdot \Phi(\boldsymbol{g}') = \sigma_2 \cdot \Lambda(\boldsymbol{g}) = \Lambda(\sigma_2 \cdot \boldsymbol{g}). \tag{81}$$

Thus, the extension of $\Lambda$ is well defined. Since the functions $\Phi, \mathrm{PE}$ are continuous the function $\Lambda$ is continuous and thus there exists a continuous function $\tilde{\Lambda} : \Gamma_b[f] \to \Gamma[f']$ such that for every $\boldsymbol{g} \in \cup_{\sigma \in S_{d_0 + \cdots + d_L}} \sigma \cdot \mathrm{PE}(\Gamma_b[f])$ it holds that $\tilde{\Lambda}(\boldsymbol{g}) = \Lambda(\boldsymbol{g})$. We define $\Psi : \Gamma_b[f] \to \Gamma[f']$ by

$$\Psi(\boldsymbol{g}) = \sum_{\substack{\sigma \in S_{d_0 + \cdots + d_L} \\ \tau \in S_b}} \tau^{-1} \cdot \sigma^{-1} \cdot \tilde{\Lambda}(\tau \cdot \sigma \cdot \boldsymbol{g}). \tag{82}$$

First, $\Psi$ is continuous and equivariant w.r.t $S_b \times S_{d_0 + \cdots + d_L}$. Second, for every $\boldsymbol{g} \in \mathrm{PE}(\Gamma_b[f])$ it holds that

$$\Psi(\boldsymbol{g}) = \sum_{\substack{\sigma \in S_{d_0 + \cdots + d_L} \\ \tau \in S_b}} \tau^{-1} \cdot \sigma^{-1} \cdot \Lambda(\tau \cdot \sigma \cdot \boldsymbol{g}) = \sum_{\substack{\sigma \in S_{d_0 + \cdots + d_L} \\ \tau \in S_b}} \tau^{-1} \cdot \sigma^{-1} \cdot \tau \cdot \sigma \cdot \Lambda(\boldsymbol{g}) = \Lambda(\boldsymbol{g}). \tag{83}$$

Thus, for every $\boldsymbol{g}' \in \Gamma_b[f]$ it holds that $\Phi(\boldsymbol{g}') = \Phi(\mathrm{PE}^{-1}(\mathrm{PE}(\boldsymbol{g}))) = \Psi(\mathrm{PE}(\boldsymbol{g}'))$ and so $\Phi = \Psi \circ \mathrm{PE}$. $\qquad\square$

**Proof of proposition E.18.** In this section, we aim to leverage the universality result of the Set2Graph architecture presented in Serviansky et al. [76] to complete the universality proof of GradMetaNet. To this aim, we first define the square gradient space $\tilde{\mathbf{\Gamma}}[f]$. Intuitively, the spaces $\mathbf{\Gamma}[f]$ and $\tilde{\mathbf{\Gamma}}[f]$ parallel set space and graph space, and $\mathbf{\Theta}[f]$ can be embedded in $\tilde{\mathbf{\Gamma}}[f]$. We now formally define $\tilde{\mathbf{\Gamma}}[f]$.

**Definition E.17.** Square gradient space $\tilde{\mathbf{\Gamma}}[f]$ is defined by

$$\tilde{\mathbf{\Gamma}}[f] = (\mathbf{\Gamma} \otimes \mathbf{\Gamma})^f. \tag{84}$$

Here, $\otimes$ denotes the tensor product, while $^f$ represents the Cartesian power product. Thus, an element $\tilde{\mathbf{g}} \in \tilde{\mathbf{\Gamma}}[f]$ is of the form

$$\tilde{\mathbf{g}} = \{\tilde{\mathbf{g}}_{i,j,:}^{(l_1,l_2)}\}_{l_1,l_2\in[L]}, \tag{85}$$

where $i \in [d_{l_1}], j \in [d_{l_2}]$ and $\tilde{\mathbf{g}}_{i,j,:}^{(l_1,l_2)} \in \mathbb{R}^f$. The space $\mathbf{\Theta}[f]$ can be naturally embedded into $\tilde{\mathbf{\Gamma}}[2f]$ by the map $L_{\text{ode}}$ defined below for $\boldsymbol{\theta} \in \mathbf{\Theta}[f]$, $\boldsymbol{\theta} = (\boldsymbol{W}^{(1)}, \boldsymbol{b}^{(1)} \ldots, \boldsymbol{W}^{(L)}, \boldsymbol{b}^{(L)})$

$$L_{\text{ode}}(\boldsymbol{\theta})_{i,j,:}^{(l_1,l_2)} = \begin{cases} [\boldsymbol{W}_{i,j,:}^{(l_2)}, \boldsymbol{b}_{j,:}^{(l_2)}] & l_1 = l_2 + 1 \\ 0 & \text{otherwise.} \end{cases} \tag{86}$$

Additionally, the map $L_{\text{odp}} : \tilde{\mathbf{\Gamma}}[2f] \to \mathbf{\Theta}[f]$ projects the space $\tilde{\mathbf{\Gamma}}[2f]$ to $\mathbf{\Theta}[f]$ and is defined by $L_{\text{odp}}(\tilde{\mathbf{g}}) = (\boldsymbol{W}^{(1)}, \boldsymbol{b}^{(1)}, \ldots, \boldsymbol{W}^{(L)}, \boldsymbol{b}^{(L)})$

$$\boldsymbol{W}_{i,j,:}^{(l)} = \tilde{\mathbf{g}}_{i,j,:f}^{(l-1,l)}, \quad \boldsymbol{b}_{i:}^{(l)} = \tilde{\mathbf{g}}_{i,i,f:}^{(l-1,l)}. \tag{87}$$

The action of the group $G = S_{d_1} \times \cdots \times S_{d_{L-1}}$ on $\mathbf{\Gamma}[f]$ extend naturally to $\tilde{\mathbf{\Gamma}}[f]$ and is defined for $\tau = (\tau_1, \ldots, \tau_{L-1}) \in G$, $\mathbf{g} \in \tilde{\mathbf{\Gamma}}[f]$ by

$$(\tau \cdot \mathbf{g})_{i,j,:}^{(l_1,l_2)} = \mathbf{g}_{\tau_{l_1}^{-1}(i),\tau_{l_2}^{-1}(j),:}^{(l_1,l_2)}. \tag{88}$$

It is easy to verify that the maps $L_{\text{ode}}, L_{\text{odp}}$ are $G$-equivariant.

Before stating the following proposition, we note that the positional encoding map $\text{PE} : \mathbf{\Gamma}_b[f] \to \mathbf{\Gamma}_b[f + k]$ can be considered as well defined on the space $\mathbf{\Gamma}_b[f] = \mathbf{\Gamma}_1[f]$.

**Proposition E.18.** *Let $\mathcal{K} \subset \mathbf{\Gamma}[f]$ be a compact set such that $\mathcal{K} = \cup_{g\in G} g \cdot \mathcal{K}$, and let $\mathbf{\Phi} : \mathcal{K} \to \mathbf{\Theta}[f']$ be a continuous $G$-equivariant function. For every $\epsilon > 0$ There exists a stack of gradient-to-gradient layers composed with a gradient-to-weight layer and a positional encoding layer $\boldsymbol{F} = L_{\text{Prod}} \circ L_{\mathbf{\Gamma}} \circ \cdots \circ L_{\mathbf{\Gamma}} \circ \text{PE}$ such that for every $\mathbf{g} \in K$:*

$$\|\mathbf{\Phi}(\mathbf{g}) - \boldsymbol{F}(\mathbf{g})\| < \epsilon. \tag{89}$$

*Proof.* First, notice that the model $\boldsymbol{F}$ is equal to $L_{\text{odp}} \circ \bar{\boldsymbol{F}} \circ \text{PE}$ where $\bar{\boldsymbol{F}}$ is exactly a Set2Graph architecture, from the space $\mathbf{\Gamma}[f]$ to $\tilde{\mathbf{\Gamma}}[f']$ which is equivariant to the action of the group $S_{d_0+\cdots+d_L}$ on both spaces. This is because the DeepSet component in Set2Graph is identical to a stack of $L_{\mathbf{\Gamma}}$ layers and the broadcast and pointwise MLP components of Set2Graph are identical to the $L_{\text{Prod}}$ component composed with the embedding $L_{\text{ode}}$. From Lemma E.19, there exists a continuous $S_{d_0+\cdots+d_L}$ equivariant function $\mathbf{\Psi} : \mathbf{\Gamma}[f + k] \to \mathbf{\Theta}[2f']$ such that $\mathbf{\Phi} = L_{\text{odp}} \circ \mathbf{\Psi} \circ \text{PE}$. Finally, it was shown in [76] that there exists a Set2Graph network $\bar{\boldsymbol{F}}$ such that for every $\mathbf{g} \in \text{PE}(K)$:

$$\|\mathbf{\Psi}(\mathbf{g}) - \bar{\boldsymbol{F}}(\mathbf{g})\| < \epsilon. \tag{90}$$

This implies for every $\mathbf{g} \in K$ there exists a gradient to weight model $\boldsymbol{F}$ such that :

$$\|\mathbf{\Phi}(\mathbf{g}) - \boldsymbol{F}(\mathbf{g})\| = \|L_{\text{ode}}(\mathbf{\Psi}(\text{PE}(\mathbf{g}))) - L_{\text{ode}}(\bar{\boldsymbol{F}}(\text{PE}(\mathbf{g})))\| < \epsilon. \tag{91}$$

This completes the proof. $\square$

**Lemma E.19.** *Let $\mathbf{\Phi} : \mathbf{\Gamma}[f] \to \mathbf{\Theta}[f']$ be a continuous $G_b$ equivariant function. There exists an $S_b \times S_{d_0+\cdots+d_L}$ equivariant function $\mathbf{\Psi} : \mathbf{\Gamma}[f + k] \to \tilde{\mathbf{\Gamma}}[2f']$ such that $\mathbf{\Phi} = L_{\text{odp}} \circ \mathbf{\Psi} \circ \text{PE}$.*

*Proof.* The proof is very similar to that of Lemma E.16. Let $\bar{\mathbf{\Phi}} : \mathbf{\Gamma}[f] \to \tilde{\mathbf{\Gamma}}[2f']$ be defined by $\bar{\mathbf{\Phi}} = L_{\text{ode}} \circ \mathbf{\Phi}$ and note that since $\mathbf{\Phi}$ and $L_{\text{ode}}$ are continuous and $G$ equivariant, $\bar{\mathbf{\Phi}}$ is also continuous and $G$ equivariant. As PE is injective we can define $\mathbf{\Lambda} : \text{PE}(\mathbf{\Gamma}[f]) \to \tilde{\mathbf{\Gamma}}[2f']$ by $\mathbf{\Lambda}(\mathbf{g}) = \bar{\mathbf{\Phi}}(\text{PE}^{-1}(\mathbf{g}))$. We now extend $\mathbf{\Lambda}$ to the domain $\cup_{\sigma \in S_{d_0+\cdots+d_L}} \sigma \cdot \text{PE}(\mathbf{\Gamma}[f])$ by defining for

every $\mathbf{g} \in \mathrm{PE}(\mathbf{\Gamma}[f]), \sigma \in S_{d_0+\cdots+d_L}, \mathbf{\Lambda}(\sigma \cdot \mathbf{g}) = \sigma \cdot \mathbf{\Lambda}(\mathbf{g})$. We show this extension is well defined (i.e. that there is no case where $\sigma_1 \cdot \mathbf{g} = \sigma_2 \cdot \mathbf{g}$ but $\sigma_1 \cdot \mathbf{\Lambda}(\mathbf{g}) \neq \sigma_2 \cdot \mathbf{\Lambda}(\mathbf{g})$).

Let $\mathbf{g}' \in \mathbf{\Gamma}[f], \mathbf{g} = \mathrm{PE}(\mathbf{g}')$ and $\sigma_1 \neq \sigma_2 \in S_{d_0+\cdots+d_L}$ such that $\sigma_1 \cdot \mathbf{g} = \sigma_2 \cdot \mathbf{g}$. It follows that $\sigma_2^{-1} \cdot \sigma_1 \cdot \mathbf{g} = \mathbf{g}$, and thus from the definition of the positional encoding function PE, $\sigma_2^{-1} \cdot \sigma_1 \in G$ and $\sigma_2^{-1} \cdot \sigma_1 \cdot \mathbf{g}' = \mathbf{g}'$. Thus, since $\bar{\mathbf{\Phi}}$ is $G$ equivariant it holds that

$$\mathbf{\Lambda}(\sigma_1 \cdot \mathbf{g}) = \sigma_1 \cdot \mathbf{\Lambda}(\mathbf{g}) = \sigma_2 \cdot \sigma_2^{-1} \cdot \sigma_1 \cdot \bar{\mathbf{\Phi}}(\mathbf{g}') = \sigma_2 \cdot \bar{\mathbf{\Phi}}(\sigma_2^{-1} \cdot \sigma_1 \cdot \mathbf{g}') = \sigma_2 \cdot \bar{\mathbf{\Phi}}(\mathbf{g}') = \sigma_2 \cdot \mathbf{\Lambda}(\mathbf{g}) = \mathbf{\Lambda}(\sigma_2 \cdot \mathbf{g}). \tag{92}$$

Thus, the extension of $\mathbf{\Lambda}$ is well defined. Since the functions $\bar{\mathbf{\Phi}}, \mathrm{PE}$ are continuous the function $\mathbf{\Lambda}$ is continuous and thus there exsists a continuous function $\tilde{\mathbf{\Lambda}} : \mathbf{\Gamma}[f] \to \tilde{\mathbf{\Gamma}}[2f']$ such that for every $\mathbf{g} \in \cup_{\sigma \in S_{d_0+\cdots+d_L}} \sigma \cdot \mathrm{PE}(\mathbf{\Gamma}[f])$ it holds that $\tilde{\mathbf{\Lambda}}(\mathbf{g}) = \mathbf{\Lambda}(\mathbf{g})$. We define $\mathbf{\Psi} : \mathbf{\Gamma}[f] \to \tilde{\mathbf{\Gamma}}[2f']$ by

$$\mathbf{\Psi}(\mathbf{g}) = \sum_{\substack{\sigma \in S_{d_0+\cdots+d_L} \\ \tau \in S_b}} \tau^{-1} \cdot \sigma^{-1} \cdot \tilde{\mathbf{\Lambda}}(\tau \cdot \sigma \cdot \mathbf{g}). \tag{93}$$

First, $\mathbf{\Psi}$ is continuous and equivariant w.r.t $S_b \times S_{d_0+\cdots+d_L}$. Second, for every $\mathbf{g} \in \mathrm{PE}(\mathbf{\Gamma}[f])$ it holds that

$$\mathbf{\Psi}(\mathbf{g}) = \sum_{\substack{\sigma \in S_{d_0+\cdots+d_L} \\ \tau \in S_b}} \tau^{-1} \cdot \sigma^{-1} \cdot \mathbf{\Lambda}(\tau \cdot \sigma \cdot \mathbf{g}) = \sum_{\substack{\sigma \in S_{d_0+\cdots+d_L} \\ \tau \in S_b}} \tau^{-1} \cdot \sigma^{-1} \cdot \tau \cdot \sigma \cdot \mathbf{\Lambda}(\mathbf{g}) = \mathbf{\Lambda}(\mathbf{g}). \tag{94}$$

Thus, for every $\mathbf{g}' \in \mathbf{\Gamma}[f]$ it holds that

$$\bar{\mathbf{\Phi}}(\mathbf{g}') = \bar{\mathbf{\Phi}}(\mathrm{PE}^{-1}(\mathrm{PE}(\mathbf{g}'))) = \mathbf{\Psi}(\mathrm{PE}(\mathbf{g}')). \tag{95}$$

Finally, since $L_{\mathrm{odp}} \circ L_{\mathrm{ode}} = \mathrm{id}$ we have

$$\mathbf{\Phi} = L_{\mathrm{odp}} \circ \bar{\mathbf{\Phi}} = L_{\mathrm{odp}} \circ \mathbf{\Psi} \circ \mathrm{PE} \tag{96}$$

completing the proof. $\square$

### E.3 Importance of $\mathcal{E}$ in Universality

Recall that Theorem 6.2 proves the universality of GradMetaNet over compact domains which do not intersect the set $\mathcal{E} \subset \mathbf{\Gamma}_b[f]$ defined by:

$$\mathcal{E} = \bigcup_{l=1}^{L-1} \bigcup_{i_1=1}^{l} \bigcup_{i_2=i_1+1}^{l} \left\{ \mathbf{g} \in \mathbf{\Gamma}_b[f] \middle| \sum_{j=1}^{b} \mathbf{g}_{i_1,j,:}^{(l)} = \sum_{j=1}^{b} \mathbf{g}_{i_2,j,:}^{(l)} \right\} \tag{97}$$

In this section, we prove that there exist compact sets $\mathcal{K} \subset \mathbf{\Gamma}_b[f]$ with $\mathcal{K} \cap \mathcal{E} \neq \emptyset$ over which GradMetaNet is not universal. We emphasize that, as discussed below, sets intersecting $\mathcal{E}$ consist of highly regular gradient values, which deviate significantly from typical gradient sets. Consequently, the requirement $\mathcal{K} \cap \mathcal{E} = \emptyset$ is a mild and realistic assumption.

**Proposition E.20.** *There exist a pair of elements $\mathbf{g}_1, \mathbf{g}_2 \in \mathcal{E}$ such that for any $\tau \in G_b, \tau \cdot \mathbf{g}_1 \neq \mathbf{g}_2$ but for every GradMetaNet model $\mathbf{F}$ it holds that $\mathbf{F}(\mathbf{g}_1) = \mathbf{F}(\mathbf{g}_2)$.*

*Proof.* For simplicity, we focus in this proof on the invariant version of GradMetaNet, $\mathbf{F} : \mathbf{\Gamma}_b[f] \to \mathbb{R}^{f'}$. The proof can be easily extended to the equivariant version of GradMetaNet, $\mathbf{F} : \mathbf{\Gamma}_b[f] \to \Theta[f']$. Let the underlying MLP used to define the spaces $\mathbf{\Gamma}_b[f]$ have an input dimension $d_0 = 1$, a single hidden layer of dimension $d_1 = n$, and an output dimension $d_2 = 1$. In this case, $\mathbf{\Gamma}_b[f]$, along with the action of $G_b$, is isomorphic to the space of "sets of sets" defined in Hartford et al. [31]. Moreover, GradMetaNet is equivalent to the architecture proposed in Hartford et al. [31]. This architecture is known to be unable to distinguish between certain highly regular, non-equivalent inputs. For example, incidence matrices of graphs can be viewed as sets of sets and, therefore, as elements of $\mathbf{\Gamma}_b[f]$. It was shown in Albooyeh et al. [2] that the ability of the architecture in Hartford et al. [31], and consequently GradMetaNet, to separate graphs based on their incidence matrices is equivalent to the 1-WL test [89].

Consider an example where $\mathbf{g}_1$ represents the incidence matrix of a graph consisting of two disconnected cycles of length 3, and $\mathbf{g}_2$ represents the incidence matrix of a single cycle of length 6. For any GradMetaNet model $\mathbf{F}$, it will hold that $\mathbf{F}(\mathbf{g}_1) = \mathbf{F}(\mathbf{g}_2)$. A straightforward check shows that for this example, $\mathbf{g}_1, \mathbf{g}_2 \in \mathcal{E}$, completing the proof. $\square$

# F   Experimental Details

In this section we provide all experimental details for all experiments in Section 7. We run all the experiments on a singel NVIDIA-A100-SXM4 GPU with 40GB of memory.

## F.1   Curvature Information Estimation

**Dataset.**   To construct the dataset for this experiment, we first generate 3000 SIREN models [78], commonly used for INRs. Each model has three layers with 32 hidden features, i.e. $1 \to 32 \to 32 \to 1$. The weights and biases of these models are randomly initialized using a standard Gaussian distribution. Each data point in the resulting dataset corresponds to a single SIREN model and consists of an input gradient set of size 128, $\mathbf{g} \in \Gamma_{128}[2]$, and a target vector, $\boldsymbol{\theta}_{\mathrm{FIM}} \in \Theta$. The input gradient set corresponding to a SIREN model $\boldsymbol{f_\theta}$ is computed by sampling 128 random points $S_{\boldsymbol{f_\theta}} = \{x_1, \dots, x_{128}\} \subset [-1, 1]$. To simulate diverse datasets, the points in $S_{\boldsymbol{f_\theta}}$ are sampled from a distribution $X_p$ defined by

$$X_p = \begin{cases} \mathrm{U}([0,1]), & \text{with probability } p, \\ \mathrm{U}([-1,0]), & \text{with probability } 1-p. \end{cases} \tag{98}$$

Here, the value of $p$ is randomly and uniformly selected over the unit interval. The set of input gradients is then given by $\{\nabla_1, \dots, \nabla_{128}\}$, $\nabla_i = \nabla \boldsymbol{f_\theta}(x_i)$. Thus the input vector $\mathbf{g} \in \Gamma_{128}[2]$ is given by $[\nabla_1, \dots, \nabla_{128}]$.

To compute the target vector $\boldsymbol{\theta}_{\mathrm{FIM}} \in \Theta$ , we first randomly sample 1024 random points from $X_p$ $S'_{\boldsymbol{f_\theta}} = \{x'_1, \dots, x'_{1024}\} \subset [-1, 1]$ (Note that the sets $S_{\boldsymbol{f_\theta}}$ and $S'_{\boldsymbol{f_\theta}}$ are independent). We then compute the true FIM (see Section A.3) by

$$\boldsymbol{F} = \frac{1}{1024} \sum_{i=1}^{1024} \nabla \boldsymbol{f_\theta}(x'_i) \nabla \boldsymbol{f_\theta}(x'_i)^\top . \in \Theta \otimes \Theta \tag{99}$$

The target vector $\boldsymbol{\theta}_{\mathrm{FIM}} \in \Theta$ is then given by $\boldsymbol{\theta}_{\mathrm{FIM}} = \mathrm{diag} \boldsymbol{F}$. The task is then to predict $\boldsymbol{\theta}_{\mathrm{FIM}}$ based on the gradient set vector $\mathbf{g}$ and is trained using $l_2$ loss. For baselines which process data in parameter space $\Theta[f]$ (rather than gradient set space $\Gamma_{128}[2]$), the gradient vector $\mathbf{g}$ computed from gradient set $\{\nabla_i\}$ is replaced by representing each gradient in parameter space $\nabla_i \in \Theta$, and then either concatenating them resulting in a vector $\boldsymbol{\theta} \in \Theta[128]$ or averaging them, resulting in a vector in $\boldsymbol{\theta} \in \Theta$.

**Baselines.**   We compare GradMetaNet and GradMetaNet++ against multiple baselines.

**MLP + concat:** This baseline takes the gradient vector $\mathbf{g}$ described above as input, flattens it, and feeds it into a standard MLP.

**Neuron Asymmetric GradMetaNet:** This variant respects the batch symmetries but not the gradient space symmetries of $\Gamma_b[f]$. It uses $\mathbf{g}$ as input and applies a Deep Sets architecture, as described in Zaheer et al. [91]. Specifically, the vector feature for element $i$ in the set is constructed as $[\mathbf{g}_{i,:,:}^{(0)}, \dots, \mathbf{g}_{i,:,:}^{(L)}]$.

**Batch Asymmetric GradMetaNet:** This variant closely resembles GradMetaNet but respects the gradient space symmetries while disregarding the batch symmetries of $\Gamma_b[f]$. It takes $\mathbf{g}$ as input and applies a GradMetaNet model defined over $\Gamma_1[256]$, where the batch axis is "flattened", resulting in a single gradient with a feature dimension of $256 = 2 \cdot 128$.

**Weight Space Models:** The variants "DWS+Concat", "DWS+Average", "GMN+Concat", and "GMN+average" take as input the gradients represented as $\boldsymbol{\theta} \in \Theta[f]$. Here, $\boldsymbol{\theta}$ is either the average of the gradients in the set, in which case $f = 1$, or the concatenation of all gradients in the set, in which case $f = 128$. These variants then apply either the deep weight space (DWS) architecture described in Navon et al. [62] or the graph metanetwork (GMN) architecture introduced in Lim et al. [48] to the corresponding input vector.

**Standard Estimator:** Finally, to highlight the benefits of learning to approximate algorithms rather than relying on fixed, non-learnable approximations, we compare GradMetaNet and GradMetaNet++ to a standard estimation of the target. Specifically, we compute $\mathrm{diag} \boldsymbol{F}_{\mathcal{B}_1}$ based on the 128 gradients

Table 5: Comparison of baseline properties.

| Baseline | Supports sets of gradients | Supports efficient gradient representation | $S_b$-invariant | $G$-equivariant |
|---|---|---|---|---|
| MLP + Concat | ✓ | ✓ | ✗ | ✗ |
| DWS + Concat | ✓ | ✓ | ✗ | ✓ |
| GMN + Concat | ✓ | ✓ | ✗ | ✓ |
| DWS + Average | ✗ | ✗ | – | ✓ |
| GMN + Average | ✗ | ✗ | – | ✓ |
| Batch Asymmetric GradMetaNet | ✓ | ✓ | ✗ | ✓ |
| Neuron Asymmetric GradMetaNet | ✓ | ✓ | ✓ | ✗ |
| GradMetaNet | ✓ | ✓ | ✓ | ✓ |
| GradMetaNet++ | ✓ | ✓ | ✓ | ✓ |

provided as input and evaluate its $\ell_2$ distance to the target, which, as a reminder, is $\mathrm{diag}\boldsymbol{F}_{\mathcal{B}_2}$. Here, $\mathcal{B}_2$ is computed using the gradients of a larger set of 1024 points, independently generated from the data points in $\mathcal{B}_1$.

**Data preparation.** We use 500 examples as a test dataset and 500 examples as a validation dataset, with the size of the training set varying between 10 and 2000 examples. As a preprocessing step, all data, including the target vectors, is normalized based on the statistics of the training dataset. The following are the three normalization methods used, based on the vector space to which the data belongs:

1. For input vectors of the form $\mathbf{g} \in \boldsymbol{\Gamma}_{128}[2]$: for each layer $l = 0, \ldots, 3$, we compute the mean $\mu^{(l)}$ and standard deviation $\sigma^{(l)}$ over the set of values:
$$\{\mathbf{g}_{i,j,k}^{(l)} \mid i = 0, \ldots, b;\; j = 1, \ldots, d_l;\; k = 0, 1;\; \mathbf{g} \in \mathcal{D}_{\text{train}}\}.$$
   We then normalize the data by: $\mathbf{g}^{(l)} \leftarrow \frac{\mathbf{g}^{(l)} - \mu^{(l)}}{\sigma^{(l)}}$.

2. For input vectors of the form $\boldsymbol{\theta} \in \boldsymbol{\Theta}[f]$ $\boldsymbol{\theta} = (\boldsymbol{W}^{(1)}, \boldsymbol{b}^{(1)}, \boldsymbol{W}^{(L)}, \boldsymbol{b}^{(L)},)$ (where $f = 1$ for averaging or $f = 128$ for concatenation): we follow the normalization scheme suggested in Navon et al. [61], where for each layer $l = 1, \ldots, 3$, we compute the means $\mu_w^{(l)}, \mu_b^{(l)}$ and standard deviations $\sigma_w^{(l)}, \sigma_b^{(l)}$ over the sets of values:
$$\{\boldsymbol{W}_{i,j,k}^{(l)} \mid i = 1, \ldots, d_{l-1};\; j = 1, \ldots, d_l;\; k = 0, \ldots, f;\; \boldsymbol{W}^{(l)} \in \mathcal{D}_{\text{train}}\}.$$
$$\{\boldsymbol{b}_{i,k}^{(l)} \mid i = 1, \ldots, d_l;\; k = 0, \ldots, f;\; \boldsymbol{b}^{(l)} \in \mathcal{D}_{\text{train}}\}.$$
   We then normalize the data as follows:
$$\boldsymbol{W}^{(l)} \leftarrow \frac{\boldsymbol{W}^{(l)} - \mu_w^{(l)}}{\sigma_w^{(l)}}, \quad \boldsymbol{b}^{(l)} \leftarrow \frac{\boldsymbol{b}^{(l)} - \mu_b^{(l)}}{\sigma_b^{(l)}}.$$

3. For target vectors $\boldsymbol{\theta}_{\text{FIM}} \in \boldsymbol{\Theta}$: recall that $\boldsymbol{\Theta} \cong \mathbb{R}^P$ for some integer $P \in \mathbb{N}$. Let $\boldsymbol{\theta}_{\text{FIM}}(i)$ denote the $i$-th entry of the vector $\boldsymbol{\theta}_{\text{FIM}} \in \mathbb{R}^P$. We compute a single mean $\mu$ and a single standard deviation $\sigma$ over the set of values: $\{\boldsymbol{\theta}_{\text{FIM}}(i) \mid i = 1, \ldots, P;\; \boldsymbol{\theta}_{\text{FIM}} \in \mathcal{D}_{\text{train}}\}$. The data is then normalized as: $\boldsymbol{\theta}_{\text{FIM}} \leftarrow \frac{\boldsymbol{\theta}_{\text{FIM}} - \mu}{\sigma}$.

**Additional experimental details.** Following the experimental setup in Navon et al. [61], all learned baselines in this experiment consist of approximately 15K learned parameters and roughly 3 layers (In some cases, where the input dimensionality was extremely large, it was not possible to maintain the 15K parameter budget with 3 layers). All models were trained for 100 epochs using the Adam optimizer with a learning rate of $1 \times 10^{-3}$ and a batch size of 32. We report the test loss corresponding to the best validation performance and repeat the experiment with random seeds 1 through 5.

### F.2 Learned Optimizers

**Learned optimization tasks.** The following are descriptions of the optimization tasks the learned optimizers were evaluated on. Across all tasks, the training loss is negative log-likelihood, and the batch size is 128 except for the transformers task.

Table 6: Multiplicative improvement in the number of steps needed to reach `Adam` 's best test loss, as well as the best test loss achieved by each `standard` and `GradMetaNet-based` learned optimizer. For each task, we run Adam, record its best test loss $L$ and the number of steps $N$ required to reach it. We then run each learned optimizer, measure how many steps it takes to reach $L$, and report the improvement relative to $N$. The "Step Reduction Factor vs. Adam" column thus indicates a multiplicative speedup in optimization steps. The "Best Test NLL" column record the best test loss achieved by the optimizer in the run. Results are averaged across 5 optimization runs.

| Dataset | Optimizer | Step reduction factor vs. Adam ($\uparrow$) | Best test NLL ($\downarrow$) |
|---|---|---|---|
| F-MNIST | Adam | $1.00 \pm 0.00$ | $0.475 \pm 0.011$ |
| | SGDM | $1.13 \pm 0.17$x | $0.463 \pm 0.019$ |
| | DS | $1.27 \pm 0.36$x | $0.460 \pm 0.011$ |
| | UNF | $1.16 \pm 0.18$x | $0.451 \pm 0.015$ |
| | DS + GradMetaNet | $1.44 \pm 0.43$x | $\mathbf{0.445 \pm 0.017}$ |
| | UNF + GradMetaNet | $\mathbf{1.51 \pm 0.59}$x | $0.447 \pm 0.011$ |
| CIFAR10 | Adam | $1.00 \pm 0.00$x | $1.616 \pm 0.039$ |
| | SGDM | $1.41 \pm 0.52$x | $1.556 \pm 0.014$ |
| | DS | $2.32 \pm 0.42$x | $1.494 \pm 0.015$ |
| | UNF | $2.64 \pm 0.53$x | $1.516 \pm 0.020$ |
| | DS + GradMetaNet | $\mathbf{4.63 \pm 1.11}$x | $1.427 \pm 0.018$ |
| | UNF + GradMetaNet | $4.26 \pm 0.56$x | $\mathbf{1.418 \pm 0.022}$ |
| CIFAR100 | Adam | $1.00 \pm 0.00$x | $3.449 \pm 0.021$ |
| | SGDM | $1.06 \pm 0.07$x | $3.424 \pm 0.026$ |
| | DS | $1.79 \pm 0.40$x | $3.356 \pm 0.027$ |
| | UNF | $1.58 \pm 0.27$x | $3.345 \pm 0.036$ |
| | DS + GradMetaNet | $\mathbf{3.15 \pm 0.56}$x | $\mathbf{3.253 \pm 0.019}$ |
| | UNF + GradMetaNet | $2.85 \pm 0.38$x | $3.262 \pm 0.013$ |
| LM1B | Adam | $1.00 \pm 0.00$x | $6.904 \pm 0.075$ |
| | SGDM | $1.01 \pm 0.03$x | $7.045 \pm 0.031$ |
| | DS | $0.88 \pm 0.14$x | $6.987 \pm 0.108$ |
| | UNF | $1.48 \pm 0.16$x | $6.702 \pm 0.080$ |
| | DS + GradMetaNet | $1.09 \pm 0.16$x | $6.993 \pm 0.076$ |
| | UNF + GradMetaNet | $\mathbf{1.82 \pm 0.39}$x | $\mathbf{6.557 \pm 0.061}$ |

(1) **MLP on FashionMNIST.** Learned optimizers are tasked with training a three-layer MLP classifier on a downsized ($8 \times 8$) flattened version of FashionMNIST [90]. The MLP has a single hidden layer of dimension 32 and ReLU activations.

(2) **MLP on CIFAR10.** Learned optimizers are tasked with training a three-layer MLP classifier on a downsized ($8 \times 8$) flattened version of CIFAR10. The MLP has a single hidden layer of dimension 32 and ReLU activations.

(3) **MLP on CIFAR100.** Learned optimizers are tasked with training a three-layer MLP classifier on a downsized ($8 \times 8$) flattened version of CIFAR100. The MLP has a single hidden layer of dimension 128 and ReLU activations.

(4) **Transformer on LM1B.** Learned optimizers are tasked with training a transformer language model on LM1B [14], using next token prediction. The transformer comprises two blocks with an embedding dimension of 32 and uses four self-attention heads. We train with a batch size of 8 on length 8 sequences.

(5) **2-parameter linear regression.** Learned optimizers are tasked with training a linear regression model with two inputs and a single output. The input training data is a mixture of two Gaussians centered around $(1, 2)$ and $(2, 1)$ and the target is always 0. This results in a loss landscape with non-standard curvature, as depicted in Figure 8.

**Learned optimizer architectural details.** In each inner training iteration, all learned optimizers are provided with the current parameters $\boldsymbol{\theta}_t$, current gradient $\nabla_t$, six momentum value $\{\boldsymbol{v}_t^{0.1}, \boldsymbol{v}_t^{0.5}, \boldsymbol{v}_t^{0.9}, \boldsymbol{v}_t^{0.99}, \boldsymbol{v}_t^{0.999}, \boldsymbol{v}_t^{0.9999}\}$, iteration number $t$ as an 11-dimensional sinusoidal encoding. All of these inputs are concatenated across the feature dimension to get a 19-dimensional vector per-parameter. I.e., the learned optimizers inputs are in $\Theta[19]$ and outputs are in $\Theta[1]$.

For GradMetaNet based learned optimizers, inputs also include current set of individual gradients $\mathbf{g} \in \mathbf{\Gamma}_b$ and six momentum value $\{\mathbf{v}_t^{0.1}, \mathbf{v}_t^{0.5}, \mathbf{v}_t^{0.9}, \mathbf{v}_t^{0.99}, \mathbf{v}_t^{0.999}, \mathbf{v}_t^{0.9999}\}$ concatenated across the feature dimension, resulting in an input in $\mathbf{\Gamma}_b[7]$. These inputs are processed by GradMetaNet, which outputs an embedding in $\mathbf{\Theta}[14]$, i.e., a 14-dimensional embedding vector per parameter. This embedding is concatenated to the other inputs of the learned optimizer, so the DeepSets/UNF based learned optimizer gets an input in $\mathbf{\Theta}[33]$.

The DeepSets [91] based learned optimizers process the inputs by applying a per-parameter MLP with three hidden layers of size 32. The UNF [92] based learned optimizers apply a per-parameter MLP with two hidden layers of size 32 and output dimensions of 32, followed by a single UNF layer. As a baseline compared against all learned optimizer,s we used a standard Adam optimizer [40] with learning rate tuned by grid-search over the values $\{0.0005 * j\}_{j=1}^{20}$ (i.e., 20 equidistant values in the range $[0.0005, 0.01]$).

**Meta-training details.** We meta-train for 50,000 steps using Adam [40] with learning rate $10^{-4}$, estimating meta-gradients over 16 parallel training runs using persistent evolutionary strategies (PES) [85] with a truncation length of 50. The meta-training objective is training loss at the end of the inner training horizon $T$, which is $T = 2,000$ for image classification tasks, $T = 5,000$ for the transformer language modeling task, and $T = 10$ for the 2D linear regression experiment. For all methods, we initialize $\alpha = 0.1$, $\mu = 0.9$ and $\beta = 0.001$ before meta-training. We use the learned optimizer meta-training setup from Metz et al. [57] (project available at `https://github.com/google/learned_optimization`, released under Apache License 2.0).

### F.3 INR Editing

**Dataset.** We utilized two previously proposed INR datasets: MNIST INRs [61] and CIFAR10 INRs [93]. Each INR represents an image from the original image datasets and consists of three layers with 32 hidden features. The target images are produced using the image processing library OpenCV [10], by dilating or increasing the contrast of the MNIST and CIFAR-10 images, respectively. To construct the input gradient set, $\mathbf{g} \in \mathbf{\Gamma}_{64}$, we sample 64 random input coordinates in $[0, 1]^2$ and compute the gradients of the editing loss w.r.t the original INR parameters. Specifically, if $\theta$ are the INR parameters and $\boldsymbol{I}_i : [0, 1]^2 \to \mathbb{R}^3$ is the target image, then for an input coordinates $(x_i, y_i)$, the gradient $\nabla_i$ is given by $\nabla_i = \nabla_{\boldsymbol{\theta}} \left( \boldsymbol{f}_{\boldsymbol{\theta}}(x_i, y_i) - \boldsymbol{I}_i(x_i, y_i) \right)^2$. We use the same set of random coordinates for all INRs.

**Combining GradMetaNet and weight-space methods.** We combine GradMetaNet and the weight-space architectures as follows: First, GradMetaNet processes the gradients $\mathbf{g} \in \mathbf{\Gamma}_{64}$ to produce outputs in $\mathbf{\Theta}[f]$. In this experiment, we choose $f = 8$. Next, the output is used as additional weight features for the weight-space network, i.e., the input to the weight-space network is in $\mathbf{\Theta}[9]$.

**Additional experimental details.** We train all methods for 150K steps using the AdamW [50] optimizer with a batch size of 64. We search over learning rates in $\{0.01, 0.005, 0.0001\}$. For ScaleGMN and GMN we use the official implementation provided in Kalogeropoulos et al. [38] with the same parameter configuration provided by the authors. We use the bidirectional variant (ScaleGMN-B, Kalogeropoulos et al. [38]) with 10 layers and a hidden dimension of 128. The DWS [61] network consists of 8 layers with 128 hidden features. Finally, GradMetaNet consists of 10 layers with 256 hidden features, while GradMetaNet++ uses 3 blocks with 8 heads and 64 hidden features.

## G Additional Experimental Results

In this section, we detail additional experimental results for GradMetaNet-based *learned optimizers* and curvature estimation experiments. The experimental setup, datasets, and baselines are all identical to the setting of Section 7 and Appendix F.

### G.1 Scaling Curvature Estimation

We extend the curvature-estimation experiment from Section 7.1 to models with over one million parameters. As in the main text, networks are trained to predict the trace of the Fisher Information Matrix, $\text{tr}(\boldsymbol{F}_{\boldsymbol{\theta}})$, using the same decomposed-gradient representation. At this scale, several original baselines, especially full-gradient methods, are computationally infeasible in memory or runtime. We therefore compare against a scalable baseline: an MLP that operates on decomposed gradients.

GradMetaNet attains substantially lower normalized test MAE than the MLP baseline (lower is better), indicating more accurate curvature estimation at scale.

Table 7: Large-scale curvature estimation (1M+ parameters). Normalized test MAE (↓).

| Model | Normalized Test MAE (↓) |
|---|---|
| MLP | 0.779 |
| GradMetaNet | **0.413** |

## G.2 Additional Learned Optimizer Speedup Results

**Step reduction details.** In Table 6, we add to Table 1 the standard deviation of the average step reduction factor for each optimizer, as well as the best test loss achieved by each optimizer in the training horizon (2,000 for image classification tasks and 5,000 for the LM task). For the reader's convenience, in Figure 9 we presented a larger version of the plots in Figure 6.

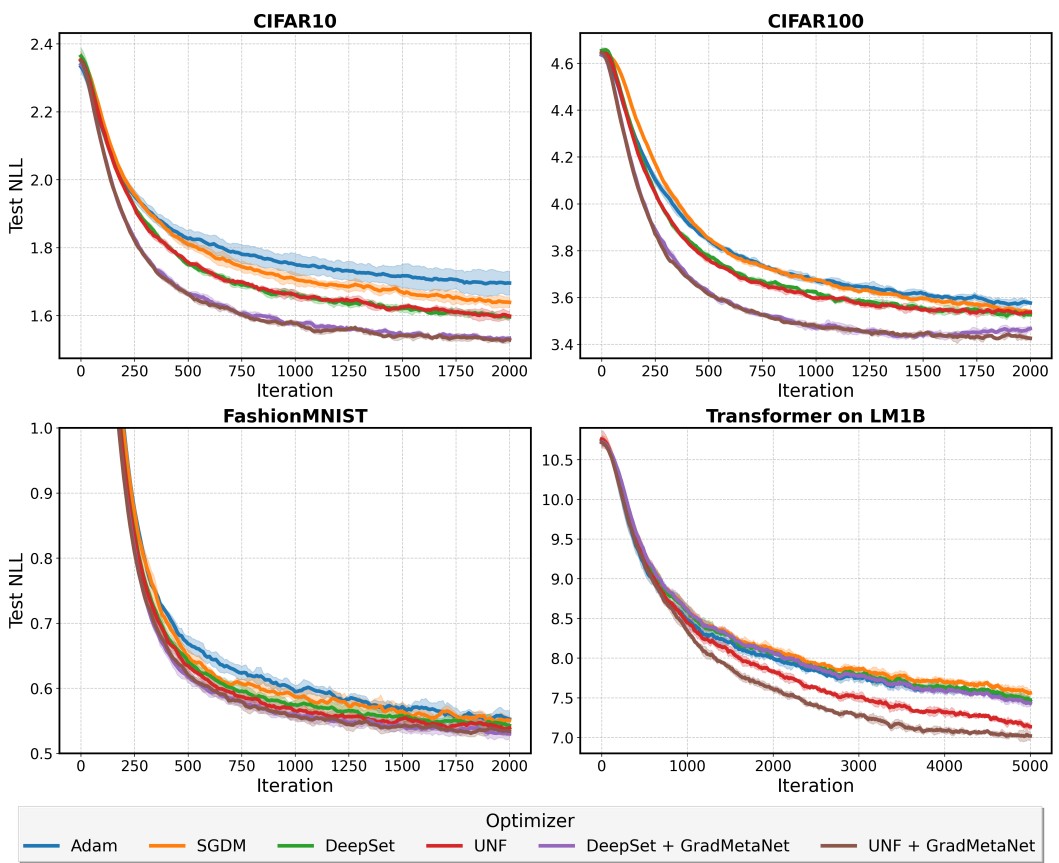

Figure 9: Test loss curves for MLP image classification tasks and a transformer language model trained on LM1B, using different optimizers and (learning rate tuned) Adam. Curves are smoothed and averaged over 5 random initializations, with shaded regions representing standard deviation.

Table 8: Training iteration speed (in iterations per second) for Adam, UNF, and UNF + GradMetaNet on LM1B (transformer) and CIFAR100 (MLP).

| Optimizer | Transformer on LM1B (It/s) | MLP on CIFAR100 (It/s) |
|---|---|---|
| Adam | $107.55 \pm 8.09$ | $359.54 \pm 5.67$ |
| UNF | $75.45 \pm 2.32$ | $304.60 \pm 12.45$ |
| UNF + GradMetaNet | $65.61 \pm 0.31$ | $264.33 \pm 4.94$ |

**Train-time comparison.** We measured the time per training iteration (in iterations per second) for Adam, UNF, and UNF + GradMetaNet on LM1B and CIFAR100. The results are reported in Table 8. All models and optimizers run on a single NVIDIA A100-SXM4-40GB GPU. As expected, learned optimizers introduce some computational overhead, and while GradMetaNet-based learned optimizers incur a slight per-step slowdown, the significant reduction in steps leads to GradMetaNet having a substantial speedup in train-time (up to $3\times$ faster than Adam). As reported in Table 9, GradMetaNet-based optimizers outperform all baselines in total training speed.

Table 9: Step reduction and train-time speedup relative to Adam across datasets.

| Dataset | Optimizer | Step reduction factor vs. Adam (↑) | Train-time speedup vs. Adam (↑) |
|---|---|---|---|
| Fashion MNIST | UNF | 1.16x | 1.05x |
| | GradMetaNet + UNF | 1.51x | 1.13x |
| CIFAR10 | UNF | 2.64x | 2.31x |
| | GradMetaNet + UNF | 4.26x | 3.13x |
| CIFAR100 | UNF | 1.58x | 1.34x |
| | GradMetaNet + UNF | 2.85x | 2.10x |
| LM1B | UNF | 1.48x | 1.04x |
| | GradMetaNet + UNF | 1.82x | 1.11x |

### G.3 Generalization to New Tasks and Model Sizes

In this section, we demonstrate that GradMetaNet-based learned optimizers generalize across base model sizes and across tasks *without* re-meta-training. This contrasts with other weight-space learned optimizers such as DWS [61], NFN [94], and UNF [92], which require defining a different weight-space metanetwork to process gradients from different base architectures. All metrics are reported relative to Adam under identical training settings.

**Model-size generalization.** For CIFAR-10 and CIFAR-100, we meta-train a learned optimizer on a specific MLP width and meta-test on a larger, previously unseen MLP: (i) CIFAR-10: meta-train on a 32-hidden-dim MLP and meta-test on 64 hidden dims; (ii) CIFAR-100: meta-train on 128 hidden dims and meta-test on 256 hidden dims. In both cases, the optimizer transfers successfully to the larger model, improving both the number of steps and wall-clock time to reach the same target.

Table 10: Model-size generalization: meta-train on a smaller MLP and meta-test on a larger MLP.

| Dataset (train width → test width) | Optimizer | Step reduction factor vs. Adam (↑) | Train-time speedup vs. Adam (↑) |
|---|---|---|---|
| CIFAR-10 (32 → 64) | DS | 2.18x | 1.85x |
| | GradMetaNet + DS | 4.15x | 3.05x |
| CIFAR-100 (128 → 256) | DS | 1.78x | 1.59x |
| | GradMetaNet + DS | 3.04x | 2.37x |

**Task generalization.** We also evaluate cross-task transfer by meta-training on CIFAR-10 and meta-testing on CIFAR-100, and vice versa. Because the output dimensionality differs, this setting additionally exercises a mild form of architecture/size transfer. GradMetaNet-based optimizers generalize in both directions.

Table 11: Task generalization across CIFAR tasks.

| Meta-train → Meta-test | Optimizer | Step reduction factor vs. Adam (↑) | Train-time speedup vs. Adam (↑) |
|---|---|---|---|
| CIFAR-100 → CIFAR-10 | DS | 2.45x | 1.72x |
| | GradMetaNet + DS | 3.57x | 2.83x |
| CIFAR-10 → CIFAR-100 | DS | 1.93x | 1.57x |
| | GradMetaNet + DS | 3.80x | 2.59x |

### G.4 Scaling to Larger Models

**Context.** Scaling learned optimizers is challenging due to extreme compute demands (e.g., VeLO [58] reportedly required ∼4,000 TPU-months to meta-train). Training at that scale is currently infeasible with our academic resources, irrespective of whether GradMetaNet is used as the architectural backbone. Nevertheless, to show the feasibility of scaling GradMetaNet-based learned optimizers to larger models and settings, we measure the update cost of a GradMetaNet-based optimizer for a GPT-2-scale model.

**Large-model update cost.** We measure the *update time* and *memory footprint* of a GradMetaNet-based optimizer when applied to the gradients of a GPT-2-scale model: a 12-layer transformer with 12 attention heads per-layer, hidden size of 768, totaling ∼117M parameters. On two NVIDIA A100-SXM4-40GB GPUs, a GradMetaNet update (including backpropagating through the base transformer) takes 1.29s versus 0.89s for Adam, with a similar memory footprint.

Table 12: Update-time and memory comparison on a GPT-2 scale transformer using 2×A100-40GB.

| Optimizer | Update Time (s) (↓) | Memory Footprint (↓) |
|---|---|---|
| Adam | 0.89 | 77.2 GB |
| GradMetaNet | 1.29 | ∼77 GB (similar to Adam) |

