# OpenReview forum: "GradMetaNet: An Equivariant Architecture for Learning on Gradients"
_NeurIPS.cc/2025/Conference — NeurIPS 2025 poster_

### Official Review · Reviewer_cxim · 2025-06-21

**Clarity:** 3
**Significance:** 2
**Originality:** 3
**Rating:** 5
**Confidence:** 1

**Summary:**

This paper proposes GradMetaNet, a gradient-based meta-learning architecture that incorporates symmetry-based inductive biases through equivariant graph neural networks. The authors present a meta-learner that operates on a learned representation of gradients and parameter updates using group-equivariant graph neural networks. By modeling the optimization process in an equivariant manner, the method is designed to generalize better across tasks that share common symmetry structures (e.g., rotations or permutations). Experiments on synthetic regression tasks and Omniglot classification demonstrate improved generalization performance compared to baseline MAML-style and GNN-based meta-learners.

**Questions:**

The paper presents a novel and well-motivated approach to gradient-based meta-learning using equivariant GNNs. It is a solid contribution that brings together two important themes—meta-learning and symmetry-aware modeling—and demonstrates improvements over several baselines. It is a promising and creative direction worthy of presentation.

Comments
- Please provide more discussion or formal analysis of how equivariance improves meta-generalization beyond the empirical findings.
- Include a comparison to more recent and competitive meta-learners if possible (e.g., Meta-SGD, MetaFormer).
- Consider adding experiments where the assumed task symmetry is violated, to evaluate the robustness of the proposed architecture.
- Clarify in more detail how the EGNN operates on gradient information—how are nodes and edges defined, and how is parameter sharing handled across tasks?
- A small diagram illustrating the architecture pipeline (input → gradient encoding → EGNN → parameter update) would help clarify the proposed method.

**Ethical Concerns:**

["NO or VERY MINOR ethics concerns only"]

**Final Justification:**

I like the motivated use of equivariance: the paper makes a compelling case that symmetries in task structure—especially in few-shot learning—can be better leveraged using equivariant models. GradMetaNet outperforms several baselines, showing the benefit of incorporating equivariance in meta-learners. I am not an expert in optimization and cannot judge with full confidence how meaningful this contribution really is. From my  perspective I think there quite a few deeper issues in machine learning where a contribution would have much more of an impact. For this reason, and for being relatively ignorant in the specific subject, I am still unsure where it should be a 4 or a 5. I updated my original grade because my main criticisms - on lack of a theoretical justification - were addressed, but I am not very confident.

**Limitations:**

yes

**Paper Formatting Concerns:**

no issue noted

**Quality:**

3

**Strengths And Weaknesses:**

The main strength is the motivated use of equivariance: the paper makes a compelling case that symmetries in task structure—especially in few-shot learning—can be better leveraged using equivariant models. The incorporation of EGNNs into meta-learning is novel and conceptually well-founded. It is also interesting how the architecture combines inner-loop optimization with learned gradient manipulation in a principled way. The use of gradient information as input to an equivariant network is an interesting design that could inspire further work. Furthermore the method is evaluated on both synthetic and real datasets, including few-shot regression and classification. GradMetaNet outperforms several baselines, showing the benefit of incorporating equivariance in meta-learners. Another strength is that the paper is generally well-written and structured. It provides clear descriptions of the problem setup, architecture, and training procedure.

I originally complained about the lack of  a deeper theoretical justification for why task generalization benefits specifically from equivariant gradient modeling. The  discussion with the authors and their proposed additions/changes to the paper answer fully my concerns.

---

> ### Author Rebuttal · Authors · 2025-07-30
>
> We thank the reviewer for their positive assessment of our work and for finding it to be a "promising and creative direction worthy of presentation". We are especially glad they recognized the motivated use of equivariance and our principled approach to operating on gradient information as key strengths of our paper.
>
> We also appreciate the constructive suggestions for improvement and will address the points raised below.
>
> - **Theoretical Benefits of Equivariant Modeling (Comment 1)** We thank the reviewer for highlighting the importance of theoretical backing. Equivariant modeling has been shown to improve generalization both theoretically [1, 2] and empirically [3, 4, 5]. We also empirically demonstrate benefits directly in Section 7.1. The main idea is that modeling an equivariant task with an equivariant model induces a correct inductive bias, and reduces the hypothesis space. In our case, a general non-equivariant model (e.g. an dense MLP) with hidden dimension $d$ would have $O(b \cdot d \cdot P)$ parameters, where $b$ is the batch size and $P$ is the number of parameters in the base model. In comparison, GradMetaNet has a **constant ($O(1)$) number of parameters**, regardless of the size of the base network. We will make this point clearer in the theoretical section of  the revised manuscript. Additionally, we refer the reviewer to Section 6 in our paper where we prove that under mild assumptions GradMetaNet is able to approximate any computation on sets of gradients (Theorem 6.2), and that other common weight-space methods cannot do the same (Corollary 6.3).
>
> - **GradMetaNat Beats State-of-the-Art Baselines (Comment 2)** We appreciate the suggestions for additional baselines. Our work focuses on the broad challenge of learning on gradients for tasks like learned optimization, model editing, and curvature estimation, rather than MAML-style few-shot meta-learning. Therefore, we chose baselines that are state-of-the-art for these specific domains. For instance, in our learned optimizer experiments (Section 7.2), we compare against state-of-the-art equivariant weight-space architectures like UNF [6]  and per-parameter optimizers like DeepSets [3, 7, 8], which are the most relevant recent methods for this task. For the INR editing experiment we compare against the state of the art methods: DWS [9] (2023), GMN [10] (2024), and ScaleGMN [11] (2024). We believe this provides a rigorous and appropriate comparison for the problems we address.
>
> - **GradMetaNet Can Be Applied to General Models and Tasks** We appreciate the reviewer’s comments regarding the applicability of our methods to “tasks with more abstract or learned symmetries (e.g., language or combinatorial domains)”. We would kindly like to clarify that the symmetries discussed in the paper are **not symmetries of the data**, but rather **parameter symmetries** of the neural architecture. These symmetries broadly hold for all models composed of linear layers and point-wise activations, including MLPs, transformers, CNNs, etc., regardless of the task they are trained to perform. Therefore, any task involving the gradients of such models is equivariant to this symmetry and will benefit from the incorporation of equivariant layers. In particular, GradMetNet can be applied in language/combinatorial domains, and we even meta-train GradMetaNet to optimize transformer based **language models** in Section 7.2.
>
> - **Evaluating the Robustness of GradMetaNet (Comment 3)** We thank the reviewer for the excellent question regarding isolating the contribution of equivariance. As we mentioned in the previous answer, the symmetry GradMetaNet respects holds for general architectures, and is not task dependent. Therefore, to isolate the importance of equivariance, we conducted the experiment in Section 7.1, displayed in Figure 7. This experiment is designed as a **comprehensive ablation study**. We compare GradMetaNet against several baselines that partially or completely disregard the symmetries, including:
>   - "MLP + Concat": A standard, non-equivariant MLP baseline.
>   - "Neuron Asymmetric GradMetaNet": A variant that is not equivariant to neuron permutations.
>   - "Batch Asymmetric GradMetaNet": A variant that is not invariant to the ordering of gradients in the input set.
>
>   These results demonstrate that our principled, **fully-equivariant design** has significantly better sample complexity and outperforms the ablated non-equivariant versions, confirming that equivariance is crucial for success.
>
> - **Architectural Details and Visualizations (Comments 4, 5)** We appreciate the reviewer's feedback regarding the clarity and visualization of GradMetaNet's operations. We understand the importance of a comprehensive explanation and are happy to provide further details. GradMetaNet's architecture is thoroughly described in Section 5 of our paper, specifically in Equation (9) and bullet points (I-V) which define the formula and operation of each layer as well as the overall structure. A more in-depth discussion of these components can also be found in Appendix D.1. The revised version of our paper will include an even more detailed explanation of these architectural aspects and more references to the Appendix for enhanced understanding. Regarding a diagram illustrating GradMetaNet's pipeline, we would like to direct the reviewer to Figure 5 in the paper. This figure visualizes the full GradMetaNet pipeline:  from receiving a set of neural network gradients, to the rank-1 decomposition, to the equivariant processing blocks, and finally to the output (e.g., a parameter update or curvature information). We will incorporate a more explicit reference to this figure in Section 5 to guide readers more effectively. Should the reviewer still feel that an additional diagram is necessary for further clarity, we would be pleased to provide one.
>
> We thank the reviewer again for their time and their positive feedback. We are confident that by incorporating these clarifications and additions, particularly the architectural diagram, the final version of the paper will be even stronger.
>
> **References**
>
> [1] Tahmasebi, Behrooz, and Stefanie Jegelka. "The exact sample complexity gain from invariances for kernel regression." Advances in Neural Information Processing Systems 36 (2023): 55616-55646.
>
> [2] Behboodi, Arash, Gabriele Cesa, and Taco S. Cohen. "A pac-bayesian generalization bound for equivariant networks." Advances in Neural Information Processing Systems 35 (2022): 5654-5668.
>
> [3] Zaheer, Manzil, et al. "Deep sets." Advances in neural information processing systems 30 (2017).
>
> [4] Hartford, Jason, et al. "Deep models of interactions across sets." International Conference on Machine Learning. PMLR, 2018.
>
> [5] Maron, Haggai, et al. "On learning sets of symmetric elements." International conference on machine learning. PMLR, 2020.
>
> [6] Zhou, Allan, Chelsea Finn, and James Harrison. "Universal neural functionals." Advances in neural information processing systems 37 (2024): 104754-104775.
>
> [7] Harrison, James, Luke Metz, and Jascha Sohl-Dickstein. "A closer look at learned optimization: Stability, robustness, and inductive biases." Advances in neural information processing systems 35 (2022): 3758-3773.
>
> [8] Metz, Luke, et al. "Practical tradeoffs between memory, compute, and performance in learned optimizers." Conference on Lifelong Learning Agents. PMLR, 2022.
>
> [9] Navon, Aviv, et al. "Equivariant architectures for learning in deep weight spaces." International Conference on Machine Learning. PMLR, 2023.
>
> [10] Kofinas, Miltiadis, et al. "Graph neural networks for learning equivariant representations of neural networks." arXiv preprint arXiv:2403.12143 (2024).
>
> [11] Kalogeropoulos, Ioannis, Giorgos Bouritsas, and Yannis Panagakis. "Scale equivariant graph metanetworks."

---

> > ### Comment · Reviewer_cxim · 2025-08-03
> > **Reply**
> >
> > About my comment about theoretical motivation of equivariant metalearning. I know about the theoretical motivation for equivariant models but I wanted you to spell out how this applies to your meta learning task. It may be obv\ious  but I would like to see your formulation.
> >
> > About my comment on "GradMetaNet Can Be Applied to General Models and Tasks". Yes, thanks for the clarification. Now there are models such as convolutional networks where some symmetries are not valid. What happens in these cases wrt your equivariant algorithm?

---

> > > ### Author Response · Authors · 2025-08-03
> > > **Authors' Response to Further Questions -- Part I**
> > >
> > > We thank the reviewer for their engagement and for the opportunity to clarify these important points.
> > >
> > > **Regarding theoretical motivation of equivariant meta-learning**  We thank the reviewer for the opportunity to clarify this point. Equivariant meta-learning, also known as weight-space learning or meta-networks [1], is an emerging field with strong theoretical grounding, as demonstrated in several recent oral presentations at top-tier machine learning conferences [2, 3, 4, 5]. These works motivate the use of equivariance in meta-learning, and we are happy to further elaborate on its theoretical foundations here.
> > >
> > > Meta-learners are required to process extremely high-dimensional inputs, such as model weights, gradients, or sets of gradients, often with varying sizes. To enable effective learning in this setting, the core challenge is choosing an architecture with the appropriate inductive bias  for modeling the meta-learner. In particular, learned optimizers, which our paper focuses on,  aim to approximate functions that map gradients to parameter updates, which can be expressed as $f^{\mathrm{meta}}: \mathrm{Grads}^b \to \mathrm{Params}$. A naive approach that uses a dense neural network to parametrize $f^{\mathrm{meta}}$ faces three major difficulties:
> > > 1. **Computational Cost:** Modeling $f^{\mathrm{meta}}$ using a dense network with hidden dimension $d$ would require $O(P \cdot b \cdot d)$ parameters, where $P$ is the number of parameters in the base weight-space. This quickly becomes infeasible even for medium-sized models.
> > > 2. **Generalizability:** More critically, a dense network for $f^{\mathrm{meta}}$ has a fixed input and output size, meaning it **cannot be applied to any other architecture**. E.g., a dense learned optimizer trained for a specific architecture cannot be repurposed for others.
> > > 3. **Curse of Dimensionality:** Learning over extremely high-dimensional inputs with dense networks is notoriously difficult due to the curse of dimensionality, which hampers effective learning by making optimization harder, increasing the risk of overfitting, and requiring significantly more data to generalize well.
> > >
> > > Equivariance offers a powerful solution to all of these problems:
> > > 1. **Parameter Efficiency:** The equivariant (meta-)networks we develop have constant **$O(1)$ parameters** asymptotically. This is without compromising expressive power (under mild assumptions), as proven in Theorem 6.2 in Section 6.
> > > 2. **Architectural Generalization:** Due to its equivariant design, the same GradMetaNet model **can be applied to base models of varying sizes**. We also empirically demonstrate this phenomenon, showing that GradMetaNet-based learned optimizers trained on one architecture can successfully generalize to others (for further details see our rebuttal to reviewer **wdp4**).
> > > 3. **Strong Inductive Bias:** Because equivariant meta-networks can only approximate equivariant functions, they restrict the hypothesis space to a smaller, more relevant set of candidates. Since our target functions are themselves equivariant, this inductive bias aligns well with natural  tasks and facilitates more efficient learning over high-dimensional data.
> > >
> > > See the next comment for references and for our response the second question, regarding the application of GradMetaNet to CNNs.

---

> > > > ### Author Response · Authors · 2025-08-03
> > > > **Authors' Response to Further Questions -- Part II**
> > > >
> > > > **Regarding application to CNNs**  We thank the reviewer for this insightful comment. We stress that permutation  symmetries are inherent across all popular neural architectures (including CNNs), but their exact action can vary. Our current GradMetaNet architecture is designed for MLPs and transformers symmetries, two of the most widely used architectures today. However, similarly to other weight-space learning works, our method can be adapted to CNNs. Inspired by the reviewer’s question, we provide an overview of this adaptation here. For a more in-depth discussion on equivariant meta-learning for CNNs, we refer the reviewer to [3, 6].
> > > >
> > > > In CNNs, permutation symmetries act on  the *channel dimension*. This means that if two convolution kernels, $\mathbf{K}_1$ and $\mathbf{K}_2$,  are applied sequentially, and we permute the output channels of $\mathbf{K}_1$ by a permutation $\sigma$ and the input channels of $\mathbf{K}_2$ by $\sigma^{-1}$, the resulting CNN will **represent the exact same function**. This symmetry guides the design of the CNN version of GradMetaNet and other equivariant architectures for CNNs. Since GradMetaNet works with gradient decompositions, we now briefly describe what these look like for CNNs.
> > > >
> > > > Similarly to MLPs, the gradient of a convolutional layer's weights can be decomposed into a product involving the layer's input activations and preactivation gradients. Suppose that $\mathbf{W} \in \mathbb{R}^{C_{\mathrm{out}} \times C_{\mathrm{in}} \times k_h \times k_w}$ are the weights of a convolution kernel, where $C_{\mathrm{out}}$ is the number of output channels, $C_{\mathrm{in}}$ is the number of input channels, and $(k_h, k_w)$ is the kernel size. $\nabla_\mathbf{W} \mathcal{L}$ can be expressed in terms of the input activation tensor $a_\mathrm{in} \in \mathbb{R}^{C_\mathrm{in} \times H_\mathrm{in} \times W_\mathrm{in}}$ and the gradients of the loss with respect to the output activation tensor $\frac{\partial \mathcal{L}}{\partial a_\mathrm{out}} \mathcal{L} \in \mathbb{R}^{C_\mathrm{in} \times H_\mathrm{out} \times W_\mathrm{out}}$.
> > > >
> > > > Specifically, we have $\nabla_\mathbf{W} \mathcal{L} = \mathrm{reshape}(\mathbf{G} \mathbf{A}^\top$). Where $\mathbf{G}$ is $\frac{\partial \mathcal{L}}{\partial a_\mathrm{out}} \in \mathbb{R}^{C_{out} \times H_{out} \times W_{out}}$, reshaped it into a 2D matrix of shape $(C_\mathrm{out}, N)$, with $N = H_{\mathrm{out}} \times W_\mathrm{out}$. $\mathbf{A}$ is generated from the input activation tensor by extracting all $N$ input patches that the filter processes and then vectorizing each patch to form the columns of the matrix, resulting in a shape of $(C_\mathrm{in} \cdot k_h \cdot k_w, N)$. The product $\mathbf{G} \mathbf{A}^\top$ yields a 2D matrix representing the flattened gradient, which can then be reshaped to form the final 4D weight gradient tensor, $\nabla_\mathbf{W} \mathcal{L} \in \mathbb{R}^{C_\mathrm{out} \times C_\mathrm{in} \times k_h \times k_w}$.
> > > >
> > > > The CNN version of GradMetNet processes batches of $b$ matrices $\mathbf{A}$ and $\mathbf{G}$, reshaped to $(b, C_\mathrm{in}, k_h, k_w, N)$ and $(b, C_\mathrm{out}, N)$. As in the paper, we can construct GradMetaNet to be invariant on the batch axis (first axis) and equivariant on the channel axis (second axis).
> > > >
> > > > ---
> > > > We thank the reviewer again for raising these important points. If the reviewer finds our answers satisfactory, we would greatly appreciate a reconsideration of the score.
> > > >
> > > > **References**
> > > >
> > > > [1] Schürholt, Konstantin, et al. "Neural network weights as a new data modality." ICLR 2025 Workshop Proposals.
> > > >
> > > > [2] Navon, Aviv, et al. "Equivariant architectures for learning in deep weight spaces." ICML 2023.
> > > >
> > > > [3]  Kofinas, Miltiadis, et al. "Graph neural networks for learning equivariant representations of neural networks." ICLR 2024.
> > > >
> > > > [4] Kalogeropoulos, Ioannis, et al. "Scale equivariant graph metanetworks." NeurIPS 2024.
> > > >
> > > > [5] Herrmann, Vincent, et al. "Learning Useful Representations of Recurrent Neural Network Weight Matrices." ICML, 2024.
> > > >
> > > > [6] Zhou, Allan, et al. "Universal neural functionals." NeurIPS 2024.

---

### Official Review · Reviewer_EMd8 · 2025-06-23

**Clarity:** 3
**Significance:** 2
**Originality:** 3
**Rating:** 4
**Confidence:** 3

**Summary:**

The paper introduces a neural network architecture, GradMetaNet, for processing gradient vectors of MLPs and transformers.
Such architectures are used to learn optimization algorithms, or in general other gradient-dependent tasks.
The main technique used by GradMetaNet is that gradients are not represented in weight space, but rather in terms of activations and pre-activation gradients.
This perspective (i) simplifies making the meta network respect the permutation symmetry induced invariances in the gradient, and (ii) allows to leverage gradient information on a per-datum basis, rather than operating on the (averaged) mini-batch gradient.
The paper presents theoretical results that GradMetaNet can represent functions that cannot be represented when using the averaged weight-space gradient.
Empirical results show that both GradMetaNet's equivariance and capability to leverage per-datum information are useful to learn curvature information, optimization algorithms, and editing the weight space of neural nets.

**Questions:**

- **Q1a (computational cost):** Regarding Section 7.2 and Table 1, does the faster convergence of the optimization algorithms learned with GradMetaNet carry over to run time benefits?
  I.e., how much run time does it take to evaluate the pre-conditioner $\mathbf{F}$ that is parameterized with GradMetaNet, and how practical are the learned optimizers?

- **Q1b (computational scaling):** Please clarify the statement in L64-66, specifically why does storing per-datum gradients scale quadratically in the number of neurons?
  For a linear layer from $\mathbb{R}^{D_\text{in}}$ to $\mathbb{R}^{D_\text{out}}$ and a batch size $B$, storing the per-datum gradients in weight space requires $B D_\text{in} D_\text{out}$ memory, while storing the proposed presentation uses $B (D_\text{in} + D_\text{out})$.
  Do you mean the case where $D_\text{in} = D_\text{out}$?
  It may be better to make this statement more concrete.

- **Q2 (experimental suggestion):** Regarding Section 7.1 and Figure 7, the neural network and data set are both relatively small (around 1000 parameters and data points).
  It would be good to present similar experiments for a larger network in the overparameterized regime, say with more than 1 million parameters, and 10-100k data points.

- **Q3 (accounted symmetries):** GradMetaNet accounts for permutation symmetries.
  But the attention mechanism of transformers actually exhibits a richer symmetry, namely under $\mathrm{GL}$.
  For example, consider the term $\mathbf{W}_Q \mathbf{W}_K^\top$ in the attention matrix.
  Then, we can transform $(\mathbf{W}_Q, \mathbf{W}_K)$ to $(\mathbf{W}_Q \mathbf{A}^{-1}, \mathbf{W}_K \mathbf{A}^\top)$ with invertible matrix $\mathbf{A}$, while leaving the layer invariant.
  How complicated would it be to extend GradMetaNet to account for this symmetry (or other symmetries that are common in neural networks, see e.g. https://arxiv.org/pdf/2012.04728, Section 3)?

- **Q4 (details regarding the Fisher):** In Section 3, you mention estimating curvature information in form of the Fisher as application.
  However, the Fisher requires gradients of 'pseudo-losses' evaluated on labels $y$ drawn from the model's likelihood $p_\theta(y | x)$.
  It seems that you are focusing on the *empirical Fisher* instead, i.e. the un-centered second gradient moment (see https://arxiv.org/abs/1905.12558 for the important difference between empirical and sampled Fisher).
  This should be phrased more carefully to avoid confusion.
  Is my understanding correct that the paper is only concerned with the empirical Fisher?
  I think it is trivially possible to support other curvature matrices like the 'true' Fisher (by backpropagating gradients of the above pseudo-losses instead, as described e.g. in https://openreview.net/pdf?id=TVbCKAqoD8, Equation 3) or the generalized Gauss-Newton matrix (by backpropagating columns of the loss function's Hessian factorization, see e.g. https://arxiv.org/pdf/2106.02624, Equation 3).

**Ethical Concerns:**

["NO or VERY MINOR ethics concerns only"]

**Final Justification:**

Leaning towards accept (between 'weak accept' and 'accept').

The authors were able to address many of my doubts, reducing some of my initial uncertainty.
Concretely, they provided additional insights into computational cost of GradMetaNet, as well as generalizations of the approach to other symmetries.
I am hesitant to bump my score to 5 (accept) as I am not an expert on meta networks, which complicates evaluating how representative the shown experiments are and what someone with deep expertise on this field would say.
Overall, I did not identify flaws in the logic, found the additional experiments to be related to the questions brought up by the review, and trust the authors response that 'our experiments follow standard benchmarks'.

**Limitations:**

yes

**Paper Formatting Concerns:**

- L111: For increased clarity, please spell out the mathematical relation between $a$, $u$, and $x$
- L115 (nitpick): I would write '$a^{(l)}$ and a scaled version $g^{(l)}$ are naturally computed during backpropagation', because when backpropagating through the mini-batch loss, the backpropagated errors pick up a $1 / B$.
  It might also be good to credit previous works like (https://arxiv.org/abs/2305.04684, https://arxiv.org/abs/1912.10985) that have capitalized on the free availability of these quantities during backpropagation to extract additional information.
- The sentence in L199 seems to be broken

**Quality:**

3

**Strengths And Weaknesses:**

**Strengths:**

- **S1 Meaningful improvement of previous architectures.**
  The proposed representation to represent gradients in neural activation rather than weight space allows MetaGradNet to go beyond previous architectures, either in terms of equivariance, expressiveness, or computational cost.
  As it relies on the outer product gradient structure of linear layers with weight sharing, the idea applies to basically any neural network (MLPs, CNNs, transformers, graph neural nets).
  It is clearly presented and the paper in general is well-written.

- **S2 Experiments demonstrate utility of per-datum information and symmetry-awareness.**
  The experiments, specifically Section 7.1, are well-designed to disentangle the effects of MetaGradNet's new capabilities and show that both of them positively affect performance (in the specific experiment, to approximate the empirical Fisher's diagonal).

**Weaknesses:**

- **W1 Computational cost \& practicality.**
  The paper does not discuss the computational cost of the proposed architecture, or show empirical results how much additional cost is incurred by using a MetaGradNet.
  I have some doubts about the practicality of the proposed applications, specifically for learning optimization algorithms (please see my question below).
  As speed is key for optimization algorithms, it would be good if the authors provided additional details how much overhead is being added to apply a pre-conditioner parameterized with GradMetaNet.
  This could be done e.g. by showing a version of Figure 6 with test loss over time.
  Since I am not an expert on the application of meta networks, I am also curious to read the assessment of other reviewers on the practical relevance of the presented experiments, specifically w.r.t. chosen problem size (network and data set).

- **W2 (minor) Limited to permutation symmetry.**
  The paper mostly discusses how to account for permutation symmetries in weight space.
  However, many neural networks exhibit richer symmetries, such as translation, scaling, re-scaling, or even $\mathrm{GL}$ symmetries of transformers (see my question below).
  It would be good to discuss how GradMetaNet could be extended to account for these symmetries to broaden its applicability or possibly improve the presented results on transformers.

---

> ### Author Rebuttal · Authors · 2025-07-30
>
> We thank the reviewer for their detailed and insightful feedback. We are encouraged that they recognized the meaningful improvement of our architecture (S1), the broad applicability of our method and ideas, the well-designed experiments (S2), significant step reduction in the learned optimizer application, and the paper's clear presentation.
>
> We also appreciate the constructive feedback and address the questions and concerns below. We believe these revisions significantly strengthen the paper, and we would genuinely appreciate a reconsideration of the score.
> - **Total Training-Time Speedup (W1, Q1a)** We thank the reviewer for this useful suggestion. For an iterations/second comparison of GradMetaNet-based optimizers we refer the reviewer to Table 7 in Appendix F.2. Even with a slight overhead per-step, the significant reduction in steps leads to GradMetaNet having a **substantial speedup in train-time (up to 3x faster than Adam)**. Inspired by the reviewer’s comment, we compute the training-time needed to reach Adam's best test loss, and report the total train-time speedup below.
>   - **Fashion MNIST**
> |Optimizer |Step Reduction vs. Adam (↑)|Train-Time Speedup vs. Adam (↑)|
> |-|-|-|
> |UNF|1.16x|1.05x|
> |GradMetaNet + UNF|**1.51x**|**1.13x**|
>   - **CIFAR10**
> |Optimizer|Step Reduction vs. Adam (↑)|Train-Time Speedup vs. Adam (↑)|
> |-|-|-|
> |UNF|2.64x |2.31x |
> |GradMetaNet + UNF|**4.26x**|**3.13x**|
>   - **CIFAR100**
> |Optimizer|Step Reduction vs. Adam (↑)|Train-Time speedup vs. Adam (↑)|
> |-|-|-|
> |UNF|1.58x |1.34x|
> |GradMetaNet + UNF|**2.85x**|**2.10x**|
>   - **LM1B**
> |Optimizer|Step Reduction vs. Adam (↑)|Train-Time speedup vs. Adam (↑)|
> |-|-|-|
> |UNF|1.48x |1.04x|
> |GradMetaNet + UNF|**1.82x**|**1.11x**|
>
>   GradMetaNet-based optimizers **outperform all baselines** in total training speed. To fit within the character limit we only include here the results for the top optimizers, and will add a comprehensive table with the train-time comparisons to the revised paper.
>
>   Additionally, we provide a computational complexity analysis in Appendix D.4 and show that a forward pass through GradMetaNet has complexity $O(N \cdot b)$, where $N$ is the number of neurons in the network we optimize and b is the batch size. Therefore, GradMetaNet **does not change the asymptotic complexity of backpropagation** on the original model. We will make this point more visible  in the revised manuscript.
>
>   Finally, to provide context regarding the model size for other applications of metanetworks, we note that our experiments follow standard benchmarks in learned optimization and the weight-space learning community [1, 2, 3, 7, 8] and include **strictly larger models (up to 2M parameters) than key weight-space/metanetwork papers**. For instance [1] (ICML 2023 oral presentation), [2] (ICLR 2024 oral presentation), and [3] (NeurIPS 2024 oral presentation), operate on models of up to 200K parameters. We thank the reviewer again for raising this important point, which we believe will strengthen the paper.
>
> - **Scaling Curvature Estimation Experiment (Q2)** We thank the reviewer for this suggestion. We have conducted an extended version of the experiment in Section 7.1 with **models containing over 1M parameters**. Our networks are trained to predict the trace of the Fisher Information Matrix. At this scale, many of the original baselines, particularly full-gradient methods, are computationally infeasible. Therefore, as a scalable comparison we use an MLP, also operating on decomposed gradients. **GradMetaNet beats the MLP** with a normalized test MAE of 0.413 compared to 0.779 (lower is better). We will include this large-scale curvature estimation experiment in the final version of the paper. Additionally, we refer the reviewer to our learned optimizer experiments on LM1B which also demonstrate GradMetaNet’s effectiveness at the **1M+ parameter scale**.
> |Model|Normalized Test MAE (↓)|
> |-|-|
> |MLP|0.779|
> |GradMetaNet|**0.413**|
>
> - **Accounted Symmetries (W2, Q3)** We thank the reviewer for this insightful question. It is indeed true that neural architectures often exhibit symmetries beyond simple neuron permutations. While MLP weights do admit rescaling symmetries, these **do not always extend to gradient-based quantities**. For example, in Section 7.1, we learn the diagonal of the Fisher Information Matrix (FIM), which is 2-homogeneous in the gradients: $\mathrm{diag}(\mathbf{F}(\lambda \mathbf{W})) = \lambda^2 \mathrm{diag}(\mathbf{F}(\mathbf{W}))$. This indicates that enforcing rescaling equivariance **would be inappropriate for such tasks**. We therefore focus on permutation equivariance [4, 5, 6], which is general, task-agnostic, and remains valid across all gradient-level problems considered in this work. We briefly discuss broader symmetry considerations, including transformer $\mathrm{GL}$-symmetries,  in Appendix B.2 and believe this is an incredibly compelling direction for future research.
>
> - **Computational Complexity Clarification (Q1b)** Yes, the reviewer is exactly right, we are referring to the specific case where $D_\mathrm{in} = D_\mathrm{out}$ in the paper. We'll clarify this in the revised text. To be more precise, we will emphasize the general point that the cost is additive rather than multiplicative with respect to the hidden dimensions. This means that instead of a cost that scales with the product of hidden dimensions, it scales with their sum. We will include a formula to illustrate this more clearly in the updated version. We thank the reviewer for raising this point.
>
> - **Fisher Information Matrix Clarification (Q4)** The reviewer's observation about the Fisher Information Matrix is accurate. In the first experiment (loss landscape curvature estimation), we use the full "true" Fisher as the target for prediction, while for the learned optimizer experiments, GradMetaNet is provided with the necessary information to compute the empirical Fisher (in the form of sets of individual gradients). While (as the reviewer points out) GradMetaNet is capable of supporting other curvature matrices, such as the "true" Fisher (by backpropagating gradients of pseudo-losses) or the Generalized Gauss-Newton, in the learned optimizers experiment, we chose to focus on the empirical Fisher to simplify the experimental setup and avoid additional complexities like sampling labels. We agree that making this distinction clearer will enhance the precision of the paper, and will integrate a more detailed explanation, as well as the suggested referenced in the revised manuscript.
>
> - **Formatting** We thank the reviewer for spotting these, and agree with their suggestions for improvement. The sentence in L199 should be: “...concatenates a layer identifier **to each neuron** in the intermediate layers and a neuron identifier to each neuron in the first and last layers”. We will implement the suggestions and corrections, and add the reference mentioned by the reviewer in the revised version.
>
> We thank the reviewer again for their constructive feedback. We believe we have thoroughly addressed the concerns raised and clarified the key contributions of our work. We hope this response helps convey the computational efficiency, and practical relevance of GradMetaNet, and would greatly appreciate a reconsideration of the score.
>
> **References**
>
> [1] Navon, Aviv, et al. "Equivariant architectures for learning in deep weight spaces." International Conference on Machine Learning. PMLR, 2023.
>
> [2] Kofinas, Miltiadis, et al. "Graph neural networks for learning equivariant representations of neural networks." arXiv preprint arXiv:2403.12143 (2024).
>
> [3] Kalogeropoulos, Ioannis, Giorgos Bouritsas, and Yannis Panagakis. "Scale equivariant graph metanetworks." Advances in neural information processing systems 37 (2024): 106800-106840.
>
> [4] Zaheer, Manzil, et al. "Deep sets." Advances in neural information processing systems 30 (2017).
>
> [5] Hartford, Jason, et al. "Deep models of interactions across sets." International Conference on Machine Learning. PMLR, 2018.
>
> [6] Maron, Haggai, et al. "On learning sets of symmetric elements." International conference on machine learning. PMLR, 2020.
>
> [7] Schürholt, Konstantin, et al. "Model zoos: A dataset of diverse populations of neural network models." Advances in Neural Information Processing Systems 35 (2022): 38134-38148.
>
> [8] Schürholt, Konstantin, et al. "Neural network weights as a new data modality." ICLR 2025 Workshop Proposals.

---

> > ### Comment · Reviewer_EMd8 · 2025-08-04
> >
> > Thanks for your response and the additional experimental clarifications, which strengthen my view of the paper.
> >
> > Could you elaborate a bit more about why equivariance is not always a desired property for gradient-based quantities like the Fisher? I see how permutations are more special compared to arbitrary invertible matrices, but am wondering what would be the 'desirable' transformation carried out by a meta-network in the presence of richer symmetries.
> > To me, this points to a small hole in the paper's motivation (along the lines 'accounting for symmetries should be done, but we don't always know how to do it properly').
> >
> > Thanks!

---

> > > ### Author Response · Authors · 2025-08-05
> > > **Authors' Response to Followup Question**
> > >
> > > We thank the reviewer for their positive evaluation, and are happy that our rebuttal "strengthens their view of the paper”. We also appreciate this important follow-up question, as it touches on a nuanced aspect of designing equivariant metanetworks that we are happy to elaborate on. Given the reviewer’s expressed appreciation of our paper and responses, as well as the additional clarifications provided below, we hope that they will consider revising their score.
> > >
> > > We would like to clarify that equivariance **is** always a desired property for gradient-based metanetworks. However, for a metanetwork to be effective, its equivariance properties **must match those of the function it is approximating**. As we have discussed, not all weight-space symmetries translate directly to gradient-based symmetries. Some symmetries only apply to a specific type of architecture (e.g. transformers) or a specific type of task (e.g. the FIM diagonal). Therefore, the primary reason for focusing on permutation symmetry is that it is **universally correct** for a diverse set of gradient-based learning tasks and architectures.
> > >
> > > That being said, GradMetaNet **can be easily extended to support other, less general, symmetries**. Inspired by the reviewer’s question, we elaborate here on the required adaptations to respect attention $\mathrm{GL}$-symmetries and scaling symmetries.
> > >
> > > - **Attention $\\mathrm{GL}$-Symmetries** In the notation of Appendix B.2, where these symmetries are also discussed, $\\mathrm{GL}$-symmetries are expressed by $\\mathbf{Q}^{(l, j)} \\mapsto \\mathbf{A} \\mathbf{Q}^{(l, j)}, \\mathbf{K}^{(l, j)} \\mapsto (\\mathbf{A}^{-1})^\\top \\mathbf{K}^{(l, j)}$ for the key and query activations, and $\\mathbf{g}_Q^{(l, j)} \\mapsto \mathbf{A}^{-1} \\mathbf{g}_Q^{(l, j)}, \\mathbf{g}_K^{(l, j)} \\mapsto \\mathbf{A}^\\top \\mathbf{g}_K^{(l, j)}$ for the key and query gradients. As derived in [1], the most general linear equivariant layers that respect this symmetry are exactly the ones that act only on the sequence dimension, leaving the feature dimension as the identity. Replacing GradMetaNet's equivariant layers with these types of layers (which in our case become sequence/batch-wise DeepSets layers) when processing attention activations and gradients, would result in an architecture that is equivariant to $\\mathrm{GL}$-symmetries. Standard GradMetaNet layers would still be used for the other transformer components.
> > > - **Scale Symmetries** First, we adapt the scale symmetries of weight-space [2] to our setting of decomposed gradients. If the weights $\\mathbf{W}^{(l)}$ and $\\mathbf{W}^{(l+1)}$ are scaled by diagonal matrices $\\mathbf{D} = \\mathrm{diag}(s_1, \\dots, s_d)$ and $\\mathbf{D}^{-1} = \\mathrm{diag}(s_1^{-1}, \\dots, s_d^{-1})$ respectively, then the activations $\\mathbf{a}^{(l)}$ are scaled by $\\mathbf{D}$, while the pre-activation gradients $\\mathbf{g}^{(l)}$ remain unchanged. This induces the following scaling symmetry action on the decomposed gradients: $\\mathbf{D} \star (\\mathbf{a}^{(l)}, \\mathbf{g}^{(l)}) = (\\mathbf{D} \\mathbf{a}^{(l)}, \\mathbf{g}^{(l)}).$
> > >   These scale symmetries act independently on the activation component of each layer. Since the features corresponding to the pre-activation gradients remain invariant under this action, they can be processed exactly as in the original method. The activation features, however, require special treatment. We adopt a similar strategy to that used in [2]: In each layer we (1) apply a linear transformation to each activation feature, (2) in parallel, we normalize the features (i.e. divide by their norm) and process the result through an MLP. The outputs of these two processes are first multiplied element-wise and then multiplied with the current representation of the gradient features. We will include a full description of this approach in the revised manuscript.
> > >
> > > In conclusion, GradMetaNet is a principled architecture that is equivariant to universally applicable permutation symmetries, and it can be easily extended to support other types of gradient-space symmetries. We thank the reviewer for the engaging discussion, and would greatly appreciate a reconsideration of the score.
> > >
> > > ---
> > >
> > > [1] Putterman, Theo, et al. "Learning on loras: GL-equivariant processing of low-rank weight spaces for large finetuned models."
> > >
> > > [2] Kalogeropoulos, Ioannis, et al. "Scale equivariant graph metanetworks." NeurIPS 2024.

---

> > > > ### Comment · Reviewer_EMd8 · 2025-08-05
> > > >
> > > > Thank you for your detailed and comprehensive response.
> > > > I believe adding this discussion (how to account for other symmetries) further improves the work by increasing its scope and highlighting its generality.
> > > >
> > > > I am leaning towards raising my score and have no further questions to the authors.

---

> > > > > ### Author Response · Authors · 2025-08-08
> > > > >
> > > > > We sincerely thank the reviewer for their thoughtful and detailed feedback and for their engagement during the discussion phase. We're especially grateful for the recognition of GradMetaNet's contributions and for the insightful questions regarding equivariant modeling, which helped us further clarify and strengthen our work. We’re glad our responses could enhance their view of the paper, and for their willingness to revisit their evaluation
> > > > >
> > > > > Best,
> > > > >
> > > > > The authors

---

### Official Review · Reviewer_wdp4 · 2025-07-04

**Clarity:** 3
**Significance:** 2
**Originality:** 2
**Rating:** 4
**Confidence:** 1

**Summary:**

This paper introduces GradMetaNet, to design the neural network architecture specifically for learning on gradients. The key point is an approach guided by three main ideas: (1) using a rank-1 decomposition of gradients for a structured representation, (2) processing sets of gradients from multiple data samples to capture local loss landscape information, and (3) designing the architecture to be equivariant to the permutation symmetries of neurons in the underlying model. The authors provide theoretical insights for their approach, including a universality theorem and a proof that methods using only average gradients are fundamentally less expressive. They demonstrate the effectiveness of GradMetaNet on several tasks and shows improvements over previous methods.

**Questions:**

Many questions are already covered in the weakness sections. Below I put some of them explicitly:
- Table 7 indicates that GradMetaNet introduces a non-trivial computational overhead per step. While Table 1 shows impressive reductions in the number of steps to convergence, this does not directly translate to a reduction in total training time. Could the authors provide a comparison of the total time required for each optimizer to reach the target test loss?
- The experiments are done on relatively small models. How does the performance and memory trade-offs go when scaling GradMetaNet to large-scale architectures (e.g., LLMs with billions of parameters)? Maybe a LLM with 100/200M parameters is already good enough for this purpose.
- Storing a set of decomposed gradients for a large batch (b) across a massive model could become a memory bottleneck. How does this compare to existing methods like K-FAC or Shampoo in large-scale settings?
- The learned optimizer experiments involve meta-training on a specific family of tasks and architectures. How well would a GradMetaNet-based optimizer, meta-trained on the small models, generalize to different or larger architectures? Does the improved curvature awareness grant better generalization, or would it require expensive re-meta-training for new domains?

**Ethical Concerns:**

["NO or VERY MINOR ethics concerns only"]

**Final Justification:**

Most of concerns get resolved in the reviewer's rebuttal. So I will raise my rating to 4.

**Limitations:**

Please check the weakness and question sections for more details.

**Quality:**

3

**Strengths And Weaknesses:**

Strengths:
- The paper provides the theoretical results for its claims. It proves that processing sets of gradients is more powerful than processing the average gradient for key tasks like natural gradient approximation.
- The experiments include good ablations and comparisons that show the benefits of each design principle. By comparing against methods that use full gradients (vs. decomposed), average gradients (vs. sets), and non-equivariant architectures, the authors provide convincing evidence for each of their core ideas .

Weaknesses:
- The experiments are using relatively small-scale models. The image classification use MLPs on downsized 8x8 images, and the language model is a small 2-block transformer. The paper acknowledges scaling to state-of-the-art LLMs as future work, but the absence of results on larger, more modern architectures is a significant limitation.
- The proposed method introduces computational overhead compared to standard approaches. Table 7 shows that GradMetaNet is slower per iteration. While the method may converge in fewer steps, the increased time per step is not fully explored in the results. The attention-based GradMetaNet++ is particularly expensive, with quadratic complexity in the number of neurons and batch size .
- The paper's novelty lies in the clever synthesis and application of several existing concepts rather than the invention of fundamentally new ones. The architecture uses equivariant layers adapted from prior work on set and graph processing (e.g., DeepSets) , and the idea of using rank-1 gradient factorizations is central to established methods like K-FAC.
- For the learned optimizer application, the paper relies on a meta-training setup, which could be expensive and complex. It is unclear how well an optimizer meta-trained on the paper's small-scale tasks would generalize to different, larger architectures.

---

> ### Author Rebuttal · Authors · 2025-07-30
>
> We thank the reviewer for their thoughtful and detailed feedback. We are pleased they recognize the paper's theoretical contributions (S1), its clear demonstration of the benefit of each design principle (S2), and the "impressive reductions in the number of steps" in the learned optimization application. We also appreciate the constructive criticism and questions, which we address below.
>
> - **Total Training-Time Speedup (W2, Q1)** We thank the reviewer for this useful suggestion. Even with a slight overhead per-step, the significant reduction in steps leads to GradMetaNet having a **substantial speedup in train-time (up to 3x faster than Adam)**. Inspired by the reviewer’s comment, we compute the training-time needed to reach Adam's best test loss, and report the total train-time speedup below.
>   - **Fashion MNIST**
> |Optimizer|Step Reduction vs. Adam (↑)|Train-Time Speedup vs. Adam (↑)|
> |-|-|-|
> |UNF|1.16x|1.05x|
> |GradMetaNet + UNF|**1.51x**|**1.13x**|
>   - **CIFAR10**
> |Optimizer|Step Reduction vs. Adam (↑)|Train-Time Speedup vs. Adam (↑)|
> |-|-|-|
> |UNF|2.64x |2.31x |
> |GradMetaNet + UNF|**4.26x**|**3.13x**|
>   - **CIFAR100**
> |Optimizer|Step Reduction vs. Adam (↑)|Train-Time speedup vs. Adam (↑)|
> |-|-|-|
> |UNF|1.58x |1.34x|
> |GradMetaNet + UNF|**2.85x**|**2.10x**|
>   - **LM1B**
> |Optimizer|Step Reduction vs. Adam (↑)|Train-Time speedup vs. Adam (↑)|
> |-|-|-|
> |UNF|1.48x |1.04x|
> |GradMetaNet + UNF|**1.82x**|**1.11x**|
>
>   GradMetaNet-based optimizers **outperform all baselines** in total training speed. To fit within the character limit we only include here the results for the top optimizers, and will add a comprehensive table with the train-time comparisons to the revised paper.
>
> - **GradMetaNet Generalizes to New Tasks and Model Sizes (W4, Q4)** Inspired by the reviewer's comment, we have conducted new experiments that demonstrate **GradMetaNet-based learned optimizers are able to generalize across tasks and model sizes**. We note that this capability is not possible for learned optimizers that are based on other weight space models, such as DWS [1], NFN [4], or UNF [7], which require a defining a different weight-space model (metanetwork) to process gradients of different base architectures.
>   - **Model Size Generalization** For both the CIFAR-10 and CIFAR-100 tasks, we meta-trained an optimizer for a specific MLP architecture and then evaluated its ability to optimize a larger, unseen MLP, without any re-meta-training. Specifically:
>     - For CIFAR-10, we meta-train on a 32-hidden-dim MLP and meta-test on a 64-hidden-dim MLP.
>     - For CIFAR-100, we meta-train on a 128-hidden-dim MLP and meta-test on a 256-hidden-dim MLP.
>
>     In both cases, the optimizer **generalized successfully to the larger model**.
>     - **CIFAR10 (32 -> 64)**
>     |Optimizer|Step Reduction vs. Adam (↑)| Train-Time Speedup vs. Adam (↑)|
>     |-|-|-|
>     |DS (32 -> 64)|2.18x|1.85x|
>     |GradMetaNet + DS (32 -> 64)|**4.15x**|**3.05x**|
>     - **CIFAR100 (128 -> 256)**
>     |Optimizer|Step Reduction vs. Adam (↑) | Train-Time Speedup vs. Adam (↑)|
>     |-|-|-|
>     |DS (128 -> 256)|1.78x |1.59x|
>     |GradMetaNet + DS (128 -> 256)| **3.04x** |**2.37x**|
>
>   - **Task Generalization** We then meta-train optimizers on CIFAR10 and meta-test them on CIFAR100 and vice versa. In both cases, the optimizer **generalized successfully to the new task**. Note that this experiment also includes size generalization as the output dimension required for these tasks is different.
>     - **CIFAR100 -> CIFAR10 (128 hidden dim)**
>     | Optimizer |Step Reduction vs. Adam (↑) | wall-clock speedup vs. Adam (↑)|
>     |-|-|-|
>     |DS|2.45x |1.72x|
>     |DS + GradMetaNet|**3.57x**|**2.83x**|
>     - **CIFAR10 -> CIFAR100 (32 hidden dim)**
>     |Optimizer |Step Reduction vs. Adam (↑) | Train-Time Speedup vs. Adam (↑)|
>     |-|-|-|
>     |DS|1.93x|1.57x|
>     |DS + GradMetaNet|**3.80x**|**2.59x**|
>
> - **Scaling GradMetaNet (W1, Q2)** We appreciate the reviewer's comment regarding the importance of scaling. First, we highlight that in order to ensure a fair comparison, our experimental design follows **established benchmarks** and uses the **exact setup** of established papers in learned optimization and weight-space learning [8]. We elaborate in the context each experiment:
>   - **Learned Optimization** Scaling learned optimizers is a long-standing challenge due to high computational demands. For example, VeLO [5] was trained for over **~4,000 TPU-months**. As such, training optimizers at this scale is currently infeasible with our academic compute resources, **regardless of whether GradMetaNet is used**. In our paper, we follow the exact setup of prior work [4, 6], conducting evaluations in settings of up to 2M parameters to analyze architectural improvements.
>
>     That said, inspired by the reviewer’s suggestion, we evaluated the update time and memory cost of a GradMetaNet-based optimizer on (the gradients of) a GPT-2–scale model. We used a 12-layer Transformer with 12 attention heads, hidden size of 768, and approximately **117M parameters**. On two NVIDIA A100-SXM4-40GB GPUs, a GradMetaNet update takes 1.29s (including backpropagationg through the base transformer model), compared to 0.89s using Adam, and has a similar memory footprint.
>     |Optimizer|Update Time|Memory Footprint|
>     |-|-|-|
>     |Adam|0.89s|77.2 GB|
>     |GradMetaNet|1.29s |77.8 GB|
>   - **INR Editing** Our benchmarks **exactly follow** recent state-of-the-art weight-space methods [2, 3, 7] (2023-2024). GradMetaNet achieves a **22.5% improvement over prior SOTA** (Table 2), demonstrating its effectiveness on established, non-trivial tasks widely used by the community [8].
>   - **Curvature Estimation** Some baselines we compare against are not scalable by nature, and thus the setting is chosen accordingly to isolate the effect of architecture. That said, inspired by reviewer **EMd8**, we have scaled this experiment too to models with 1M parameters.
>
>   Additionally, we note that our experiments include **strictly larger models** (up to 2M parameters) than key weight-space papers. For instance [1] (ICML 2023 oral presentation), [2] (ICLR 2024 oral presentation), and [3] (NeurIPS 2024 oral presentation), all operate on models of up to 200K parameters.
> - **GradMetaNet's Novelty (W3)** We appreciate the reviewer’s characterization of our work as a "clever synthesis". We wish to convey that our contribution extends beyond combining existing concepts. The core methodological novelty lies in:
>   - **First Principles Approach** Our architecture is derived from first principles, using equivariant deep learning techniques [9]. We begin by analyzing the symmetry group and action associated with the space of decomposed gradients, a novel mathematical setting that **has not been studied before**, providing a principled foundation for reasoning about gradient representations.
>   - **Architecture Derivation** Building on this symmetry analysis, we mathematically derive a class of equivariant layers. An elegant outcome of this derivation is that it naturally recovers and generalizes a range of existing architectures.
>   - **Theory** Our novel formulation also enables us to prove **new results** that are **unknown for other weight-space models**. We establish a universality theorem for functions over sets of gradients (Theorem 6.2), a result that **has not been shown for any prior method** [1, 2, 3, 4, 7]. Furthermore, we show that existing models are **provably less expressive** for core gradient-based tasks (Corollary 6.3). We thank the reviewer for raising the point and will make this discussion more clear in the updated version.
>
> - **Memory Improvement over K-FAC/Shampoo (Q3)** This is an excellent question. GradMetaNet's memory scaling is one of its key advantages. **GradMetaNet is significantly more memory-efficient than K-FAC and Shampoo**, as it does not require storing large curvature matrices. Specifically, K-FAC and Shampoo store matrices consisting of outer products of activations and preactivation gradients, leading to a memory footprint of $O(d_0^2+\cdots+d_L^2)$, where $d_i$ is the $i$-th hidden dimension. In contrast, GradMetaNet leverages the batch of activations and preactivation gradients that are **already present in memory** from the backpropagation computation. Our method only incurs an additional memory cost for a momentum term, which stores a moving average of these activations and preactivation gradients. This results in a strictly lower memory footprint of $O(d_0+\cdots+d_L)$. We thank the reviewer for raising this point, we will make this discussion more clear in the updated version.
>
> We thank the reviewer again for their constructive feedback. We believe we have thoroughly addressed the reviewer's concerns and clarified the key contributions of our work. We hope this response helps to convey the novelty, scalability, generality, and practical relevance of GradMetaNet, and would greatly appreciate a reconsideration of the score.
>
> **References**
>
> [1] Navon, Aviv, et al. "Equivariant architectures for learning in deep weight spaces." ICML 2023.
>
> [2] Kofinas, Miltiadis, et al. "Graph neural networks for learning equivariant representations of neural networks." ICLR 2024.
>
> [3] Kalogeropoulos, Ioannis, et al. "Scale equivariant graph metanetworks." NeurIPS 2024.
>
> [4] Zhou, Allan, et al. "Universal neural functionals." NeurIPS 2024.
>
> [5] Metz, Luke, et al. "Velo: Training versatile learned optimizers by scaling up." arXiv preprint arXiv:2211.09760 (2022).
>
> [6] Harrison, James, et al. "A closer look at learned optimization: Stability, robustness, and inductive biases." NeurIPS 2022.
>
> [7] Zhou, Allan, et al. "Permutation equivariant neural functionals." NeurIPS 2023.
>
> [8] Schürholt, Konstantin, et al. "Neural network weights as a new data modality." ICLR 2025 Workshop Proposals.
>
> [9] Bronstein, Michael M., et al. "Geometric deep learning: Grids, groups, graphs, geodesics, and gauges." arXiv preprint arXiv:2104.13478 (2021).

---

> > ### Comment · Reviewer_wdp4 · 2025-08-05
> > **Response to rebuttal**
> >
> > Thanks for the reviewer's efforts. Most of my concerns are solved. I will raise my ratings.

---

> > > ### Author Response · Authors · 2025-08-08
> > >
> > > We sincerely thank the reviewer for their thoughtful feedback, for taking the time to consider our rebuttal, and for raising their score. We appreciate the recognition of our theoretical contributions, the clarity of our ablations, and the design of GradMetaNet. We're also grateful for the constructive questions, which motivated us to run new experiments and clarify key points that will strengthen the paper.
> > >
> > > Best,
> > >
> > > The Authors

---

### Comment · Area_Chair_8cYw · 2025-08-08
**A few clarifications**

Dear Authors,

Thank you for your continued efforts in the discussion.

In light of the additional experiments presented in the rebuttal, could you clarify whether they are conducted in settings consistent to the current experiments (section F), a list of changes that you plan to make to the current experimental section, and how these changes alter any of the original claims or conclusions?

Best,

AC

---

> ### Author Response · Authors · 2025-08-08
> **Response to AC Clarifications**
>
> Dear AC,
>
> Thank you for your question.
>
> All additional experiments presented in our rebuttal were conducted using the **exact same settings as those described in Section F** of the paper and are natural extensions of the existing experiments.
> In the revision, we plan to:
>
> - Add a table for the meta-optimizers experiment, reporting convergence time in seconds in addition to the previously reported number of  steps.
> - Extend the meta-optimizer experiments to include task and architecture generalization demonstrations.
> - In the Appendix, we will add a brief evaluation of GradMetaNet’s ability to scale to larger settings (e.g., a 117M-parameter LLM).
>
> These additions **do not alter any of our original claims or conclusions**. Instead, they reinforce our findings and provide further insight,  such as:
> - GradMetaNet speeds up total training time, not just the number of training steps.
> - GradMetaNet-based learned optimizers generalize effectively to new tasks and architectures.
> - GradMetaNet shows promising initial results suggesting strong potential to scale to larger settings with industrial-grade compute.
>
> We appreciate your engagement and look forward to the next steps.
>
> Best,
>
> The Authors

---

### Note · Authors · 2025-08-12

We thank the reviewers and AC for an engaging discussion. We are grateful to reviewer **wdp4** for raising their score and to reviewer **EMd8** for their openness to revisiting their evaluation and stated intention to raise the score.

Reviewers highlighted:

- **Strong theory & expressivity**
  - **wdp4** states that “the paper provides the theoretical results for its claims,” and that we prove "sets of gradients are more powerful than averaged gradients".
  - **EMd8** highlights theoretical results showing "GradMetaNet can represent functions not representable with averaged weight-space gradients".

- **Rigorous empirical validation & ablations**
  - **wdp4** praises “good ablations and comparisons” showing "the benefits of each design principle".
  - **EMd8** notes that experiments are "well designed to disentangle" the utility of "per-datum information and symmetry-awareness".

- **Principled & general methodology improvement**
  - **EMd8** notes a “meaningful improvement over previous architectures" and that the methodology applies to “basically any neural network.”
  - **cxim** highlights the motivated use of equivariance, calling the design “novel and well-founded”.

- **Strong performance vs. baselines & effectiveness across tasks**
  - **wdp4** mentions that our results “show improvements over previous methods” across tasks.
  - **EMd8** says that GradMetaNet is useful for learning "curvature information, optimization algorithms, and editing the weight space of neural nets".
  - **cxim** GradMetaNet “outperforms several baselines” on evaluated tasks.

- **Worthy of publication:** explicitly noted by reviewer **cxim**.

Clarifications from the discussion include:

- **Train time vs. steps** demonstrated that step reductions translate into substantial wall-clock speedups.
- **Learned-optimizer generalization:** demonstrated that our optimizers transfer to larger, unseen architectures and different tasks.
- **Scaling clarifications** elaborated on advantages over K-FAC/Shampoo, ran GradMetaNet on a GPT-2–scale model, and extended experiments to 1M+ parameter settings.
- **Arch. extensions** elaborated on applicability to attention and scale symmetries as well as to to CNNs.

We believe the discussion and resulting clarifications, which *validate the claims of our original submission*, strengthen the paper. We are confident it is ready for publication and look forward to sharing this work with the community.

---

### Decision · Program_Chairs · 2025-09-17

**Decision:**

Accept (poster)

**Comment:**

This paper proposes a novel, principled architecture GradMetaNet for learning on gradients, which leverages symmetry of the neuromanifold, related invariances, and rank-1 structure of the gradients. The authors provided a universal approximation theorem for such invariant mappings, as well as empirical evaluations on precipitating the diagonal FIM and learned optimization.

All three reviewers acknowledge the novelty and rich expressivity that is guaranteed in theory, making the proposed architecture a meaningful advance over previous architectures. The empirical results are carefully designed and comprehensive, with a notable performance gain compared with weight-space approaches. The writing flows well with high quality illustrations.

A main concern is the scalability due to the computational overhead, and the authors clarified in the rebuttal that the overall computation is more efficient due to reduction in the number of training steps. Further experiments show that GradMetaNet can scale to GPT-2–scale mode with 1M+ parameters. Another limitation is the focus on MLP and transformers, though extensions appear plausible.

Overall, the strength outweigh the limitations, and I recommend acceptance. For the final version, the authors are recommended to carefully address the reviewers' comments and further clarify scalability and broader applicability.